# Improving Sample Efficiency of Model-Free Algorithms for Zero-Sum Markov Games

## Abstract

The problem of two-player zero-sum Markov games has recently attracted increasing interests in theoretical studies of multi-agent reinforcement learning (RL). In particular, for finite-horizon episodic Markov decision processes (MDPs), it has been shown that model-based algorithms can find an $\epsilon$-optimal Nash Equilibrium (NE) with the sample complexity of $O(H^3 SAB/\epsilon^2)$, which is optimal in the dependence of the horizon $H$ and the number of states $S$ (where $A$ and $B$ denote the number of actions of the two players, respectively). However, none of the existing model-free algorithms can achieve such an optimality. In this work, we propose a model-free stage-based algorithm and show that it achieves the same sample complexity as the best model-based algorithm, and hence for the first time demonstrate that model-free algorithms can enjoy the same optimality in the $H$ dependence as model-based algorithms. The main improvement of the dependency on $H$ arises by leveraging the popular variance reduction technique based on the reference-advantage decomposition previously used only for single-agent RL. However, such a technique relies on a critical monotonicity property of the value function, which does not hold in Markov games due to the update of the policy via the coarse correlated equilibrium (CCE) oracle. Thus, to extend such a technique to Markov games, our algorithm features a key novel design of updating the reference value functions as the pair of optimistic and pessimistic value functions whose value difference is the smallest in the history in order to achieve the desired improvement in the sample efficiency.

## 1 Introduction

Multi-agent reinforcement learning (MARL) commonly refers to the sequential decision making framework, in which more than one agent learn to make decisions in an unknown shared environment to maximize their cumulative rewards. MARL has achieved great success in a variety of practical applications, including the game of GO [29; 30], real-time strategy games involving team play [32], autonomous driving [26], and behavior learning in complex social scenarios [6]. Despite the great empirical success, one major bottleneck for many RL algorithms is that they require enormous samples. For example, in many practical MARL scenarios, a large number of samples are often required to achieve human-like performance due to the necessity of exploration. It is thus important to understand how to design sample-efficient algorithms.

As a prevalent approach to the MARL, model-based methods use the existing visitation data to estimate the model, run a planning algorithm on the estimated model to obtain the policy, and execute the policy in the environment. In two-player zero-sum Markov games, an extensive series of studies [3; 39; 22] have shown that *model-based* algorithms are provably efficient in MARL, and can achieve minimax-optimal sample complexity $O(H^3 SAB/\epsilon^2)$ except for the term $AB$ [39; 22], where $H$ denotes the horizon, $S$ denotes the number of states, and $A$ and $B$ denote the numbers of actions of the two players, respectively. On the other hand, *model-free* methods directly estimate the (action-)value functions at the equilibrium policies instead of estimating the model. However, none of the existing *model-free* algorithms can achieve the aforementioned optimality (attained by model-based algorithms) [5; 23; 31; 18; 24]. Specifically, the number of episodes required for model-free algorithms scales sub-optimally in step $H$, which naturally motivates the following open question:

**Can we design model-free algorithms with the optimal sample dependence on the time horizon for learning two-player zero-sum Markov games?**

In this paper, we give an affirmative answer to the above question. We highlight our main contributions as follows.

**Algorithm design.** We design a new model-free algorithm of Q-learning with **min-gap** based reference-advantage decomposition. In particular, we extend the reference-advantage decomposition technique [40] proposed for single-agent RL to zero-sum Markov games with the following key novel design. Unlike the single-agent scenario, the optimistic (or pessimistic) value function in Markov games does not necessarily preserve the monotone property due to the nature of the CCE oracle. In order to obtain the "best" optimistic and pessimistic value function pair, we update the reference value functions as the pair of optimistic and pessimistic value functions whose value difference is the smallest (i.e., with the minimal gap) in the history. Moreover, our algorithm relies on the stage-based approach, which simplifies the algorithm design and subsequent analysis.

**Sample complexity bound.** We show that our algorithm provably finds an $\epsilon$-optimal Nash equilibrium for the two-player zero-sum Markov game in $\widetilde{O}(H^3 SAB/\epsilon^2)$ episodes, which improves upon the sample complexity of all existing model-free algorithms for zero-sum Markov game. Further, comparison to the existing lower bound shows that it is minimax-optimal on the dependence of $H$, $S$ and $\epsilon$. This is the first result that establishes such optimality for model-free algorithms, although model-based algorithms have been shown to achieve such optimality in the past [22].

**Technical analysis.** We establish a few new properties on the cumulative occurrence of the large V-gap and the cumulative bonus term to enable the upper-bounding of several new error terms arising due to the incorporation of the new min-gap based reference-advantage decomposition technique. These properties have not been established for the single-agent RL with such a technique, because our properties are established for policies generated by the CCE oracle in zero-sum Markov games. Further, the analysis of both the optimistic and pessimistic accumulative bonus terms requires a more refined analysis compared to their counterparts in single-agent RL [40].

## 1.1 RELATED WORK

**Markov games.** The Markov game, also known as the stochastic game, was first proposed in [27] to model the multi-agent RL. Early attempts to find the Nash equilibra of Markov games include [21; 14; 13; 35]. However, they often relied on strong assumptions such as known transition matrix and reward, or focused on the asymptotic setting. Thus, these results do not apply to the non-asymptotic setting where the transition and reward are unknown and only limited data is available.

There is a line of works focusing on non-asymptotic guarantees with certain reachability assumptions. A popular approach is to assume access to simulators, which enables the agent to sample transition and reward directly for any state-action pair [16; 28; 39; 20]. Alternatively, [34] studied the Markov game under the assumption that one player can always reach all states by playing certain policy no matter what strategy the other player sticks to.

**Two-player zero-sum games.** [3; 36] initialized the study of non-asymptotic guarantee for two-player zero-sum Markov games without reachability assumptions. [3] proposed a model-based algorithm for tabular Markov game while [36] considered linear function approximation in game and adopted a model-free approach. [22] proposed a model-based algorithm which achieves the minimax-optimal samples complexity $O(H^3 SAB/\epsilon)$ except for the $AB$ term. For the discounted setting and having access to a generative model, [39] developed a model-based algorithm that achieves the minimax-optimal sample complexity except for the $AB$ term. Then, model-free Nash Q-learning and Nash V-learning were proposed in [5] for two-player zero-sum game to achieve optimal dependence on actions (i.e., $(A + B)$ instead of $AB$). Further, [8; 15] studied the two-player zero-sum game under linear and general function approximation.

**Multi-player general-sum games.** [22] developed model-free algorithm in episodic setting, which suffers from the curse of multi-agent. To alleviate this issue, [23; 31; 18; 24] proposed V-learning algorithm, coupled with the adversarial bandit subroutine, to break the curse of multi-agent. [23] considered learning an $\epsilon$-optimal CCE and used V-learning with stabilized online mirror descent as the adversarial bandit subroutine. Both [31; 18] utilized the weighted follow the regularized

leader (FTRL) algorithm as the adversarial subroutine, and considered $\epsilon$-optimal CCE and $\epsilon$-optimal correalted equilibrium (CE). The work [24] featured the standard uniform weighted FTRL and staged-based design, both of which simplifies the algorithm design and the corresponding analysis. While the V-learning algorithms generate non-Markov, history dependent policies, [11; 33] learned an approximate CCEs that is guaranteed to be Markov.

**Markov games with function approximation.** Recently, a few works considered learning in Markov games with linear function approximation [36; 8] and general function approximation [19; 15; 38; 37; 7; 25]. While all of the works require centralized function classes and suffer from the curse of multi-agency, [9; 33] proposed decentralized MARL algorithms to resolve the issue under linear and general function approximation.

**Single-agent RL.** Broadly speaking, our work is also related to single-agent RL [1; 2; 10; 17; 40]. As a special case of Markov games, only one agent interacts with the environment in single-agent RL. For tabular episodic setting, the minimax-optimal sample complexity is $\widetilde{O}(H^3 SA/\epsilon^2)$, achieved by a model-based algorithm in [2] and a model-free algorithm in [40]. Technically, the reference-advantage decomposition used in our algorithm is similar to that of [40], as both employ variance reduction techniques for faster convergence. However, our approaches differ significantly, particularly in the way of handling the interplay between the CCE oracle and the reference-advantage decomposition in the context of two-player zero-sum Markov game.

## 2 PRELIMINARIES

We consider the tabular episodic two-player zero-sum Markov game $\mathrm{MG}(H, \mathcal{S}, \mathcal{A}, \mathcal{B}, P, r)$, where $H$ is the number of steps in each episode, $\mathcal{S}$ is the set of states with $|\mathcal{S}| = S$, $(\mathcal{A}, \mathcal{B})$ are the sets of actions of the max-player and the min-player respectively with $|\mathcal{A}| = A$ and $|\mathcal{B}| = B$, $P = \{P_h\}_{h \in [H]}$ is the collection of the transition matrices with $P_h : \mathcal{S} \times \mathcal{A} \times \mathcal{B} \mapsto \mathcal{S}$, $r = \{r_h\}_{h \in [H]}$ is the collection of deterministic reward functions with $r_h : \mathcal{S} \times \mathcal{A} \times \mathcal{B} \mapsto [0, 1]$. Here the reward represents both the gain of the max-player and the loss of the min-player. We assume each episode starts with a fixed initial state $s_1$.

Suppose the max-player and the min-player interact with the environment sequentially captured by the Markov game $\mathrm{MG}(H, \mathcal{S}, \mathcal{A}, \mathcal{B}, P, r)$. At each step $h \in [H]$, both players observe the state $s_h \in \mathcal{S}$, take their actions $a_h \in \mathcal{A}$ and $b_h \in \mathcal{B}$ simultaneously, receive the reward $r_h(s_h, a_h, b_h)$, and then the Markov game evolves into the next state $s_{h+1} \sim P_h(\cdot|s_h, a_h, b_h)$. The episode ends when $s_{H+1}$ is reached.

**Markov policy, value function.** A Markov policy $\mu$ of the max-player is the collection of the functions $\{\mu_h : \mathcal{S} \mapsto \Delta_{\mathcal{A}}\}_{h \in [H]}$, each of which maps from a state to a distribution over actions. Similarly, a policy $\nu$ of the min-player is the collection of functions $\{\nu_h : \mathcal{S} \mapsto \Delta_{\mathcal{B}}\}_{h \in [H]}$. We use $\mu_h(a|s)$ and $\nu_h(b|s)$ to denote the probability of taking actions $a$ and $b$ given the state $s$ under the Markov policies $\mu$ and $\nu$ at step $h$, respectively.

Given a max-player policy $\mu$, a min-player policy $\nu$, and a state $s$ at step $h$, the value function is defined as

$$V_h^{\mu,\nu}(s) = \mathbb{E}_{(s_{h'}, a_{h'}, b_{h'}) \sim (\mu,\nu)} \left[ \sum_{h'=h}^{H} r_{h'}(s_{h'}, a_{h'}, b_{h'}) \middle| s_h = s \right].$$

For a given $(s, a, b) \in \mathcal{S} \times \mathcal{A} \times \mathcal{B}$ under a max-player policy $\mu$ and a min-player policy $\nu$ at step $h$, we define

$$Q_h^{\mu,\nu}(s, a, b) = \mathbb{E}_{(s_{h'}, a_{h'}, b_{h'}) \sim (\mu,\nu)} \left[ \sum_{h'=h}^{H} r_{h'}(s_{h'}, a_{h'}, b_{h'}) \middle| s_h = s, a_h = a, b_h = b \right].$$

For ease of exposition, we define $(P_h f)(s, a, b) = \mathbb{E}_{s' \sim P_h(\cdot|s,a,b)}[f(s')]$ for any function $f : \mathcal{S} \mapsto \mathbb{R}$, and $(\mathbb{D}_\pi g)(s) = \mathbb{E}_{(a,b) \sim \pi(\cdot,\cdot|s)}[g(s, a, b)]$ for any function $g : \mathcal{S} \times \mathcal{A} \times \mathcal{B}$. Then, the following Bellman equations hold for all $(s, a, b, h) \in \mathcal{S} \times \mathcal{A} \times \mathcal{B} \times [H]$:

$$Q_h^{\mu,\nu}(s, a, b) = (r_h + P_h V_{h+1}^{\mu,\nu})(s, a, b), \quad V_h^{\mu,\nu}(s) = (\mathbb{D}_{\mu_h \times \nu_h} Q_h^{\mu,\nu})(s), \quad V_{H+1}^{\mu,\nu}(s) = 0.$$

**Best response, Nash equilibrium (NE).** For any Markov policy $\mu$ of the max-player, there exists a *best response* of the min-player, which is a policy $\nu^\dagger(\mu)$ satisfying $V_h^{\mu,\nu^\dagger(\mu)}(s) = \inf_\nu V_h^{\mu,\nu}$ for any $(s,h) \times \mathcal{S} \times [H]$. We denote $V_h^{\mu,\dagger} = V_h^{\mu,\nu^\dagger(\mu)}$. Similarly, the best response of the max-player with respect to the Markov policy $\nu$ of the min-player is a policy $\mu^\dagger(\nu)$ satisfying $V_h^{\mu^\dagger(\nu),\nu}(s) = \sup_\mu V_h^{\mu,\nu}$ for any $(s,h) \times \mathcal{S} \times [H]$, and we use $V_h^{\dagger,\nu}$ to denote $V_h^{\mu^\dagger(\nu),\nu}$. Further, there exists Markov policies $\mu^*, \nu^*$, which are optimal against the best responses of the other player [12], i.e.,

$$V_h^{\mu^*,\dagger}(s) = \sup_\mu V_h^{\mu,\dagger}(s), \quad V_h^{\dagger,\nu^*}(s) = \inf_\nu V_h^{\dagger,\nu},$$

for all $(s,h) \in \mathcal{S} \times [H]$. We call the strategies $(\mu^*, \nu^*)$ the *Nash equilibrium* of a Markov game, if they satisfy the following minimax equation

$$\sup_\mu \inf_\nu V_h^{\mu,\nu}(s) = V_h^{\mu^*,\nu^*}(s) = \inf_\nu \sup_\mu V_h^{\mu,\nu}(s).$$

**Learning objective.** We consider the Nash equilibrium of Markov games. We measure the sub-optimality of any pair of general policies $(\mu, \nu)$ using the following gap between their performance and the performance of the optimal strategy (i.e., Nash equilibrium) when playing against the best responses respectively:

$$V_1^{\dagger,\nu}(s_1) - V_1^{\mu,\dagger}(s_1) = \left(V_1^{\dagger,\nu}(s_1) - V_1^*(s_1)\right) + \left(V_1^*(s_1) - V_1^{\mu,\dagger}(s_1)\right).$$

**Definition 2.1** ($\epsilon$-optimal Nash equilibrium (NE))**.** A pair of general policies $(\mu, \nu)$ is an $\epsilon$-optimal Nash equilibrium if $V_1^{\dagger,\nu}(s_1) - V_1^{\mu,\dagger}(s_1) \leq \epsilon$.

Our goal is to design algorithms for two-player zero-sum Markov games that can find an $\epsilon$-optimal NE using a number episodes that is small in its dependency on $S, A, B, H$ as well as $1/\epsilon$.

## 3 ALGORITHM DESIGN

In this section, we propose an algorithm called Q-learning with min-gap based reference-advantage decomposition (Algorithm 1), for learning $\epsilon$-optimal NE in two-player zero-sum Markov games. Our algorithm builds upon the Nash Q-learning framework [5] but incorporates a novel **min-gap** based reference-advantage decomposition technique and stage-based update design. We start by reviewing the algorithm with reference-advantage decomposition in single agent RL [40].

**Reference-advantage decomposition in single-agent RL.** In single-agent RL, we greedily select and action to maximize the action value function $\overline{Q}_h(s,a)$ to obtain the optimistic value function $\overline{V}_h(s) = \max_a \overline{Q}_h(s,a)$, and the action-value function update follows $\overline{Q}_h(s,a) \leftarrow \min\{\overline{Q}_h^{(1)}(s,a), \overline{Q}_h^{(2)}(s,a), \overline{Q}_h(s,a)\}$, where $\overline{Q}_h^{(1)}, \overline{Q}_h^{(2)}$ represent the standard update rule and the advantage-based update rule

$$\overline{Q}_h^{(1)} \leftarrow r_h(s,a) + \widehat{P_h \overline{V}_{h+1}}(s,a) + \text{bonus}_1,$$

$$\overline{Q}_h^{(2)} \leftarrow r_h(s,a) + \widehat{P_h \overline{V}_{h+1}^{\text{ref}}}(s,a) + \widehat{P_h(\overline{V}_{h+1} - \overline{V}_{h+1}^{\text{ref}})}(s,a) + \text{bonus}_2.$$

In standard update rule, one major drawback is that the early samples collected for estimating $\overline{V}_{h+1}$ at that moment deviates from the true value of $\overline{V}_{h+1}$, and we have to only use the latest samples to estimate $\widehat{P_h \overline{V}_{h+1}}(s,a)$ in order not to ruin the whole estimate, which leads to the suboptimal sample complexity of such an algorithm. To achieve the optimal sample complexity, reference-advantage decomposition was introduced. At high level, we first learn an accurate estimation $\overline{V}_h^{\text{ref}}$ of the optimal value function $V_h^*$ satisfying $V_h^*(s) \leq V_h^{\text{ref}}(s) \leq V_h^*(s) + \beta$, where the accuracy is controlled by parameter $\beta$ independent of the number of episodes $K$. For the second term, since $\overline{V}_{h+1}^{\text{ref}}$ is almost fixed, we are able to conduct the estimate using all collected samples. For the third term, we still have to only use the latest samples to limit the deviation error. Thanks to the reference-advantage decomposition, and since $\overline{V}_{h+1}$ is learned based on $\overline{V}_{h+1}^{\text{ref}}$, and $\overline{V}_{h+1}^{\text{ref}}$ is already an accurate estimate

of $V_{h+1}^*$, it turns out that estimating $\overline{V}_{h+1} - \overline{V}_{h+1}^{\text{ref}}$ instead of directly estimating $\overline{V}_{h+1}$ offsets the weakness of using the latest samples.

In single-agent RL, one key design to facilitate the reference-advantage decomposition is to ensure that the action-value function $\overline{Q}_h(s,a)$ is non-increasing. Observe that the optimistic value function $\overline{V}_h(s)$ preserves the monotonic structure as long as the optimistic action-value function $\overline{Q}_h(s)$ is non-increasing, since $\overline{V}_h^{k+1}(s) = \max_a \overline{Q}_h^{k+1}(s,a) \leq \max_a \overline{Q}_h^k(s,a) = \overline{V}_h^k(s)$. When enough samples are collected, the reference value $\overline{V}^{\text{ref}}$ is then updated as the latest optimistic value function, which we remark is also the smallest optimistic value function in the up-to-date learning history.

**Min-gap**[1] **based reference-advantage decomposition.** In the two-player zero-sum game, we keep track of both the optimistic and the pessimistic action-value functions, and update the value functions using the CCE oracle at the end of each stage. Unlike the single-agent scenario, the optimistic (or pessimistic) value function does not necessarily preserve the monotone property even if the optimistic (or pessimistic) action-value function is non-increasing (or non-decreasing) due to the nature of the CCE oracle. In order to obtain the "best" optimistic and pessimistic value function pair, we come up with the key novel "**min-gap**" design where we update the reference value functions as the pair of optimistic and pessimistic value functions whose value difference is the smallest in the history (line 12-15). Formally, we define the min-gap $\Delta(s,h)$ for a state $s$ at step $h$ to keep track of the smallest value difference between optimistic and pessimistic value functions in the history, and the corresponding pair of value functions are recorded (line 12-13). When enough samples are collected (line 14-15), the pair of reference value functions is then set to be the pair of optimistic and pessimistic value functions whose value difference is the smallest in the history.

---

**Algorithm 1** Q-learning with min-gap based reference-advantage decomposition (Algorithm 3 sketch)

---

1: Set accumulators and (action)-value functions properly, and initialize the gap $\Delta(s,h) = H$.
2: **for** episodes $k \leftarrow 1, 2, \ldots, K$ **do**
3:     **for** $h \leftarrow 1, 2, \ldots, H$ **do**
4:         Take action $(a_h, b_h) \leftarrow \pi_h(s_h)$, receive $r_h(s_h, a_h, b_h)$, and observe $s_{h+1}$.
5:         Update accumulators.
6:         **if** $n \in \mathcal{L}$ **then**
7:             $\overline{Q}_h(s_h, a_h, b_h) \leftarrow \min\{\overline{Q}_h^{(1)}(s_h, a_h, b_h), \overline{Q}_h^{(2)}(s_h, a_h, b_h), \overline{Q}_h(s_h, a_h, b_h)\}$.
8:             $\underline{Q}_h(s_h, a_h, b_h) \leftarrow \max\{\underline{Q}_h^{(1)}(s_h, a_h, b_h), \underline{Q}_h^{(2)}(s_h, a_h, b_h), \underline{Q}_h(s_h, a_h, b_h)\}$.
9:             $\pi_h(s_h) \leftarrow \text{CCE}(\overline{Q}(s_h, \cdot, \cdot), \underline{Q}_h(s_h, \cdot, \cdot))$.
10:            $\overline{V}_h(s_h) \leftarrow \mathbb{E}_{(a,b) \sim \pi_h(s_h)} \overline{Q}_h(s_h, a, b)$, and $\underline{V}_h(s_h) \leftarrow \mathbb{E}_{(a,b) \sim \pi_h(s_h)} \underline{Q}_h(s_h, a, b)$.
11:            Reset all intra-stage accumulators to 0.
12:            **if** $\overline{V}_h(s_h) - \underline{V}_h(s_h) < \Delta(s, h)$ **then**
13:                $\Delta(s,h) = \overline{V}_h(s_h) - \underline{V}_h(s_h)$, $\widetilde{\overline{V}}_h(s_h) = \overline{V}_h(s_h)$, and $\widetilde{\underline{V}}_h(s_h) = \underline{V}_h(s_h)$.
14:         **if** $\sum_{a,b} N_h(s_h, a, b) = N_0$ **then**
15:            $\overline{V}_h^{\text{ref}}(s_h) \leftarrow \widetilde{\overline{V}}_h(s_h)$, $\underline{V}_h^{\text{ref}}(s_h) \leftarrow \widetilde{\underline{V}}_h(s_h)$.

---

Now we introduce reference-advantage decomposition to the two-player zero-sum game. For ease of exposition, we use $\text{bonus}_i$ to represent different exploration bonus, which is specified in line 9-11 of Algorithm 3. In standard update rule, we have

$$\overline{Q}_h^{(1)}(s,a,b) \leftarrow r_h(s,a,b) + \widehat{P_h \overline{V}_{h+1}}(s,a,b) + \text{bonus}_3, \tag{1}$$

$$\underline{Q}_h^{(1)}(s,a,b) \leftarrow r_h(s,a,b) + \widehat{P_h \underline{V}_{h+1}}(s,a,b) + \text{bonus}_4, \tag{2}$$

where $\widehat{P_h \overline{V}_{h+1}}, \widehat{P_h \underline{V}_{h+1}}$ are the empirical estimate of $P_h \overline{V}_{h+1}, P_h \underline{V}_{h+1}$. Similar to the single-agent RL, the standard update rule suffers from the large deviation between $\overline{V}_{h+1}$ learned by the early samples and the value of Nash equilibrium. As a result, we have to use only the samples from the last stage (i.e., the latest $O(1/H)$ fraction of samples, see stage-based update approach below) to estimate $P_h \overline{V}_{h+1}$. In order to improve the horizon dependence, we incorporate the advantage-based

---

[1]We remark that min-gap has nothing to do with the notion of gap in gap-dependent RL.

update rule

$$\overline{Q}_h^{(2)}(s,a,b) \leftarrow r_h(s,a,b) + \widehat{P_h \overline{V}_{h+1}^{\mathrm{ref}}}(s,a,b) + \widehat{P_h(\overline{V}_{h+1} - \overline{V}_{h+1}^{\mathrm{ref}})}(s,a,b) + \mathrm{bonus}_5, \quad (3)$$

$$\underline{Q}_h^{(2)}(s,a,b) \leftarrow r_h(s,a,b) + \widehat{P_h \underline{V}_{h+1}^{\mathrm{ref}}}(s,a,b) + \widehat{P_h(\underline{V}_{h+1} - \underline{V}_{h+1}^{\mathrm{ref}})}(s,a,b) + \mathrm{bonus}_6, \quad (4)$$

where the middle terms in (3) are the empirical estimates of $P_h \overline{V}_{h+1}^{\mathrm{ref}}$ and $P_h(\overline{V}_{h+1} - \overline{V}_{h+1}^{\mathrm{ref}})$, and the middle terms in (4) are the empirical estimates of $P_h \underline{V}_{h+1}^{\mathrm{ref}}$ and $P_h(\underline{V}_{h+1} - \overline{V}_{h+1}^{\mathrm{ref}})$. We still need to use only the samples from the last stage to limit the deviation for the third terms in both (3) and (4). For ease of exposition, assume we have access to a $\beta$-optimal $\overline{V}^{\mathrm{ref}}, \underline{V}^{\mathrm{ref}}$. Thanks to the min-gap based reference-advantage decomposition, the learned $\overline{V}_{h+1}$ (or $\underline{V}_{h+1}$) is learned based on $\overline{V}_{h+1}^{\mathrm{ref}}$ (or $\underline{V}_{h+1}^{\mathrm{ref}}$), and $\overline{V}^{\mathrm{ref}}$ (or $\underline{V}^{\mathrm{ref}}$) is already an accurate estimate of $V_{h+1}^*$, it turns out that estimating $\overline{V}_{h+1} - \overline{V}_{h+1}^{\mathrm{ref}}$ (or $\underline{V}_{h+1} - \underline{V}_{h+1}^{\mathrm{ref}}$) instead of directly estimating $\overline{V}$ (or $\underline{V}$) offsets the weakness of using only $O(1/H)$ fraction of data. Further, since $\overline{V}^{\mathrm{ref}}, \underline{V}^{\mathrm{ref}}$ is fixed, we are able to use all samples collected to estimate the second term, without suffering any deviation. Now we remove the assumption that $\overline{V}^{\mathrm{ref}}, \underline{V}^{\mathrm{ref}}$ is fixed. Note that $\beta$ is selected independently of $K$. Therefore, learning a $\beta$-optimal reference value function $\overline{V}^{\mathrm{ref}}, \underline{V}^{\mathrm{ref}}$ only incurs lower order terms in our final result.

**Stage-based update approach.** For each tuple $(s,a,b,h) \in \mathcal{S} \times \mathcal{A} \times \mathcal{B} \times [H]$, we divide the visitations for the tuple into consecutive stages. The length of each stage increases exponentially with a growth rate $(1 + 1/H)$. Specifically, we define $e_1 = H$, and $e_{i+1} = \lfloor (1 + 1/H)e_i \rfloor$ for all $i \geq 1$, to denote the lengths of stages. Further, we also define $\mathcal{L} = \{\sum_{i=1}^j e_i | j = 1, 2, 3, \ldots\}$ to denote the the set of ending indices of the stages. For each $(s,a,b,h)$ tuple, we update both the optimistic and pessimistic value estimates at the end of each stage (i.e., when the total number of visitations of $(s,a,b,h)$ lies in $\mathcal{L}$), using samples only from this single stage (line 6-15). This updating rule ensures that only the last $O(1/H)$ fraction of the collected samples are used to estimate the value estimates.

**Coarse correlated equilibrium (CCE).** We use the CCE oracle to update the policy (line 14). The CCE oracle was first introduced in [36] and an $\epsilon$-optimal CCE is shown to be a $O(\epsilon)$-optimal Nash equilibrium in two-player zero-sum Markov games [36]. For any pair of matrices $\overline{Q}, \underline{Q} \in [0, H]^{A \times B}$, $\mathrm{CCE}(\overline{Q}, \underline{Q})$ returns a distribution $\pi \in \Delta_{\mathcal{A} \times \mathcal{B}}$ such that

$$\mathbb{E}_{(a,b) \sim \pi} \overline{Q}(a,b) \geq \sup_{a^*} \mathbb{E}_{(a,b) \sim \pi} \overline{Q}(a^*,b), \quad \mathbb{E}_{(a,b) \sim \pi} \underline{Q}(a,b) \leq \inf_{b^*} \mathbb{E}_{(a,b) \sim \pi} \overline{Q}(a,b^*).$$

The players choose their actions in a potentially correlated way so that no one can benefit from unilateral unconditional deviation. Since Nash equilibrium is also a CCE and a Nash equilibrium always exists, a CCE therefore always exist. Moreover, CCE can be efficiently implemented by linear programming in polynomial time. We remark that the policies generated by CCE are in general correlated, and executing such policies requires the cooperation of the two players (line 6).

**Algorithm description.** For clarity, we provide a schematic algorithm here (Algorithm 1) and defer the detail to the appendix (Algorithm 3). Besides the standard optimistic and pessimistic value estimates $\overline{Q}_h(s,a,b), \overline{V}_h(s), \underline{Q}_h(s,a,b), \underline{V}_h(s)$, and the reference value functions $\overline{V}_h^{\mathrm{ref}}(s)$, $\underline{V}_h^{\mathrm{ref}}(s)$, the algorithm keeps multiple different accumulators to facilitate the update: 1) $N_h(s,a,b)$ and $\check{N}_h(s,a,b)$ are used to keep the total visit number and the visits counting for the current stage with respect to $(s,a,b,h)$, respectively. 2) Intra-stage accumulators are used in the latest stage and are reset at the beginning of each stage. 3) The global accumulators are used for the samples in all stages: All accumulators are initialized to 0 at the beginning of the algorithm. The details of the accumulators are deferred to Appendix A.

The algorithm set $\iota = \log(2/\delta)$, $\beta = O(1/H)$ and $N_0 = c_4 SABH^5\iota/\beta^2$ for some sufficiently large universal constant $c_4$, denoting the number of visits required to learn $\beta$-accurate pair of reference value functions.

**Certified policy.** Based on the policy trajectories collected from Algorithm 3, we construct an output policy profile $(\mu^{\mathrm{out}}, \nu^{\mathrm{out}})$ that we will show is an approximate NE. For any step $h \in [H]$, an episode $k \in [K]$ and any state, we let $\mu_h^k(\cdot|s) \in \Delta(\mathcal{A})$ and $\nu_h^k(\cdot|s) \in \Delta(\mathcal{B})$ be the distribution prescribed by

Algorithm 3 at this step. Let $\check{N}_h^k(s)$ be the value $\check{N}_h^k(s)$ at the beginning of the $k$-th episode. Our construction of the output policy $\mu^{\mathrm{out}}$ is presented in Algorithm 2 (whereas the certified policy $\nu^{\mathrm{out}}$ of the min-player can be obtained similarly), which follows the "certified policies" introduced in [3]. We remark that the episode index from the previous stage is uniformly sampled in our algorithm while the certified policies in [3] uses a weighted mixture.

---

**Algorithm 2** Certified policy $\mu^{\mathrm{out}}$ (max-player version)

---

1: Sample $k \leftarrow \mathrm{Unif}([K])$.
2: **for** step $h \leftarrow 1, \ldots, H$ **do**
3:      Receive $s_h$, and take action $a_h \sim \mu_h^k(\cdot|s_h)$.
4:      Observe $b_h$, and sample $j \leftarrow \mathrm{Unif}([N_h^k(s_h, a_h, b_h)])$.
5:      Set $k \leftarrow \check{\ell}_{h,j}^k$.

---

## 4 THEORETICAL ANALYSIS

### 4.1 MAIN RESULT

In this subsection, we present the main theoretical result for Algorithm 3. The following theorem presents the sample complexity guarantee for Algorithm 3 to learn a near-optimal Nash equilibrium policy in two-player zero-sum Markov games, which improves the best-known model-free algorithms in the same setting.

**Theorem 4.1.** *For any $\delta \in (0,1)$, let the agents run Algorithm 3 for $K$ episodes with $K \geq \widetilde{O}(H^3 SAB/\epsilon^2)$. Then, with probability at least $1 - \delta$, the output policy $(\mu^{\mathrm{out}}, \nu^{\mathrm{out}})$ of Algorithm 2 is an $\epsilon$-approximate Nash equilibrium.*

Compared to the lower bound $\Omega(H^3 S(A + B)/\epsilon^2)$ on the sample complexity to find a near-optimal Nash equilibrium [4], the sample complexity in Theorem 4.1 is minimax-optimal on the dependence of $H$, $S$ and $\epsilon$. This is the first result that establishes such optimality for model-free algorithms, although model-based algorithms have been shown to achieve such optimality in the past [22].

We also note that the result in Theorem 4.1 is not tight on the dependence on the cardinality of actions $A, B$. Such a gap has been closed by popular V-learning algorithms [22; 24], which achieve the sample complexity of $O(H^5 S(A + B)/\epsilon^2)$ [24]. Clearly, V-learning achieves a tight dependence on $A, B$, but suffers from worse horizon dependence on $H$. More specifically, one $H$ factor is due to the nature of implementing the adversarial bandit subroutine in exchange for a better action dependence $A + B$. The other $H$ factor could potentially be improved via the reference-advantage decomposition technique that we adopt here for our Q-learning algorithm. We leave this promising yet challenging direction as a future study.

### 4.2 PROOF OUTLINE

In this section, we present the proof sketch of Theorem 4.1, and defer all the details to the appendix.

Our main technical development lies in establishing a few new properties on the cumulative occurrence of the large V-gap and the cumulative bonus term, which enable the upper-bounding of several new error terms arising due to the incorporation of the new min-gap based reference-advantage decomposition technique. These properties have not been established for the single-agent RL with such a technique, because our properties are established for policies generated by the CCE oracle in zero-sum Markov games. Further, we perform a more refined analysis for both the optimistic and pessimistic accumulative bonus terms in order to obtain the desired result.

For certain functions, we use the superscript $k$ to denote the value of the function at the beginning of the $k$-th episode, and use the superscript $K + 1$ to denote the value of the function after all $K$ episodes are played. For instance, we denote $N_h^k(s, a, b)$ as the value of $N_h(s, a, b)$ at the beginning of the $k$-th episode, and $N_h^{K+1}(s, a, b)$ to denote the total number of visits of $(s, a, b)$ at step $h$ after $K$ episodes are played. When $h$ and $k$ are clear from the context, we omit the subscript $h$ and superscript $k$

for notational convenience. For example, we use $\ell_i$ and $\check{\ell}_i$ to denote $\ell_{h,i}^k$ and $\check{\ell}_{h,i}^k$ when $h$ and $k$ are obvious.

**Preliminary step.** We build connection between the certified policy generated by Algorithm 2, and the difference between the optimistic and pessimistic value functions.

**Lemma 4.2.** *Let* $(\mu^{\mathrm{out}}, \nu^{\mathrm{out}})$ *be the output policy induced by the certified policy algorithm (Algorithm 2), then, we have* $V_1^{\dagger, \nu^{\mathrm{out}}}(s_1) - V_1^{\mu^{\mathrm{out}}, \dagger}(s_1) \leq \frac{1}{K} \sum_{k=1}^K (\overline{V}_1^k - \underline{V}_1^k)(s_1)$.

In the remaining steps, we aim to bound $\sum_{k=1}^K (\overline{V}_1^k - \underline{V}_1^k)(s_1)$.

**Step I:** We show that the Nash equilibrium (action-)value functions are always bounded between the optimistic and pessimistic (action-)value functions.

**Lemma 4.3.** *With high probability, it holds that for any* $s, a, b, k, h$,

$$\underline{Q}_h^k(s, a, b) \leq Q_h^*(s, a, b) \leq \overline{Q}_h^k(s, a, b), \qquad \underline{V}_h^k(s) \leq V_h^*(s) \leq \overline{V}_h^k(s).$$

Our new technical development lies in proving the inequality with respect to the action-value function, whose update rule features the min-gap reference-advantage decomposition.

The proof is by induction. We will focus on the optimistic (action-)value function and the other direction can be proved similarly. Suppose the two inequalities hold in episode $k$. We first establish the inequality for action-value function, and then prove the inequality for value functions. Based on the update rule of the optimistic action-value functions (line 12 in Algorithm 3), the action-value function is determined by the first two non-trial terms and last trivial term. While the first term is shown to upper bound the action-value function at Nash equilibrium $Q_h^*(s, a, b)$, we make the effort to showcase that the second term involving the min-gap based reference-advantage decomposition also upper bounds $Q_h^*(s, a, b)$. Since the optimistic action-value function takes the minimum of the three terms, we conclude that the optimistic action-value function in episode $k + 1$ satisfy the inequality. The proof of the inequality for value function (second inequality in Lemma 4.3) is based on the property of the policy distribution output by the CCE oracle.

Note that the optimistic (or pessimistic) action-value function is non-increasing (or non-decreasing). However, the optimistic and the pessimistic value functions do not preserve such monotonic property due to the nature of the CCE oracle. This motivates our design of the min-gap based reference-advantage decomposition.

**Step II:** We show that the reference value can be learned with bounded sample complexity in the following lemma.

**Lemma 4.4.** *With high probability, it holds that* $\sum_{k=1}^K \mathbf{1}\{\overline{V}_h^k(s_h^k) - \underline{V}_h^k(s_h^k) \geq \epsilon\} \leq O(SABH^5 \iota / \epsilon^2)$.

We show that in the two-player zero-sum Markov game, the occurrence of the large V-gap, induced by the policy generated by the CCE oracle, is bounded independent of the number of episodes $K$. Our new development in proving this lemma lies in handling an additional martingale difference arising due to the CCE oracle.

In order to extract the best pair of optimistic and pessimistic value functions, a key novel min-gap based reference-advantage decomposition is proposed (see Section 3), based on which we pick up the pair of optimistic and pessimistic value functions whose gap is the smallest in the history (line 17-20 in Algorithm 3). By the selection of the reference value functions, Lemma 4.4 with $\epsilon$ set to $\beta$, and the definition of $N_0$, we have the following corollary.

**Corollary 4.5.** *Conditioned on the successful events of Proposition 4.3 and Lemma 4.4, for every state* $s$, *we have* $n_h^k(s) \geq N_0 \implies \overline{V}_h^{\mathrm{ref},k}(s) - \underline{V}_h^{\mathrm{ref},k}(s) \leq \beta$.

**Step III:** We bound $\sum_{k=1}^K (\overline{V}_1^k - \underline{V}_1^k)(s_1)$. Compared to single-agent RL, the CCE oracle leads to a possibly mixed policy and we need to bound the additional term due to the CCE oracle.

For ease of exposition, define $\Delta_h^k = (\overline{V}_h^k - \underline{V}_h^k)(s_h^k)$, and martingale difference $\zeta_h^k = \Delta_h^k - (\overline{Q}_h^k - \underline{Q}_h^k)(s_h^k, a_h^k, b_h^k)$. Note that $n_h^k = N_h^k(s_h^k, a_h^k, b_h^k)$ and $\check{n}_h^k = \check{N}_h^k(s_h^k, a_h^k, b_h^k)$ when $N_h^k(s_h^k, a_h^k, b_h^k) \in$

$\mathcal{L}$. Following the update rule, we have (omitting the detail)

$$\Delta_h^k = \zeta_h^k + (\overline{Q}_h^k - \underline{Q}_h^k)(s_h^k, a_h^k, b_h^k) \leq \zeta_h^k + H\mathbf{1}\{n_h^k = 0\} + \frac{1}{\check{n}_h^k}\sum_{i=1}^{\check{n}_h^k}\Delta_{h+1}^{\check{\ell}_i} + \Lambda_{h+1}^k,$$

where the definition of $\Lambda_{h+1}^k$ is provided in the appendix.

Summing over $k \in [K]$, we have

$$\sum_{k=1}^K \Delta_h^k \leq \sum_{k=1}^K \zeta_h^k + \sum_{k=1}^K H\mathbf{1}\{n_h^k = 0\} + \sum_{k=1}^K \frac{1}{\check{n}_h^k}\sum_{i=1}^{\check{n}_h^k}\Delta_{h+1}^{\check{\ell}_{h,i}^k} + \sum_{k=1}^K \Lambda_{h+1}^k$$

$$\leq \sum_{k=1}^K \zeta_h^k + SABH^2 + (1+\frac{1}{H})\sum_{k=1}^K \Delta_{h+1}^k + \sum_{k=1}^K \Lambda_{h+1}^k,$$

where in the last inequality, we use the pigeon-hole argument for the second term, and the third term is due to the $(1+1/H)$ growth rate of the length of the stages.

Iterating over $h = H, H-1, \ldots, 1$ gives

$$\sum_{k=1}^K \Delta_1^k \leq \mathcal{O}\left(SABH^3 + \sum_{h=1}^H\sum_{k=1}^K(1+\frac{1}{H})^{h-1}\zeta_h^k + \sum_{h=1}^H\sum_{k=1}^K(1+\frac{1}{H})^{h-1}\Lambda_{h+1}^k\right).$$

The additional term $\sum_{h=1}^H\sum_{k=1}^K(1+\frac{1}{H})^{h-1}\zeta_h^k$ is new in the two-player zero-sum Markov game, which can be bounded by Azuma-Hoeffding's inequality. I.e., it holds that with probability at least $1 - T\delta$,

$$\sum_{h=1}^H\sum_{k=1}^K(1+\frac{1}{H})^{h-1}\zeta_h^k \leq \mathcal{O}(\sqrt{H^2 T\iota}),$$

which turns out to be a lower-order term compared to $\sum_{h=1}^H\sum_{k=1}^K(1+\frac{1}{H})^{h-1}\Lambda_{h+1}^k$.

**Step IV:** We bound $\sum_{h=1}^H\sum_{k=1}^K(1+\frac{1}{H})^{h-1}\Lambda_{h+1}^k$ in the following lemma.

**Lemma 4.6.** *With high probability, it holds that*

$$\sum_{h=1}^H\sum_{k=1}^K(1+\frac{1}{H})^{h-1}\Lambda_{h+1}^k = O\left(\sqrt{SABH^2\iota} + H\sqrt{T\iota}\log T + S^2(AB)^{\frac{3}{2}}H^8\iota T^{\frac{1}{4}}\right).$$

We capture the accumulative error of the bonus terms $\sum_{h=1}^H\sum_{k=1}^K(1+\frac{1}{H})^{h-1}(\overline{\beta}_{h+1}^k + \underline{\beta}_{h+1}^k)$ in the expression $\sum_{h=1}^H\sum_{k=1}^K(1+\frac{1}{H})^{h-1}\Lambda_{h+1}^k$. Since we first implement the reference-advantage decomposition technique in the two-player zero-sum game, our accumulative bonus term is much more challenging to analyze than the existing Q-learning algorithms for games. Compared to the analysis for the model-free algorithm with reference-advantage decomposition in single-RL [40], our analysis features the following **new developments**. First, we need to bound both the optimistic and pessimistic accumulative bonus terms, and the analysis is not identical. Second, the analysis of the optimistic accumulative bonus term differs due to the CCE oracle and the new min-gap base reference-advantage decomposition for two-player zero-sum Markov game.

Finally, combining all steps, we conclude that with high probability,

$$V_1^{\dagger,\nu^{\text{out}}}(s_1) - V_1^{\mu^{\text{out}},\dagger}(s_1) \leq \frac{1}{K}\sum_{k=1}^K \Delta_h^k = O\left(\frac{H^3 SAB}{\epsilon^2}\right).$$

## 5 CONCLUSION

In this paper, we proposed a new model-free algorithm Q-learning with min-gap based reference-advantage decomposition for two-player zero-sum Markov games, which improved the existing results and achieved a near-optimal sample complexity $O(H^3 SAB/\epsilon^2)$ except for the $AB$ term. Due to the nature of the CCE oracle employed in the algorithm, we designed a novel min-gap based reference-advantage decomposition to learn the pair of optimistic and pessimistic reference value functions whose value difference has the minimum gap in the history. An interesting future direction would be to study whether the horizon dependence could be further tightened in model-free V-learning.

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
