# Supplementary Materials

## A  DETAILS OF ALGORITHM 1

---

**Algorithm 3** Q-learning with min-gap based reference-advantage decomposition

---

1: **Initialize:** Set all accumulators to 0. For all $(s, a, b, h) \in \mathcal{S} \times \mathcal{A} \times \mathcal{B} \times [H]$, set $\overline{V}_h(s), \overline{Q}_h(s, a, b)$
    to $H - h + 1$, set $\overline{V}_h^{\mathrm{ref}}(s)$ to $H$, set $\underline{V}_h(s), \underline{Q}_h(s, a, b), \underline{V}_h^{\mathrm{ref}}(s, a, b)$ to 0; and

2: let $\pi_h(s) \sim \mathrm{Unif}(\mathcal{A}) \times \mathrm{Unif}(\mathcal{B}), \Delta(s, h) = H, \widetilde{\overline{V}}_h(s_h) = H, \widetilde{\underline{V}}_h(s_h) = 0.$
3: **for** episodes $k \leftarrow 1, 2, \dots, K$ **do**
4:    Observe $s_1$.
5:    **for** $h \leftarrow 1, 2, \dots, H$ **do**
6:        Take action $(a_h, b_h) \leftarrow \pi_h(s_h)$, receive $r_h(s_h, a_h, b_h)$, and observe $s_{h+1}$.
7:        Update accumulators $n := N_h(s_h, a_h, b_h) \overset{+}{\leftarrow} 1, \check{n} := \check{N}_h(s_h, a_h, b_h) \overset{+}{\leftarrow} 1$ and (5)-(9).
8:        **if** $n \in \mathcal{L}$ **then**
9:            $\gamma \leftarrow 2\sqrt{\frac{H^2}{\check{n}}\iota}.$
10:            $\overline{\beta} \leftarrow c_1\sqrt{\frac{\overline{\sigma}^{\mathrm{ref}}/n - (\overline{\mu}^{\mathrm{ref}}/n)^2}{n}\iota} + c_2\sqrt{\frac{\check{\overline{\sigma}}/\check{n} - (\check{\overline{\mu}}/\check{n})^2}{\check{n}}\iota} + c_3(\frac{H\iota}{n} + \frac{H\iota}{\check{n}} + \frac{H\iota^{3/4}}{n^{3/4}} + \frac{H\iota^{3/4}}{\check{n}^{3/4}}).$
11:            $\underline{\beta} \leftarrow c_1\sqrt{\frac{\underline{\sigma}^{\mathrm{ref}}/n - (\underline{\mu}^{\mathrm{ref}}/n)^2}{n}\iota} + c_2\sqrt{\frac{\check{\underline{\sigma}}/\check{n} - (\check{\underline{\mu}}/\check{n})^2}{\check{n}}\iota} + c_3(\frac{H\iota}{n} + \frac{H\iota}{\check{n}} + \frac{H\iota^{3/4}}{n^{3/4}} + \frac{H\iota^{3/4}}{\check{n}^{3/4}}).$
12:            $\overline{Q}_h(s_h, a_h, b_h) \leftarrow \min\{r_h(s_h, a_h, b_h) + \frac{\check{\overline{v}}}{\check{n}} + \gamma, r_h(s_h, a_h, b_h) + \frac{\overline{\mu}^{\mathrm{ref}}}{n} + \frac{\check{\overline{\mu}}}{\check{n}} + \overline{\beta}, \overline{Q}_h(s_h, a_h, b_h)\}.$

13:            $\underline{Q}_h(s_h, a_h, b_h) \leftarrow \max\{r_h(s_h, a_h, b_h) + \frac{\check{\underline{v}}}{\check{n}} - \gamma, r_h(s_h, a_h, b_h)) + \frac{\underline{\mu}^{\mathrm{ref}}}{n} + \frac{\check{\underline{\mu}}}{\check{n}} - \underline{\beta}, \underline{Q}_h(s_h, a_h, b_h)\}.$

14:            $\pi_h(s_h) \leftarrow \mathrm{CCE}(\overline{Q}(s_h, \cdot, \cdot), \underline{Q}_h(s_h, \cdot, \cdot)).$
15:            $\overline{V}_h(s_h) \leftarrow \mathbb{E}_{(a,b) \sim \pi_h(s_h)} \overline{Q}_h(s_h, a, b)$, and $\underline{V}_h(s_h) \leftarrow \mathbb{E}_{(a,b) \sim \pi_h(s_h)} \underline{Q}_h(s_h, a, b).$
16:            Reset all intra-stage accumulators to 0.
17:            **if** $\overline{V}_h(s_h) - \underline{V}_h(s_h) < \Delta(s, h)$ **then**
18:                $\Delta(s, h) = \overline{V}_h(s_h) - \underline{V}_h(s_h).$
19:                $\widetilde{\overline{V}}_h(s_h) = \overline{V}_h(s_h), \widetilde{\underline{V}}_h(s_h) = \underline{V}_h(s_h).$
20:        **if** $\sum_{a,b} N_h(s_h, a, b) = N_0$ **then**
21:            $\overline{V}_h^{\mathrm{ref}}(s_h) \leftarrow \widetilde{\overline{V}}_h(s_h), \underline{V}_h^{\mathrm{ref}}(s_h) \leftarrow \widetilde{\underline{V}}_h(s_h).$

---

**Algorithm description.** Let $c_1, c_2, c_3$ be some sufficiently large universal constants so that the concentration inequalities can be applied in the analysis. Besides the standard optimistic and pessimistic value estimates $\overline{Q}_h(s, a, b), \overline{V}_h(s), \underline{Q}_h(s, a, b), \underline{V}_h(s)$, and the reference value functions $\overline{V}_h^{\mathrm{ref}}(s), \underline{V}_h^{\mathrm{ref}}(s)$, the algorithm keeps multiple different accumulators to facilitate the update: 1) $N_h(s, a, b)$ and $\check{N}_h(s, a, b)$ are used to keep the total visit number and the visits counting for the current stage with respect to $(s, a, b, h)$, respectively. 2) Intra-stage accumulators are used in the latest stage and are reset at the beginning of each stage. The update rule of the intra-stage accumulators are as follows:

$$\check{\overline{v}}_h(s_h, a_h, b_h) \overset{+}{\leftarrow} \overline{V}_{h+1}(s_{h+1}), \quad \check{\underline{v}}_h(s_h, a_h, b_h) \overset{+}{\leftarrow} \underline{V}_{h+1}(s_{h+1}), \tag{5}$$

$$\check{\overline{\mu}}_h(s_h, a_h, b_h) \overset{+}{\leftarrow} \overline{V}_{h+1}(s_{h+1}) - \overline{V}_{h+1}^{\mathrm{ref}}(s_{h+1}), \quad \check{\underline{\mu}}_h(s_h, a_h, b_h) \overset{+}{\leftarrow} \underline{V}_{h+1}(s_{h+1}) - \underline{V}_{h+1}^{\mathrm{ref}}(s_{h+1}), \tag{6}$$

$$\check{\overline{\sigma}}_h(s_h, a_h, b_h) \overset{+}{\leftarrow} (\overline{V}_{h+1}(s_{h+1}) - \overline{V}_{h+1}^{\mathrm{ref}}(s_{h+1}))^2, \check{\underline{\sigma}}_h(s_h, a_h, b_h) \overset{+}{\leftarrow} (\underline{V}_{h+1}(s_{h+1}) - \underline{V}_{h+1}^{\mathrm{ref}}(s_{h+1}))^2. \tag{7}$$

3) The following global accumulators are used for the samples in all stages:

$$\overline{\mu}_h^{\mathrm{ref}}(s_h, a_h, b_h) \overset{+}{\leftarrow} \overline{V}_{h+1}^{\mathrm{ref}}(s_{h+1}), \quad \underline{\mu}_h^{\mathrm{ref}}(s_h, a_h, b_h) \overset{+}{\leftarrow} \underline{V}_{h+1}^{\mathrm{ref}}(s_{h+1}), \tag{8}$$

$$\overline{\sigma}_h^{\mathrm{ref}}(s_h, a_h, b_h) \overset{+}{\leftarrow} (\overline{V}_{h+1}^{\mathrm{ref}}(s_{h+1}))^2, \quad \underline{\sigma}_h^{\mathrm{ref}}(s_h, a_h, b_h) \overset{+}{\leftarrow} (\underline{V}_{h+1}^{\mathrm{ref}}(s_{h+1}))^2. \tag{9}$$

All accumulators are initialized to 0 at the beginning of the algorithm. The algorithm set $\iota = \log(2/\delta)$, $\beta = O(1/H)$ and $N_0 = c_4 SABH^5/\beta^2$ for some sufficiently large universal constant $c_4$.

## B    Comparison to Existing Algorithms

**Compare to Optimistic Nash Q-learning [5].** The Optimistic Nash Q-learning is a model-free Q-learning algorithm for two-player zero-sum Markov games. The algorithm design differences between our algorithm and the optimistic Nash Q-learning is two-fold. First, we adopt the stage-based design instead of traditional Q-learning update $Q_{new} \leftarrow (1-\alpha)Q_{old}+\alpha(r+V)$. The optimistic Nash Q-learning updates the value function with a learning rate, while our algorithm adopts greedy update. We remark that both frameworks are viable, and in our opinion, the stage-based design is easier to follow and analyse. Second, we propose a novel min-gap based reference-advantage decomposition, a variance reduction technique, to further improve the sample complexity. Specifically, we use both the standard update rule and the advantage-based update rule in our action-value function (Q function) while the optimistic Nash Q-learning only uses the standard update rule.

Aside from the obvious distinction of the proofs caused by stage-based design, the main difference is the analysis for the advantage-based update rule, which does not show up in the optimistic Nash Q-learning. Due to the incorporation of the new min-gap based reference-advantage decomposition technique, several new error terms arise in our analysis. Our main development lies in establishing a few new properties on the cumulative occurrence of the large V-gap and the cumulative bonus term, which enable the upper-bounding of those new error terms. More specifically, as we explain in our proof outline in Section 4.2, our analysis include the following novel developments. (i) Step I shows that the Nash equilibrium (action-)value functions are always bounded between the optimistic and pessimistic (action-)value functions (see Lemma 4.3). Our new technical development here lies in proving the inequality with respect to the action-value function, whose update rule features the min-gap reference-advantage decomposition. (ii) Step II shows that the reference value can be learned with bounded sample complexity (see Lemma 4.4). Our new development here lies in handling an additional martingale difference arising due to the CCE oracle. (iii) In step IV, there are a few new developments. First, we need to bound both the optimistic and pessimistic accumulative bonus terms, and the analysis is more refined compared to that for single-agent RL. Second, the analysis of the optimistic accumulative bonus term need to handle the CCE oracle together with the new min-gap base reference-advantage decomposition for two-player zero-sum Markov game.

**Compare to UCB-advantage [40].** The UCB-advantage is a model-free algorithm with reference-advantage decomposition for single-agent RL. Our main novel design idea lies in the **min-gap** based advantage reference value decomposition. Unlike the single-agent scenario, the optimistic (or pessimistic) value function in Markov games does not necessarily preserve the monotone property due to the nature of the CCE oracle. In order to obtain the "best" optimistic and pessimistic value function pair, we propose the key **min-gap** design to update the reference value functions as the pair of optimistic and pessimistic value functions whose value difference is the smallest (i.e., with the minimal gap) in the history. It turns out that such a design is critical to guarantee the provable sample efficiency.

For the proof techniques, there are the fundamental differences between single-agent RL and two-player zero-sum games. Thanks to the key min-gap based reference-advantage decomposition, we provide a new guarantee for the learned pair of reference value (Corollary 4.5) in the context of two-player zero-sum Markov games, which is crucial in obtaining an optimal horizon dependence.

## C    Notations

For any function $f : \mathcal{S} \mapsto \mathbb{R}$, we use $P_{s,a,b}f$ and $(P_h f)(s, a, b)$ interchangeably. Define $\mathbb{V}(x, y) = x^\top(y^2) - (x^\top y)^2$ for two vectors of the same dimension, where $y^2$ is obtained by squaring each entry of $y$.

For ease of exposition, we define $\overline{\nu}_h^{\text{ref},k} = \frac{\overline{\sigma}_h^{\text{ref},k}}{n_h^k} - (\frac{\overline{\mu}_h^{\text{ref},k}}{n_h^k})^2$, $\underline{\nu}_h^{\text{ref},k} = \frac{\underline{\sigma}_h^{\text{ref},k}}{n_h^k} - (\frac{\underline{\mu}_h^{\text{ref},k}}{n_h^k})^2$ and $\breve{\nu}_h^k = \frac{\breve{\sigma}_h^k}{\breve{n}_h^k} - (\frac{\breve{\mu}_h^k}{\breve{n}_h^k})^2$, $\underline{\breve{\nu}}_h^k = \frac{\breve{\sigma}_h^k}{\breve{n}_h^k} - (\frac{\breve{\mu}_h^k}{\breve{n}_h^k})^2$. Moreover, we define $\Delta_h^k = \overline{V}_h^k(s_h^k) - \underline{V}_h^k(s_h^k)$ and $\zeta_h^k = \Delta_h^k - (\overline{Q}_h^k - \underline{Q}_h^k)(s_h^k, a_h^k, b_h^k)$. For convenience, we also define $\lambda_h^k(s) = \mathbf{1}\left\{n_h^k(s) < N_0\right\}$.

For certain functions, we use the superscript $k$ to denote the value of the function at the beginning of the $k$-th episode, and use the superscript $K + 1$ to denote the value of the function after all $K$

episodes are played. For instance, we denote $N_h^k(s, a, b)$ as the value of $N_h(s, a, b)$ at the beginning of the $k$-th episode, and $N_h^{K+1}(s, a, b)$ to denote the total number of visits of $(s, a, b)$ at step $h$ after $K$ episodes. When it is clear from the context, we omit the subscript $h$ and the superscript $k$ for notational convenience. For example, we use $\ell_i$ and $\check{\ell}_i$ to denote $\ell_{h,i}^k$ and $\check{\ell}_{h,i}^k$ when it is obvious what values that the indices $h$ and $k$ take.

## D  NOTATIONS

For any function $f : \mathcal{S} \mapsto \mathbb{R}$, we use $P_{s,a,b} f$ and $(P_h f)(s, a, b)$ interchangeably. Define $\mathbb{V}(x, y) = x^\top(y^2) - (x^\top y)^2$ for two vectors of the same dimension, where $y^2$ is obtained by squaring each entry of $y$.

For ease of exposition, we define $\overline{\nu}_h^{\mathrm{ref},k} = \frac{\overline{\sigma}_h^{\mathrm{ref},k}}{n_h^k} - (\frac{\overline{\mu}_h^{\mathrm{ref},k}}{n_h^k})^2$, $\underline{\nu}_h^{\mathrm{ref},k} = \frac{\underline{\sigma}_h^{\mathrm{ref},k}}{n_h^k} - (\frac{\underline{\mu}_h^{\mathrm{ref},k}}{n_h^k})^2$ and $\overline{\breve{\nu}}_h^k = \frac{\overline{\breve{\sigma}}_h^k}{\breve{n}_h^k} - (\frac{\overline{\breve{\mu}}_h^k}{\breve{n}_h^k})^2$, $\underline{\breve{\nu}}_h^k = \frac{\underline{\breve{\sigma}}_h^k}{\breve{n}_h^k} - (\frac{\underline{\breve{\mu}}_h^k}{\breve{n}_h^k})^2$. Moreover, we define $\Delta_h^k = \overline{V}_h^k(s_h^k) - \underline{V}_h^k(s_h^k)$ and $\zeta_h^k = \Delta_h^k - (\overline{Q}_h^k - \underline{Q}_h^k)(s_h^k, a_h^k, b_h^k)$. For convenience, we also define $\lambda_h^k(s) = \mathbf{1}\left\{n_h^k(s) < N_0\right\}$.

For certain functions, we use the superscript $k$ to denote the value of the function at the beginning of the $k$-th episode, and use the superscript $K + 1$ to denote the value of the function after all $K$ episodes are played. For instance, we denote $N_h^k(s, a, b)$ as the value of $N_h(s, a, b)$ at the beginning of the $k$-th episode, and $N_h^{K+1}(s, a, b)$ to denote the total number of visits of $(s, a, b)$ at step $h$ after $K$ episodes. When it is clear from the context, we omit the subscript $h$ and the superscript $k$ for notational convenience. For example, we use $\ell_i$ and $\check{\ell}_i$ to denote $\ell_{h,i}^k$ and $\check{\ell}_{h,i}^k$ when it is obvious what values that the indices $h$ and $k$ take.

## E  PROOF OF THEOREM 4.1

In this section, we provide the proof of Theorem 4.1, which consists of four main steps and one final step. In order to provide a clear proof flow here, we defer the proofs of the main lemmas in these steps to later sections (i.e., Appendix F-Appendix I).

We start by replacing $\delta$ by $\delta/\mathrm{poly}(H, T)$, and it suffices to show the desired bound for $V_1^{\dagger, \nu^{\mathrm{out}}}(s_1) - V_1^{\mu^{\mathrm{out}}, \dagger}(s_1)$ with probability $1 - \mathrm{poly}(H, T)\delta$.

**Step I:** We show that the Nash equilibrium (action-)value functions are always bounded between the optimistic and pessimistic (action-)value functions.

**Lemma E.1** (Restatement of Lemma 4.3). *Let $\delta \in (0, 1)$. With probability at least $1 - 2T(2H^2T^3 + 7)\delta$, it holds that for any $s, a, b, k, h$,*

$$\underline{Q}_h^k(s, a, b) \le Q_h^*(s, a, b) \le \overline{Q}_h^k(s, a, b),$$
$$\underline{V}_h^k(s) \le V_h^*(s) \le \overline{V}_h^k(s).$$

The proof of Lemma E.1 is provided in Appendix F. The **new technical development** lies in proving the inequality with respect to the action-value function, whose update rule features the min-gap reference-advantage decomposition.

**Step II:** We show that the occurrence of the large V-gap has bounded sample complexity independent of the number of episodes $K$.

**Lemma E.2** (Restatement of Lemma 4.4). *With probability $1 - O(T\delta)$, it holds that*

$$\sum_{k=1}^K \mathbf{1}\{\overline{V}_h^k(s_h^k) - \underline{V}_h^k(s_h^k) \ge \epsilon\} \le O(SABH^5\iota/\epsilon^2).$$

The proof is provided in Appendix G.

By the selection of the reference value functions, Lemma E.2 with $\epsilon$ setting to $\beta$, and the definition of $N_0$, we have the following corollary.

**Corollary E.3** (Restatement of Corollary 4.5). *Conditioned on the successful events of Lemma E.1 and Lemma E.2, for every state $s$, we have*

$$n_h^k(s) \geq N_0 \Longrightarrow \overline{V}_h^{\mathrm{ref},k}(s) - \underline{V}_h^{\mathrm{ref},k}(s) \leq \beta.$$

**Step III:** We bound $\sum_{k=1}^K (\overline{V}_1^k - \underline{V}_1^k)(s_1)$. Compared to single-agent RL, the CCE oracle leads to a possibly mixed policy and we need to bound the additional term due to the CCE oracle.

Recall the definition of $\Delta_h^k = \overline{V}_h^k(s_h^k) - \underline{V}_h^k(s_h^k)$ and $\zeta_h^k = \Delta_h^k - (\overline{Q}_h^k - \underline{Q}_h^k)(s_h^k, a_h^k, b_h^k)$. Following the update rule, we have

$$\Delta_h^k = \zeta_h^k + (\overline{Q}_h^k - \underline{Q}_h^k)(s_h^k, a_h^k, b_h^k)$$

$$\leq \zeta_h^k + H\mathbf{1}\{n_h^k = 0\} + \frac{1}{n_h^k} \sum_{i=1}^{n_h^k} \overline{V}_{h+1}^{\mathrm{ref},\ell_i}(s_{h+1}^{\ell_i}) - \frac{1}{n_h^k} \sum_{i=1}^{n_h^k} \underline{V}_{h+1}^{\mathrm{ref},\ell_i}(s_{h+1}^{\ell_i})$$

$$+ \frac{1}{\check{n}_h^k} \sum_{i=1}^{\check{n}_h^k} (\overline{V}_{h+1}^{\check{\ell}_i} - \overline{V}_{h+1}^{\mathrm{ref},\check{\ell}_i})(s_{h+1}^{\check{\ell}_i}) - \frac{1}{\check{n}_h^k} \sum_{i=1}^{\check{n}_h^k} (\underline{V}_{h+1}^{\check{\ell}_i} - \underline{V}_{h+1}^{\mathrm{ref},\check{\ell}_i})(s_{h+1}^{\check{\ell}_i}) + \overline{\beta}_h^k + \underline{\beta}_h^k$$

$$\leq \zeta_h^k + H\mathbf{1}\{n_h^k = 0\} + \frac{1}{n_h^k} \sum_{i=1}^{n_h^k} P_{s_h^k, a_h^k, b_h^k, h} \overline{V}_{h+1}^{\mathrm{ref},\ell_i} - \frac{1}{n_h^k} \sum_{i=1}^{n_h^k} P_{s_h^k, a_h^k, b_h^k, h} \underline{V}_{h+1}^{\mathrm{ref},\ell_i}$$

$$+ \frac{1}{\check{n}_h^k} \sum_{i=1}^{\check{n}_h^k} P_{s_h^k, a_h^k, b_h^k, h} (\overline{V}_{h+1}^{\check{\ell}_i} - \overline{V}_{h+1}^{\mathrm{ref},\check{\ell}_i}) - \frac{1}{\check{n}_h^k} \sum_{i=1}^{\check{n}_h^k} P_{s_h^k, a_h^k, b_h^k, h} (\underline{V}_{h+1}^{\check{\ell}_i} - \underline{V}_{h+1}^{\mathrm{ref},\check{\ell}_i}) + 2\overline{\beta}_h^k + 2\underline{\beta}_h^k$$

$$\tag{10}$$

$$= \zeta_h^k + H\mathbf{1}\{n_h^k = 0\} + P_{s_h^k, a_h^k, b_h^k, h} \left( \frac{1}{n_h^k} \sum_{i=1}^{n_h^k} \overline{V}_{h+1}^{\mathrm{ref},\ell_i} - \frac{1}{\check{n}_h^k} \sum_{i=1}^{\check{n}_h^k} \overline{V}_{h+1}^{\mathrm{ref},\check{\ell}_i} \right)$$

$$- P_{s_h^k, a_h^k, b_h^k, h} \left( \frac{1}{n_h^k} \sum_{i=1}^{n_h^k} \underline{V}_{h+1}^{\mathrm{ref},\ell_i} - \frac{1}{\check{n}_h^k} \sum_{i=1}^{\check{n}_h^k} \underline{V}_{h+1}^{\mathrm{ref},\check{\ell}_i} \right) + P_{s_h^k, a_h^k, b_h^k, h} \frac{1}{\check{n}_h^k} \sum_{i=1}^{\check{n}_h^k} \left( \overline{V}_{h+1}^{\check{\ell}_i} - \underline{V}_{h+1}^{\check{\ell}_i} \right)$$

$$+ 2\overline{\beta}_h^k + 2\underline{\beta}_h^k$$

$$\leq \zeta_h^k + H\mathbf{1}\{n_h^k = 0\} + P_{s_h^k, a_h^k, b_h^k, h} \left( \frac{1}{n_h^k} \sum_{i=1}^{n_h^k} \overline{V}_{h+1}^{\mathrm{ref},\ell_i} - \overline{V}_{h+1}^{\mathrm{REF}} \right)$$

$$- P_{s_h^k, a_h^k, b_h^k, h} \left( \frac{1}{n_h^k} \sum_{i=1}^{n_h^k} \underline{V}_{h+1}^{\mathrm{ref},\ell_i} - \underline{V}_{h+1}^{\mathrm{REF}} \right) + P_{s_h^k, a_h^k, b_h^k, h} \frac{1}{\check{n}_h^k} \sum_{i=1}^{\check{n}_h^k} \left( \overline{V}_{h+1}^{\check{\ell}_i} - \underline{V}_{h+1}^{\check{\ell}_i} \right) + 2\overline{\beta}_h^k + 2\underline{\beta}_h^k$$

$$\tag{11}$$

$$= \zeta_h^k + H\mathbf{1}\{n_h^k = 0\} + \frac{1}{\check{n}_h^k} \sum_{i=1}^{\check{n}_h^k} \Delta_{h+1}^{\check{\ell}_i} + \Lambda_{h+1}^k,$$

$$\tag{12}$$

where we define

$$\Lambda_{h+1}^k = \psi_{h+1}^k + \xi_{h+1}^k + 2\overline{\beta}_h^k + 2\underline{\beta}_h^k,$$

$$\psi_{h+1}^k = P_{s_h^k, a_h^k, b_h^k, h} \left( \frac{1}{n_h^k} \sum_{i=1}^{n_h^k} \left( \overline{V}_{h+1}^{\mathrm{ref},\ell_i} - \underline{V}_{h+1}^{\mathrm{ref},\ell_i} \right) - \left( \overline{V}_{h+1}^{\mathrm{REF}} - \underline{V}_{h+1}^{\mathrm{REF}} \right) \right),$$

$$\xi_{h+1}^k = \frac{1}{\check{n}_h^k} \sum_{i=1}^{\check{n}_h^k} \left( P_{s_h^k, a_h^k, b_h^k, h} - \mathbf{1}_{s_{h+1}^{\check{\ell}_i}} \right) \left( \overline{V}_{h+1}^{\check{\ell}_i} - \underline{V}_{h+1}^{\check{\ell}_i} \right).$$

Here, (10) follows from the successful event of martingale concentration (29) and (43) in Lemma E.1, (11) follows from the fact that $\overline{V}_{h+1}^{\mathrm{ref},u}(s)$ (or $\underline{V}_{h+1}^{\mathrm{ref},u}(s)$) is non-increasing (or non-decreasing) in $u$, because $\overline{V}_h^{\mathrm{ref}}(s)$ (or $\underline{V}_h^{\mathrm{ref}}(s)$) for a pair $(s,h)$ can only be updated once and the updated value is obviously greater (or less) than the initial value, and (12) follows from the definition of $\Lambda_{h+1}^k$ defined above.

Taking the summation over $k \in [K]$ gives

$$\sum_{k=1}^K \Delta_h^k \le \sum_{k=1}^K \zeta_h^k + \sum_{k=1}^K H \mathbf{1}\{n_h^k = 0\} + \sum_{k=1}^K \frac{1}{\check{n}_h^k} \sum_{i=1}^{\check{n}_h^k} \Delta_{h+1}^{\check{\ell}_{h,i}^k} + \sum_{k=1}^K \Lambda_{h+1}^k. \tag{13}$$

Note that $n_h^k \ge H$ if $N_h^k(s_h^k, a_h^k, b_h^k) \ge H$. Therefore $\sum_{k=1}^K \mathbf{1}\{n_h^k = 0\} \le SABH$, and

$$\sum_{k=1}^K H \mathbf{1}\{n_h^k = 0\} \le SABH^2. \tag{14}$$

Now we focus on the term $\sum_{k=1}^K \frac{1}{\check{n}_h^k} \sum_{i=1}^{\check{n}_h^k} \Delta_{h+1}^{\check{\ell}_{h,i}^k}$. The following lemma is useful.

**Lemma E.4.** *For any $j \in [K]$, we have $\sum_{k=1}^K \frac{1}{\check{n}_h^k} \sum_{i=1}^{\check{n}_h^k} \mathbf{1}\{j = \check{\ell}_{h,i}^k\} \le 1 + \frac{1}{H}$.*

*Proof.* Fix an episode $j$. Note that $\sum_{i=1}^{\check{n}_h^k} \mathbf{1}\{j = \check{\ell}_{h,i}^k\} = 1$ if and only if $(s_h^j, a_h^j, b_h^j) = (s_h^k, a_h^k, b_h^k)$ and $(j, h)$ falls in the previous stage that $(k, h)$ falls in with respect to $(s_h^k, a_h^k, b_h^k, h)$. Define $\mathcal{K} = \{k \in [K] : \sum_{i=1}^{\check{n}_h^k} \mathbf{1}\{j = \check{\ell}_{h,i}^k\} = 1\}$. Then every element $k \in \mathcal{K}$ has the same value of $\check{n}_h^k$, i.e., there exists an integer $N_j > 0$ such that $\check{n}_h^k = N_j$ for all $k \in \mathcal{K}$. By the definition of stages, $|\mathcal{K}| \le (1 + \frac{1}{H})N_j$. Therefore, for any $j$, we have $\sum_{k=1}^K \frac{1}{\check{n}_h^k} \sum_{i=1}^{\check{n}_h^k} \mathbf{1}\{j = \check{\ell}_{h,i}^k\} \le (1 + \frac{1}{H})$. $\square$

By Lemma E.4, we have

$$\begin{aligned}
\sum_{k=1}^K \frac{1}{\check{n}_h^k} \sum_{i=1}^{\check{n}_h^k} \Delta_{h+1}^{\check{\ell}_{h,i}^k} &= \sum_{k=1}^K \frac{1}{\check{n}_h^k} \sum_{j=1}^K \Delta_{h+1}^j \sum_{i=1}^{\check{n}_h^k} \mathbf{1}\{j = \check{\ell}_{h,i}^k\} \\
&= \sum_{j=1}^K \Delta_{h+1}^j \sum_{k=1}^K \frac{1}{\check{n}_h^k} \sum_{i=1}^{\check{n}_h^k} \mathbf{1}\{j = \check{\ell}_{h,i}^k\} \\
&\le (1 + \frac{1}{H}) \sum_{k=1}^K \Delta_{h+1}^k. \tag{15}
\end{aligned}$$

Combining (13), (14) and (15), we have

$$\sum_{k=1}^K \Delta_h^k \le SABH^2 + (1 + \frac{1}{H}) \sum_{k=1}^K \Delta_{h+1}^k + \sum_{k=1}^K \Lambda_{h+1}^k.$$

Iterating over $h = H, H-1, \ldots, 1$ gives

$$\sum_{k=1}^K \Delta_1^k \le \mathcal{O}\left(SABH^3 + \sum_{h=1}^H \sum_{k=1}^K (1 + \frac{1}{H})^{h-1} \zeta_h^k + \sum_{h=1}^H \sum_{k=1}^K (1 + \frac{1}{H})^{h-1} \Lambda_{h+1}^k\right).$$

By Azuma's inequality, it holds that with probability at least $1 - T\delta$,

$$\sum_{k=1}^K \Delta_1^k \le \mathcal{O}\left(SABH^3 + \sqrt{H^2 T\iota} + \sum_{h=1}^H \sum_{k=1}^K (1 + \frac{1}{H})^{h-1} \Lambda_{h+1}^k\right). \tag{16}$$

**Step IV:** We bound $\sum_{h=1}^H \sum_{k=1}^K (1 + \frac{1}{H})^{h-1} \Lambda_{h+1}^k$ in the following lemma.

**Lemma E.5** (Restatement of Lemma 4.6). *With probability at least $1 - O(H^2T^4)\delta$, it holds that*

$$\sum_{h=1}^{H}\sum_{k=1}^{K}(1+\frac{1}{H})^{h-1}\Lambda_{h+1}^{k} = O\left(\sqrt{SABH^2T\iota} + H\sqrt{T\iota}\log T + S^2(AB)^{\frac{3}{2}}H^8\iota^{\frac{3}{2}}T^{\frac{1}{4}}\right).$$

The proof of Lemma E.5 is provided in Appendix H.

**Final step:** We show the value difference induced by the certified policies is bounded, as summarized in the next lemma.

**Lemma E.6** (Restatement of Lemma 4.2). *Conditioned on the successful event of Lemma E.1, let $(\mu^{\mathrm{out}}, \nu^{\mathrm{out}})$ be the output policy induced by the certified policy algorithm (Algorithm 2). Then we have*

$$V_1^{\dagger,\nu^{\mathrm{out}}}(s_1) - V_1^{\mu^{\mathrm{out}},\dagger}(s_1) \leq \frac{1}{K}\sum_{k=1}^{K}(\overline{V}_1^k - \underline{V}_1^k)(s_1).$$

The proof of Lemma E.6 is provided in Appendix I.

Combining (16), Lemma E.5 and Lemma E.6, and taking the union bound over all probability events, we conclude that with probability at least $1 - O(H^2T^4)\delta$, it holds that

$$V_1^{\dagger,\nu^{\mathrm{out}}}(s_1) - V_1^{\mu^{\mathrm{out}},\dagger}(s_1) \leq \frac{1}{K}O\left(\sqrt{SABH^2T\iota} + H\sqrt{T\iota}\log T + S^2(AB)^{\frac{3}{2}}H^8\iota^{\frac{3}{2}}T^{\frac{1}{4}}\right), \quad (17)$$

which gives the desired result.

## F  PROOF OF LEMMA E.1 (STEP I)

The proof is by induction on $k$. We establish the inequalities for the optimistic action-value and value functions in **step i**, and the inequalities for the pessimistic counterparts in **step ii**.

**Step i:** We establish the inequality for the optimistic action-value and value functions in the following.

It is clear that the conclusion holds for the based case with $k = 1$. For $k \geq 2$, assume $Q_h^*(s,a,b) \leq \overline{Q}_h^u(s,a,b)$ and $V_h^*(s) \leq \overline{V}_h^u(s)$ for any $(s,a,h) \in \mathcal{S} \times \mathcal{A} \times [H]$ and $u \in [1,k]$. Fix tuple $(s,a,b,h)$. We next show that the conclusion holds for $k+1$.

First, we show the inequality with respect to the action-value function. If $\overline{Q}_h(s,a,b), \overline{V}_h(s)$ are not updated in the $k$-th episode, then

$$Q_h^*(s,a,b) \leq \overline{Q}_h^k(s,a,b) = \overline{Q}_h^{k+1}(s,a,b),$$
$$V_h^*(s) \leq \overline{V}_h^k(s) = \overline{V}_h^{k+1}(s).$$

Otherwise, we have

$$\overline{Q}_h^{k+1}(s,a,b) \leftarrow \min\left\{r_h(s,a,b) + \frac{\check{v}}{\check{n}} + \gamma, r_h(s,a,b) + \frac{\overline{\mu}^{\mathrm{ref}}}{n} + \frac{\check{\mu}}{\check{n}} + \overline{\beta}, \overline{Q}_h^k(s,a,b)\right\}.$$

Besides the last term, there are two non-trivial cases.

**For the first case**, by Hoeffding's inequality, with probability at least $1 - \delta$ it holds that

$$\overline{Q}_h^{k+1}(s,a,b) = r_h(s,a,b) + \frac{\check{v}}{\check{n}} + \gamma$$

$$= r_h(s,a,b) + \frac{1}{\check{n}}\sum_{i=1}^{\check{n}}\overline{V}_{h+1}^{\check{\ell}_i}(s_{h+1}^{\check{\ell}_i}) + 2\sqrt{\frac{H^2}{\check{n}}\iota}$$

$$\geq r_h(s,a,b) + \frac{1}{\check{n}}\sum_{i=1}^{\check{n}}V_{h+1}^*(s_{h+1}^{\check{\ell}_i}) + 2\sqrt{\frac{H^2}{\check{n}}\iota} \quad (18)$$

$$\geq r_h(s,a,b) + (P_h V_{h+1}^*)(s,a,b) \quad (19)$$

$$= Q_h^*(s, a, b),$$

where (18) follows from the induction hypothesis $\overline{V}_{h+1}^u(s) \geq V^*(s)$ for all $u \in [k]$, and (19) follows from Azuma-Hoeffding's inequality.

**For the second case**, we have

$$\overline{Q}_h^{k+1}(s, a, b) = r_h(s, a, b) + \frac{\overline{\mu}^{\text{ref}}}{n} + \frac{\check{\overline{\mu}}}{\check{n}} + \overline{\beta}$$

$$= r_h(s, a, b) + \frac{1}{n} \sum_{i=1}^n \overline{V}_{h+1}^{\text{ref}, \ell_i}(s_{h+1}^{\ell_i}) + \frac{1}{\check{n}} \sum_{i=1}^{\check{n}} \left( \overline{V}_{h+1}^{\check{\ell}_i} - \overline{V}_{h+1}^{\text{ref}, \check{\ell}_i} \right)(s_{h+1}^{\check{\ell}_i}) + \overline{\beta}$$

$$= r_h(s, a, b) + \left( P_h \left( \frac{1}{n} \sum_{i=1}^n \overline{V}_{h+1}^{\text{ref}, \ell_i} \right) \right)(s, a, b) + \left( P_h \left( \frac{1}{\check{n}} \sum_{i=1}^{\check{n}} \left( \overline{V}_{h+1}^{\check{\ell}_i} - \overline{V}_{h+1}^{\text{ref}, \check{\ell}_i} \right) \right) \right)(s, a, b)$$

$$\quad + \chi_1 + \chi_2 + \overline{\beta}$$

$$\geq r_h(s, a, b) + \left( P_h \left( \frac{1}{\check{n}} \sum_{i=1}^{\check{n}} \overline{V}_{h+1}^{\check{\ell}_i} \right) \right)(s, a, b) + \chi_1 + \chi_2 + \overline{\beta} \tag{20}$$

$$\geq r_h(s, a, b) + \left( P_h V_{h+1}^* \right)(s, a, b) + \chi_1 + \chi_2 + \overline{\beta} \tag{21}$$

$$= \overline{Q}_h^*(s, a, b) + \chi_1 + \chi_2 + \overline{\beta},$$

where

$$\chi_1(k, h) = \frac{1}{n} \sum_{i=1}^n \left( \overline{V}_h^{\text{ref}, \ell_i}(s_{h+1}^{\ell_i}) - \left( P_h \overline{V}_{h+1}^{\text{ref}, \ell_i} \right)(s, a, b) \right),$$

$$\overline{W}_{h+1}^\ell = \overline{V}_{h+1}^\ell - \overline{V}_{h+1}^{\text{ref}, \ell}$$

$$\chi_2(k, h) = \frac{1}{\check{n}} \sum_{i=1}^{\check{n}} \left( \overline{W}_{h+1}^{\check{\ell}_i}(s_{h+1}^{\check{\ell}_i}) - \left( P_h \overline{W}_{h+1}^{\check{\ell}_i} \right)(s, a, b) \right).$$

Here, (20) follows from the fact that $\overline{V}_{h+1}^{\text{ref}, u}(s)$ is non-increasing in $u$ (since $\overline{V}_h^{\text{ref}}(s)$ for a pair $(s, h)$ can only be updated once and the updated value is obviously smaller than the initial value $H$), and (21) follows from the the induction hypothesis $\overline{V}_{h+1}^k(s) \geq V_{h+1}^*(s)$.

By Lemma J.2 with $\epsilon = \frac{1}{T^2}$, with probability at least $1 - 2(H^2 T^3 + 1)\delta$ it holds

$$|\chi_1(k, h)| \leq 2\sqrt{\frac{\sum_{i=1}^n \mathbb{V}(P_{s,a,b,h}, \overline{V}_{h+1}^{\text{ref}, \ell_i})\iota}{n^2}} + \frac{2\sqrt{\iota}}{Tn} + \frac{2H\iota}{n}, \tag{22}$$

$$|\chi_2(k, h)| \leq 2\sqrt{\frac{\sum_{i=1}^{\check{n}} \mathbb{V}(P_{s,a,b,h}, \overline{V}_{h+1}^{\text{ref}, \ell_i})\iota}{\check{n}^2}} + \frac{2\sqrt{\iota}}{T\check{n}} + \frac{2H\iota}{\check{n}}. \tag{23}$$

**Lemma F.1.** *With probability at least $1 - 2\delta$, it holds that*

$$\sum_{i=1}^n \mathbb{V}(P_{s,a,b,h}, \overline{V}_{h+1}^{\text{ref}, \ell_i}) \leq n\overline{\nu}^{\text{ref}} + 3H^2\sqrt{n\iota}. \tag{24}$$

**Proof:** Note that

$$\sum_{i=1}^n \mathbb{V}(P_{s,a,b,h}, \overline{V}_{h+1}^{\text{ref}, \ell_i}) = \sum_{i=1}^n \left( P_{s,a,b,h}(\overline{V}_{h+1}^{\text{ref}, \ell_i})^2 - (P_{s,a,b,h}\overline{V}_{h+1}^{\text{ref}, \ell_i})^2 \right)$$

$$= \sum_{i=1}^n (\overline{V}_{h+1}^{\text{ref}, \ell_i}(s_{h+1}^{\ell_i}))^2 - \frac{1}{n} \left( \sum_{i=1}^n \overline{V}_{h+1}^{\text{ref}, \ell_i}(s_{h+1}^{\ell_i}) \right)^2 + \chi_3 + \chi_4 + \chi_5$$

$$= n\overline{\nu}^{\text{ref}} + \chi_3 + \chi_4 + \chi_5, \tag{25}$$

where

$$\chi_3 = \sum_{i=1}^n \left( (P_{s,a,b,h}(\overline{V}_{h+1}^{\mathrm{ref},\ell_i})^2 - (\overline{V}_{h+1}^{\mathrm{ref},\ell_i}(s_{h+1}^{\ell_i}))^2 \right),$$

$$\chi_4 = \frac{1}{n} \left( \sum_{i=1}^n \overline{V}_{h+1}^{\mathrm{ref},\ell_i}(s_{h+1}^{\ell_i}) \right)^2 - \frac{1}{n} \left( \sum_{i=1}^n P_{s,a,b,h}\overline{V}_{h+1}^{\mathrm{ref},\ell_i} \right)^2,$$

$$\chi_5 = \frac{1}{n} \left( \sum_{i=1}^n P_{s,a,b,h}\overline{V}_{h+1}^{\mathrm{ref},\ell_i} \right)^2 - \sum_{i=1}^n (P_{s,a,b,h}\overline{V}_{h+1}^{\mathrm{ref},\ell_i})^2.$$

By Azuma's inequality, with probability at least $1-\delta$ it holds that $|\chi_3| \le H^2\sqrt{2n\iota}$.

By Azuma's inequality, with probability at least $1-\delta$, it holds that

$$\begin{aligned}
|\chi_4| &= \frac{1}{n} \left| \left( \sum_{i=1}^n \overline{V}_{h+1}^{\mathrm{ref},\ell_i}(s_{h+1}^{\ell_i}) \right)^2 - \left( \sum_{i=1}^n P_{s,a,b,h}\overline{V}_{h+1}^{\mathrm{ref},\ell_i} \right)^2 \right| \\
&\le 2H \left| \sum_{i=1}^n \overline{V}_{h+1}^{\mathrm{ref},\ell_i}(s_{h+1}^{\ell_i}) - \sum_{i=1}^n P_{s,a,b,h}\overline{V}_{h+1}^{\mathrm{ref},\ell_i} \right| \\
&\le 2H^2\sqrt{2n\iota}.
\end{aligned}$$

Moreover, $\chi_5 \le 0$ by Cauchy-Schwartz inequality. Plugging the above inequalities gives the desired result. ∎

Combining (22) with (24) gives

$$|\chi_1| \le 2\sqrt{\frac{\nu^{\mathrm{ref}}\iota}{n}} + \frac{5H\iota^{\frac{3}{4}}}{n^{\frac{3}{4}}} + \frac{2\sqrt{\iota}}{Tn} + \frac{2H\iota}{n}. \tag{26}$$

Similar to Lemma F.1, we have the following lemma.

**Lemma F.2.** *With probability at least $1-2\delta$, it holds that*

$$\sum_{i=1}^{\check{n}} \mathbb{V}(P_{s,a,b,h}, \overline{W}_{h+1}^{\mathrm{ref},\ell_i}) \le \check{n}\check{\nu} + 3H^2\sqrt{\check{n}\iota}. \tag{27}$$

Combining (23) with (27) gives

$$|\chi_2| \le 2\sqrt{\frac{\check{\nu}\iota}{\check{n}}} + \frac{5H\iota^{\frac{3}{4}}}{\check{n}^{\frac{3}{4}}} + \frac{2\sqrt{\iota}}{T\check{n}} + \frac{2H\iota}{\check{n}}. \tag{28}$$

Finally, combining (26) and (28), noting the definition of $\overline{\beta}$ with $(c_1, c_2, c_3) = (2, 2, 5)$, and taking a union bound over all probability events, we have that with probability at least $1 - 2(H^2T^3 + 3)\delta$, it holds that

$$\overline{\beta} \ge |\chi_1| + |\chi_2|. \tag{29}$$

which means $\overline{Q}_h^{k+1}(s,a,b) \ge Q_h^*(s,a,b)$.

Combining the two cases and taking the union bound over all steps, we have with probability at least $1 - T(2H^2T^3 + 7)\delta$, it holds that $\overline{Q}_h^{k+1}(s,a,b) \ge Q_h^*(s,a,b)$.

Next, we show that $V_h^*(s) \le \overline{V}_h^{k+1}(s)$. Note that

$$\begin{aligned}
\overline{V}_h^{k+1}(s) &= (\mathbb{D}_{\pi_h^{k+1}}\overline{Q}_h^{k+1})(s) \\
&\ge \sup_{\mu \in \Delta_{\mathcal{A}}} (\mathbb{D}_{\mu \times \nu_h^{k+1}}\overline{Q}_h^{k+1})(s)
\end{aligned} \tag{30}$$

$$\geq \sup_{\mu \in \Delta_{\mathcal{A}}} (\mathbb{D}_{\mu \times \nu_h^{k+1}} Q_h^*)(s) \tag{31}$$

$$\geq \sup_{\mu \in \Delta_{\mathcal{A}}} \inf_{\nu \in \Delta_{\mathcal{B}}} (\mathbb{D}_{\mu \times \nu} Q_h^*)(s)$$

$$= V_h^*(s),$$

where (30) follows from the property of the CCE oracle, (31) follows because $\overline{Q}_h^{k+1}(s,a,b) \geq \overline{Q}_h^*(s,a,b)$, which has just been proved.

**Step ii:** We show the inequalities for the pessimistic action-value function and value function below.

The two inequalities with respect to pessimistic (action-)value functions clearly hold for $k = 1$. For $k \geq 2$, suppose $Q_h^*(s,a,b) \geq \underline{Q}_h^u(s,a,b)$ and $V_h^*(s) \geq \underline{V}_h^u(s)$ for any $(s,a,h) \in \mathcal{S} \times \mathcal{A} \times [H]$ and $u \in [1, k]$. Now we fix tuple $(s, a, b, h)$ and we only need to consider the case when $\underline{Q}_h(s,a,b)$ and $\underline{V}_h(s)$ are updated.

We show $Q_h^*(s,a,b) \geq \underline{Q}_h^{k+1}(s,a,b)$. Note that

$$\underline{Q}_h^{k+1}(s,a,b) \leftarrow \min \left\{ r_h(s,a,b) + \frac{\check{\underline{v}}}{\check{n}} + \gamma, r_h(s,a,b) + \frac{\underline{\mu}^{\text{ref}}}{n} + \frac{\check{\underline{\mu}}}{\check{n}} + \underline{\beta}, \underline{Q}_h^k(s,a,b) \right\},$$

and we have two non-trivial cases.

**For the first case**, by Hoeffding's inequality, with probability at least $1 - \delta$, it holds that

$$\underline{Q}_h^{k+1}(s,a,b) = r_h(s,a,b) + \frac{\check{\underline{v}}}{\check{n}} - \gamma$$

$$= r_h(s,a,b) + \frac{1}{\check{n}} \sum_{i=1}^{\check{n}} \underline{V}_{h+1}^{\check{\ell}_i}(s_{h+1}^{\check{\ell}_i}) - 2\sqrt{\frac{H^2}{\check{n}}\iota}$$

$$\leq r_h(s,a,b) + \frac{1}{\check{n}} \sum_{i=1}^{\check{n}} V_{h+1}^*(s_{h+1}^{\check{\ell}_i}) - 2\sqrt{\frac{H^2}{\check{n}}\iota} \tag{32}$$

$$\leq r_h(s,a,b) + (P_h V_{h+1}^*)(s,a,b) \tag{33}$$

$$= Q_h^*(s,a,b),$$

where (32) follows from the induction hypothesis $\underline{V}_{h+1}^u(s) \geq V^*(s)$ for all $u \in [k]$, and (33) follows from Azuma-Hoeffding's inequality.

**For the second case**, we have

$$\underline{Q}_h^{k+1}(s,a,b) = r_h(s,a,b) + \frac{\underline{\mu}^{\text{ref}}}{n} + \frac{\check{\underline{\mu}}}{\check{n}} - \underline{\beta}$$

$$= r_h(s,a,b) + \frac{1}{n} \sum_{i=1}^{n} \underline{V}_{h+1}^{\text{ref},\ell_i}(s_{h+1}^{\ell_i}) + \frac{1}{\check{n}} \sum_{i=1}^{\check{n}} \left( \underline{V}_{h+1}^{\check{\ell}_i} - \underline{V}_{h+1}^{\text{ref},\check{\ell}_i} \right) (s_{h+1}^{\check{\ell}_i}) - \underline{\beta}$$

$$= r_h(s,a,b) + \left( P_h \left( \frac{1}{n} \sum_{i=1}^{n} \underline{V}_{h+1}^{\text{ref},\ell_i} \right) \right)(s,a,b) + \left( P_h \left( \frac{1}{\check{n}} \sum_{i=1}^{\check{n}} \left( \underline{V}_{h+1}^{\check{\ell}_i} - \underline{V}_{h+1}^{\text{ref},\check{\ell}_i} \right) \right) \right)(s,a,b)$$

$$\quad + \underline{\chi}_1 + \underline{\chi}_2 - \underline{\beta}$$

$$\leq r_h(s,a,b) + \left( P_h \left( \frac{1}{\check{n}} \sum_{i=1}^{\check{n}} \underline{V}_{h+1}^{\check{\ell}_i} \right) \right)(s,a,b) + \underline{\chi}_1 + \underline{\chi}_2 - \underline{\beta} \tag{34}$$

$$\leq r_h(s,a,b) + \left( P_h V_{h+1}^* \right)(s,a,b) + \underline{\chi}_1 + \underline{\chi}_2 - \underline{\beta} \tag{35}$$

$$= \underline{Q}_h^*(s,a,b) + \underline{\chi}_1 + \underline{\chi}_2 - \underline{\beta},$$

where

$$\underline{\chi}_1(k,h) = \frac{1}{n} \sum_{i=1}^{n} \left( \underline{V}_h^{\text{ref},\ell_i}(s_{h+1}^{\ell_i}) - \left( P_h \underline{V}_{h+1}^{\text{ref},\ell_i} \right)(s,a,b) \right),$$

$$\underline{W}_{h+1}^{\ell} = \underline{V}_{h+1}^{\ell} - \underline{V}_{h+1}^{\mathrm{ref},\ell}$$

$$\underline{\chi}_2(k,h) = \frac{1}{\check{n}} \sum_{i=1}^{\check{n}} \left( \underline{W}_{h+1}^{\check{\ell}_i}(s_{h+1}^{\check{\ell}_i}) - \left( P_h \underline{W}_{h+1}^{\check{\ell}_i} \right)(s,a,b) \right).$$

Here, (34) follows from the fact that $\underline{V}_{h+1}^{\mathrm{ref},u}(s)$ is non-decreasing in $u$ (since $\underline{V}_h^{\mathrm{ref}}(s)$ for a pair $(s,h)$ can only be updated once and the updated value is obviously greater than the initial value 0), and (35) follows from the induction hypothesis $\underline{V}_{h+1}^k(s) \le V_{h+1}^*(s)$.

By Lemma J.2 with $\epsilon = \frac{1}{T^2}$, with probability at least $1 - 2(H^2 T^3 + 1)\delta$ it holds

$$|\underline{\chi}_1(k,h)| \le 2\sqrt{\frac{\sum_{i=1}^n \mathbb{V}(P_{s,a,b,h}, \underline{V}_{h+1}^{\mathrm{ref},\ell_i})\iota}{n^2}} + \frac{2\sqrt{\iota}}{Tn} + \frac{2H\iota}{n}, \tag{36}$$

$$|\underline{\chi}_2(k,h)| \le 2\sqrt{\frac{\sum_{i=1}^{\check{n}} \mathbb{V}(P_{s,a,b,h}, \underline{V}_{h+1}^{\mathrm{ref},\ell_i})\iota}{\check{n}^2}} + \frac{2\sqrt{\iota}}{T\check{n}} + \frac{2H\iota}{\check{n}}. \tag{37}$$

**Lemma F.3.** *With probability at least $1 - 2\delta$, it holds that*

$$\sum_{i=1}^n \mathbb{V}(P_{s,a,b,h}, \underline{V}_{h+1}^{\mathrm{ref},\ell_i}) \le n\underline{\nu}^{\mathrm{ref}} + 3H^2\sqrt{n\iota} \tag{38}$$

**Proof:** Note that

$$\sum_{i=1}^n \mathbb{V}(P_{s,a,b,h}, \underline{V}_{h+1}^{\mathrm{ref},\ell_i}) = \sum_{i=1}^n \left( P_{s,a,b,h}(\underline{V}_{h+1}^{\mathrm{ref},\ell_i})^2 - (P_{s,a,b,h}\underline{V}_{h+1}^{\mathrm{ref},\ell_i})^2 \right)$$

$$= \sum_{i=1}^n (\underline{V}_{h+1}^{\mathrm{ref},\ell_i}(s_{h+1}^{\ell_i}))^2 - \frac{1}{n}\left( \sum_{i=1}^n \underline{V}_{h+1}^{\mathrm{ref},\ell_i}(s_{h+1}^{\ell_i}) \right)^2 + \underline{\chi}_3 + \underline{\chi}_4 + \underline{\chi}_5$$

$$= n\underline{\nu}^{\mathrm{ref}} + \underline{\chi}_3 + \underline{\chi}_4 + \underline{\chi}_5, \tag{39}$$

where

$$\underline{\chi}_3 = \sum_{i=1}^n \left( (P_{s,a,b,h}(\underline{V}_{h+1}^{\mathrm{ref},\ell_i})^2 - (\underline{V}_{h+1}^{\mathrm{ref},\ell_i}(s_{h+1}^{\ell_i}))^2 \right),$$

$$\underline{\chi}_4 = \frac{1}{n}\left( \sum_{i=1}^n \underline{V}_{h+1}^{\mathrm{ref},\ell_i}(s_{h+1}^{\ell_i}) \right)^2 - \frac{1}{n}\left( \sum_{i=1}^n P_{s,a,b,h}\underline{V}_{h+1}^{\mathrm{ref},\ell_i} \right)^2,$$

$$\underline{\chi}_5 = \frac{1}{n}\left( \sum_{i=1}^n P_{s,a,b,h}\underline{V}_{h+1}^{\mathrm{ref},\ell_i} \right)^2 - \sum_{i=1}^n (P_{s,a,b,h}\underline{V}_{h+1}^{\mathrm{ref},\ell_i})^2.$$

By Azuma's inequality, with probability at least $1 - \delta$ it holds that $|\underline{\chi}_3| \le H^2\sqrt{2n\iota}$.

By Azuma's inequality, with probability at least $1 - \delta$, it holds that

$$|\underline{\chi}_4| = \frac{1}{n}\left| \left( \sum_{i=1}^n \underline{V}_{h+1}^{\mathrm{ref},\ell_i}(s_{h+1}^{\ell_i}) \right)^2 - \left( \sum_{i=1}^n P_{s,a,b,h}\underline{V}_{h+1}^{\mathrm{ref},\ell_i} \right)^2 \right|$$

$$\le 2H\left| \sum_{i=1}^n \underline{V}_{h+1}^{\mathrm{ref},\ell_i}(s_{h+1}^{\ell_i}) - \sum_{i=1}^n P_{s,a,b,h}\underline{V}_{h+1}^{\mathrm{ref},\ell_i} \right|$$

$$\le 2H^2\sqrt{2n\iota}.$$

Moreover, $\chi_5 \le 0$ by Cauchy-Schwartz inequality. Substituting the above inequalities gives the desired result. ∎

Combining (36) with (38) gives

$$|\underline{\chi}_1| \leq 2\sqrt{\frac{\nu^{\mathrm{ref}}\iota}{n}} + \frac{5H\iota^{\frac{3}{4}}}{n^{\frac{3}{4}}} + \frac{2\sqrt{\iota}}{Tn} + \frac{2H\iota}{n}. \tag{40}$$

Similar to Lemma F.3, we have the following lemma.

**Lemma F.4.** *With probability at least $1 - 2\delta$, it holds that*

$$\sum_{i=1}^{\check{n}} \mathbb{V}(P_{s,a,b,h}, \underline{W}_{h+1}^{\mathrm{ref},\ell_i}) \leq \check{n}\underline{\nu} + 3H^2\sqrt{\check{n}\iota}. \tag{41}$$

Combining (37) with (41) gives

$$|\underline{\chi}_2| \leq 2\sqrt{\frac{\check{\nu}\iota}{\check{n}}} + \frac{5H\iota^{\frac{3}{4}}}{\check{n}^{\frac{3}{4}}} + \frac{2\sqrt{\iota}}{T\check{n}} + \frac{2H\iota}{\check{n}}. \tag{42}$$

Finally, combining (40) and (42), noting the definition of $\underline{\beta}$ with $(c_1, c_2, c_3) = (2, 2, 5)$, and taking a union bound over all probability events, we have that with probability at least $1 - 2(H^2T^3 + 3)\delta$, it holds that

$$\underline{\beta} \geq |\underline{\chi}_1| + |\underline{\chi}_2|. \tag{43}$$

which gives $\underline{Q}_h^{k+1}(s, a, b) \leq Q_h^*(s, a, b)$.

Combining the two cases and taking union bound over all steps, we have with probability at least $1 - T(2H^2T^3 + 7)\delta$, it holds that $\underline{Q}_h^{k+1}(s, a, b) \leq Q_h^*(s, a, b)$.

We show that $V_h^*(s) \leq \underline{V}_h^k(s)$. Note that

$$\underline{V}_h^{k+1}(s) = (\mathbb{D}_{\pi_h^{k+1}}\underline{Q}_h^{k+1})(s)$$

$$\leq \inf_{\nu \in \Delta_{\mathcal{B}}} (\mathbb{D}_{\mu_h^{k+1} \times \nu}\underline{Q}_h^{k+1})(s) \tag{44}$$

$$\leq \inf_{\nu \in \Delta_{\mathcal{B}}} (\mathbb{D}_{\mu_h^{k+1} \times \nu}Q_h^*)(s) \tag{45}$$

$$\leq \inf_{\nu \in \Delta_{\mathcal{B}}} \sup_{\mu \in \Delta_{\mathcal{A}}} (\mathbb{D}_{\mu \times \nu}Q_h^*)(s)$$

$$= V_h^*(s),$$

where (44) follows from the property of the CCE oracle, (45) follows because $\underline{Q}_h^{k+1}(s, a, b) \leq \underline{Q}_h^*(s, a, b)$, which has just been proved.

The entire proof is completed by combining **step i** and **step ii**, and taking a union bound over all probability events.

## G   PROOF OF LEMMA E.2 (STEP II)

First, by Hoeffding's inequality, for any $(k, h) \in [K] \times [H]$, with probability at least $1 - 2T\delta$ it holds that

$$\left| \frac{1}{\check{n}_h^k} \sum_{i=1}^{\check{n}_h^k} \overline{V}_{h+1}^{\check{\ell}_i}(s_{h+1}^{\check{\ell}_i}) - \overline{Q}_h^k(s_h^k, a_h^k, b_h^k) \right| \leq \gamma_h^k, \quad \left| \frac{1}{\check{n}_h^k} \sum_{i=1}^{\check{n}_h^k} \underline{V}_{h+1}^{\check{\ell}_i}(s_{h+1}^{\check{\ell}_i}) - \underline{Q}_h^k(s_h^k, a_h^k, b_h^k) \right| \leq \gamma_h^k.$$

The entire proof will be conditioned on the above event.

For any weight sequence $\{w_k\}_{k=1}^K$ where $w_k \geq 0$, let $\|w\|_\infty = \max_{1 \leq k \leq K} w_k$ and $\|w\|_1 = \sum_{k=1}^K w_k$.

By the update rule of the action-value function, we have

$$\Delta_h^k = (\overline{V}_h^k - \underline{V}_h^k)(s_h^k)$$

$$= \zeta_h^k + (\overline{Q}_h^k - \underline{Q}_h^k)(s_h^k, a_h^k, b_h^k)$$

$$\leq \zeta_h^k + 2\gamma_h^k + \frac{1}{\check{n}_h^k} \sum_{i=1}^{\check{n}_h^k} (\overline{V}_{h+1}^{\check{\ell}_i} - \underline{V}_{h+1}^{\check{\ell}_i})(s_{h+1}^{\check{\ell}_i}) + H\mathbf{1}\{n_h^k = 0\}$$

$$= \zeta_h^k + 2\gamma_h^k + \frac{1}{\check{n}_h^k} \sum_{i=1}^{\check{n}_h^k} \Delta_{h+1}^{\check{\ell}_i} + H\mathbf{1}\{n_h^k = 0\}. \tag{46}$$

Note that

$$\sum_{k=1}^{K} \frac{w_k}{\check{n}_h^k} \sum_{i=1}^{\check{n}_h^k} \Delta_{h+1}^{\check{\ell}_i} = \sum_{j=1}^{K} \frac{w_j}{\check{n}_h^j} \sum_{i=1}^{\check{n}_h^j} \Delta_{h+1}^{\check{\ell}_{h,i}^j}$$

$$= \sum_{j=1}^{K} \frac{w_j}{\check{n}_h^j} \sum_{k=1}^{K} \Delta_{h+1}^k \sum_{i=1}^{\check{n}_h^j} \mathbf{1}\{k = \check{\ell}_{h,i}^j\}$$

$$= \sum_{k=1}^{K} \Delta_{h+1}^k \sum_{j=1}^{K} \frac{w_j}{\check{n}_h^j} \sum_{i=1}^{\check{n}_h^j} \mathbf{1}\{k = \check{\ell}_{h,i}^j\}$$

$$= \sum_{k=1}^{K} \widetilde{w}_k \Delta_{h+1}^k, \tag{47}$$

where we define $\widetilde{w}_k = \sum_{j=1}^{K} \frac{w_j}{\check{n}_h^j} \sum_{i=1}^{\check{n}_h^j} \mathbf{1}\{k = \check{\ell}_{h,i}^j\}$. Similar to the proof of Lemma E.4, we have

$$\|\widetilde{w}\|_\infty = \max_k \widetilde{w}_k \leq (1 + \frac{1}{H}) \|w\|_\infty. \tag{48}$$

Moreover,

$$\|\widetilde{w}\|_1 = \sum_{k=1}^{K} \sum_{j=1}^{K} \frac{w_j}{\check{n}_h^j} \sum_{i=1}^{\check{n}_h^j} \mathbf{1}\{k = \check{\ell}_{h,i}^j\} = \sum_{j=1}^{K} \frac{w_j}{\check{n}_h^j} \sum_{k=1}^{K} \sum_{i=1}^{\check{n}_h^j} \mathbf{1}\{k = \check{\ell}_{h,i}^j\} = \sum_{j=1}^{K} w_j = \|w\|_1. \tag{49}$$

Combining (46), (47), (48) and (49), we have

$$\sum_{k=1}^{K} w_k \Delta_h^k \leq \sum_{k=1}^{K} w_k \zeta_h^k + 2\sum_{k=1}^{K} w_k \gamma_h^k + \sum_{k=1}^{K} \widetilde{w}_k \Delta_{h+1}^k + H\sum_{k=1}^{K} w_k \mathbf{1}\{n_h^k = 0\}$$

$$\leq \sum_{k=1}^{K} w_k \zeta_h^k + 2\sum_{k=1}^{K} w_k \gamma_h^k + \sum_{k=1}^{K} \widetilde{w}_k \Delta_{h+1}^k + SABH^2 \|w\|_\infty. \tag{50}$$

By Azuma-Hoeffding's inequality, with probability at least $1 - H\delta$, it holds that for any $h \in [H]$

$$\sum_{k=1}^{K} w_k \zeta_h^k \leq \sqrt{2}H\iota \sqrt{\sum_{k=1}^{K} w_k} \leq \sqrt{2}H\iota \|w\|_\infty. \tag{51}$$

We now bound the second term of (50). Define $\Xi(s, a, b, j) = \sum_{k=1}^{K} w_k \mathbf{1}\{\check{n}_h^k = e_j, (s_h^k, a_h^k, b_h^k) = (s, a, b)\}$ and $\Xi(s, a, b) = \sum_{j \geq 1} \Xi(s, a, b, j)$. Similar to (48) and (49), we then have $\Xi(s, a, b, j) \leq \|w\|_\infty (1 + \frac{1}{H})e_j$ and $\sum_{s,a} \Xi(s, a, b) = \sum_k w_k$. Then

$$\sum_k w_k \gamma_h^k = \sum_k 2\sqrt{H^2\iota} w_k \sqrt{\frac{1}{\check{n}_h^k}}$$

$$= 2\sqrt{H^2\iota} \sum_{s,a,b,j} \sqrt{\frac{1}{e_j} \sum_{j=1}^{K} w_k \mathbf{1}\{\check{n}_h^k = e_j, (s_h^k, a_h^k, b_h^k) = (s,a,b)\}}$$

$$= 2\sqrt{H^2\iota} \sum_{s,a,b} \sum_{j\geq 1} \Xi(s,a,b,j)\sqrt{\frac{1}{e_j}}.$$

Fix $(s,a,b)$ and consider $\sum_{j\geq 1} \Xi(s,a,b,j)\sqrt{\frac{1}{e_j}}$. Note that $\sqrt{\frac{1}{e_j}}$ is decreasing in $j$. Given $\sum_{j\geq 1} \Xi(s,a,b,j) = \Xi(s,a,b)$ is fixed, rearranging the inequality gives

$$\sum_{j\geq 1} \Xi(s,a,b,j)\sqrt{\frac{1}{e_j}} \leq \sum_{j\geq 1} \sqrt{\frac{1}{e_j}} \|w\|_\infty (1+\frac{1}{H})e_j \mathbf{1}\left\{\sum_{i=1}^{j-1} \|w\|_\infty (1+\frac{1}{H})e_i \leq \Xi(s,a,b)\right\}$$

$$= \|w\|_\infty (1+\frac{1}{H}) \sum_j \sqrt{e_j} \mathbf{1}\left\{\sum_{i=1}^{j-1} \|w\|_\infty e_i \leq \Xi(s,a,b)\right\}$$

$$\leq 10(1+\frac{1}{H})\sqrt{\|w\|_\infty H\Xi(s,a,b)}.$$

Therefore, by Cauchy-Schwartz inequality, we have

$$\sum_{k=1}^{K} w_k \gamma_h^k \leq 2\sqrt{H^2\iota} \sum_{s,a,b} 10(1+\frac{1}{H})\sqrt{\|w\|_\infty H}\sqrt{\Xi(s,a,b)}$$

$$\leq 20\sqrt{H^2\iota}(1+\frac{1}{H})\sqrt{\|w\|_\infty SABH \|w\|_1}. \tag{52}$$

Combining (50), (51) and (52), we have

$$\sum_{k=1}^{K} w_k \Delta_h^k \leq \sum_{k=1}^{K} \widetilde{w}_k \Delta_{h+1}^k + (\sqrt{2}H\iota + SABH^2) \|w\|_\infty + 80H\sqrt{\|w\|_\infty SABH \|w\|_1 \iota}. \tag{53}$$

We expand the expression by iterating over step $h+1, \cdots, H$,

$$\sum_{k=1}^{K} w_k \Delta_h^k \leq (1+\frac{1}{H})^H \cdot H \cdot \left((\sqrt{2}H\iota + SABH^2)\|w\|_\infty + 80H\sqrt{\|w\|_\infty SABH \|w\|_1 \iota}\right)$$

$$\leq 6(H^2\iota + SABH^3)\|w\|_\infty + 240H^{\frac{5}{2}}\sqrt{\|w\|_\infty SAB \|w\|_1 \iota}.$$

Now we set $w_k = \mathbf{1}\{\Delta_h^k \geq \epsilon\}$, and obtain

$$\sum_{k=1}^{K} \mathbf{1}\{\Delta_h^k \geq \epsilon\}\Delta_h^k \leq 6(H^2\iota + SABH^3)\|w\|_\infty + 240H^{\frac{5}{2}}\sqrt{\|w\|_\infty SAB\iota \sum_{k=1}^{K} \mathbf{1}\{\Delta_h^k \geq \epsilon\}}.$$

Note that $\|w\|_\infty$ is either 0 or 1. If $\|w\|_\infty = 0$, the claim obviously holds. In the case when $\|w\|_\infty = 1$, solving the following quadratic equation (ignoring coefficients) with respect to $\left(\sum_{k=1}^{K} \mathbf{1}\{\Delta_h^k \geq \epsilon\}\right)^{1/2}$ gives the desired result

$$\epsilon\left(\sum_{k=1}^{K} \mathbf{1}\{\Delta_h^k \geq \epsilon\}\right) - H^{5/2}(SAB\iota)^{1/2}\left(\sum_{k=1}^{K} \mathbf{1}\{\Delta_h^k \geq \epsilon\}\right)^{1/2} - (SABH^3 + H^2\iota) \leq 0.$$

## H    PROOF OF LEMMA E.5 (STEP IV)

The entire proof is conditioned on the successful events of Lemma E.1 and Lemma E.2, which occur with probability at least $1 - O(H^2T^4)\delta$.

By the definition of $\Lambda_{h+1}^k$, we have

$$
\sum_{h=1}^{H}\sum_{k=1}^{K}(1+\frac{1}{H})^{h-1}\Lambda_{h+1}^k = \underbrace{\sum_{h=1}^{H}\sum_{k=1}^{K}(1+\frac{1}{H})^{h-1}\psi_{h+1}^k}_{T_1} + \underbrace{\sum_{h=1}^{H}\sum_{k=1}^{K}(1+\frac{1}{H})^{h-1}\xi_{h+1}^k}_{T_2}
$$

$$
+ 2\underbrace{\sum_{h=1}^{H}\sum_{k=1}^{K}(1+\frac{1}{H})^{h-1}\overline{\beta}_h^k}_{T_3} + 2\underbrace{\sum_{h=1}^{H}\sum_{k=1}^{K}(1+\frac{1}{H})^{h-1}\underline{\beta}_h^k}_{T_4}. \quad (54)
$$

We next bound each of the above four terms in one subsection, and summarize the final result in Appendix H.5.

## H.1 BOUND $T_1$

Recall the definition $\lambda_h^k(s) = \mathbf{1}\left\{n_h^k(s) < N_0\right\}$. Since $\psi$ is always non-negative, we have

$$
\sum_{h=1}^{H}\sum_{k=1}^{K}(1+\frac{1}{H})^{h-1}\psi_{h+1}^k
$$

$$
\leq 3\sum_{h=1}^{H}\sum_{k=1}^{K}\psi_{h+1}^k
$$

$$
= 3\sum_{h=1}^{H}\sum_{k=1}^{K}P_{s_h^k,a_h^k,b_h^k,h}\left(\frac{1}{n_h^k}\sum_{i=1}^{n_h^k}\left(\overline{V}_{h+1}^{\mathrm{ref},\ell_i}-\underline{V}_{h+1}^{\mathrm{ref},\ell_i}\right)-\left(\overline{V}_{h+1}^{\mathrm{REF}}-\underline{V}_{h+1}^{\mathrm{REF}}\right)\right)
$$

$$
\leq 3H\sum_{h=1}^{H}\sum_{k=1}^{K}P_{s_h^k,a_h^k,b_h^k,h}\left(\frac{1}{n_h^k}\sum_{i=1}^{n_h^k}\lambda_{h+1}^{\ell_i}\right)
$$

$$
\leq 3H\sum_{h=1}^{H}\sum_{j=1}^{K}\sum_{k=1}^{K}P_{s_h^k,a_h^k,b_h^k,h}\lambda_{h+1}^j\frac{1}{n_h^k}\sum_{i=1}^{n_h^k}\mathbf{1}\{j=\ell_{h,i}^k\}
$$

$$
\leq 3H\sum_{h=1}^{H}\sum_{j=1}^{K}P_{s_h^j,a_h^j,b_h^j,h}\lambda_{h+1}^j\sum_{k=1}^{K}\frac{1}{n_h^k}\sum_{i=1}^{n_h^k}\mathbf{1}\{j=\ell_{h,i}^k\} \quad (55)
$$

$$
\leq 6(\log T+1)H\sum_{h=1}^{H}\sum_{k=1}^{K}P_{s_h^k,a_h^k,b_h^k,h}\lambda_{h+1}^k \quad (56)
$$

$$
\leq 6(\log T+1)H\left(\sum_{h=1}^{H}\sum_{k=1}^{K}\lambda_{h+1}^k(s_{h+1}^k)+\sum_{h=1}^{H}\sum_{k=1}^{K}\left(P_{s_h^k,a_h^k,b_h^k,h}-\mathbf{1}_{s_{h+1}^k}\right)\lambda_{h+1}^k\right)
$$

$$
\leq 6(\log T+1)H\left(HSN_0+\sum_{h=1}^{H}\sum_{k=1}^{K}\left(P_{s_h^k,a_h^k,b_h^k,h}-\mathbf{1}_{s_{h+1}^k}\right)\lambda_{h+1}^k\right)
$$

$$
\leq 6(\log T+1)H\left(HSN_0+2\sqrt{T\iota}\right), \quad (57)
$$

where (55) follows from the fact that $\frac{1}{n_h^k}\sum_{i=1}^{n_h^k}\mathbf{1}\{j=\ell_{h,i}^k\}\neq 0$ only if $(s_h^k,a_h^k,b_h^k)=(s_h^j,a_h^j,b_h^j)$, (56) follows because

$$
\sum_{k=1}^{K}\frac{1}{n_h^k}\sum_{i=1}^{n_h^k}\mathbf{1}\{j=\ell_{h,i}^k\}\leq\sum_{z:j\leq\sum_{i=1}^{z-1}e_i\leq T}\frac{e_z}{\sum_{i=1}^{z-1}e_i}\leq 2(\log T+1),
$$

and (57) holds with probability at least $1-\delta$ by Azuma's inequality.

To conclude, with probability at least $1 - \delta$, it holds that

$$\sum_{h=1}^{H}\sum_{k=1}^{K}(1+\frac{1}{H})^{h-1}\psi_{h+1}^{k} \le O(\log T) \cdot (H^2 S N_0 + H\sqrt{T\iota}). \tag{58}$$

## H.2 Term $T_2$

We first derive

$$\sum_{h=1}^{H}\sum_{k=1}^{K}(1+\frac{1}{H})^{h-1}\xi_{h+1}^{k}$$

$$=\sum_{h=1}^{H}\sum_{k=1}^{K}\frac{1}{\check{n}_h^k}\sum_{i=1}^{\check{n}_h^k}\left(P_{s_h^k,a_h^k,b_h^k,h}-\mathbf{1}_{s_{h+1}^{\check{\ell}_i}}\right)\left(\overline{V}_{h+1}^{\check{\ell}_i}-\underline{V}_{h+1}^{\check{\ell}_i}\right)$$

$$=\sum_{h=1}^{H}\sum_{k=1}^{K}\sum_{j=1}^{K}(1+\frac{1}{H})^{h-1}\frac{1}{\check{n}_h^k}\sum_{i=1}^{\check{n}_h^k}\left(P_{s_h^k,a_h^k,b_h^k,h}-\mathbf{1}_{s_{h+1}^{j}}\right)\left(\overline{V}_{h+1}^{j}-\underline{V}_{h+1}^{j}\right)\mathbf{1}\{\check{\ell}_{h,i}^k=j\}.$$

Note that $\check{\ell}_{h,i}^k = j$ if and only if $(s_h^k, a_h^k, b_h^k) = (s_h^j, a_h^j, b_h^j)$. Therefore,

$$\sum_{h=1}^{H}\sum_{k=1}^{K}(1+\frac{1}{H})^{h-1}\xi_{h+1}^{k}$$

$$\le \sum_{h=1}^{H}\sum_{j=1}^{K}(1+\frac{1}{H})^{h-1}\left(P_{s_h^j,a_h^j,b_h^j,h}-\mathbf{1}_{s_{h+1}^{j}}\right)\left(\overline{V}_{h+1}^{j}-\underline{V}_{h+1}^{j}\right)\sum_{k=1}^{K}\frac{1}{\check{n}_h^k}\sum_{i=1}^{\check{n}_h^k}\mathbf{1}\{\check{\ell}_{h,i}^k=j\}$$

$$=\sum_{h=1}^{H}\sum_{k=1}^{K}\theta_{h+1}^{j}\left(P_{s_h^j,a_h^j,b_h^j,h}-\mathbf{1}_{s_{h+1}^{j}}\right)\left(\overline{V}_{h+1}^{j}-\underline{V}_{h+1}^{j}\right),$$

where in the last equation we define $\theta_{h+1}^{j} = (1+\frac{1}{H})^{h-1}\sum_{k=1}^{K}\frac{1}{\check{n}_h^k}\sum_{i=1}^{\check{n}_h^k}\mathbf{1}\{\check{\ell}_{h,i}^k=j\}$.

For $(j,h) \in [K] \times [H]$, let $x_h^j$ be the number of elements in current state with respect to $(s_h^j, a_h^j, b_h^j, h)$ and $\tilde{\theta}_{h+1}^{j} := (1+\frac{1}{H})^{h-1}\frac{\lfloor(1+\frac{1}{H})x_h^j\rfloor}{x_h^j} \le 3$. Define $\mathcal{K} = \{(k,h) : \theta_{h+1}^k = \tilde{\theta}_{h+1}^k\}$. Note that if $k$ is before the second last stage of the tuple $(s_h^k, a_h^k, b_h^k, h)$, then we have that $\theta_{h+1}^k = \tilde{\theta}_{h+1}^k$ and $(k,h) \in \mathcal{K}$. Given $(k,h) \in \mathcal{K}$, $s_{h+1}^k$ follows the transition $P_{s_h^k,a_h^k,b_h^k,h}$.

Let $\mathcal{K}_h^\perp(s,a,b) = \{k : (s_h^k, a_h^k, b_h^k) = (s,a,b), \text{where } k \text{ is in the second last stage of } (s,a,b,h)\}$. Note that for different $j, k$, if $(s_h^k, a_h^k, b_h^k) = (s_h^j, a_h^j, b_h^j)$ and $j, k$ are in the same stage of $(s_h^k, a_h^k, b_h^k, h)$, then $\theta_{h+1}^k = \theta_{h+1}^j$ and $\tilde{\theta}_{h+1}^k = \tilde{\theta}_{h+1}^j$. Denote $\theta_{h+1}$ and $\tilde{\theta}_{h+1}$ as $\theta_{h+1}(s,a,b)$ and $\tilde{\theta}_{h+1}(s,a,b)$ respectively for some $k \in \mathcal{K}_h^\perp(s,a,b)$.

We have

$$\sum_{h=1}^{H}\sum_{k=1}^{K}(1+\frac{1}{H})^{h-1}\xi_{h+1}^{k}$$

$$=\sum_{(k,h)}\tilde{\theta}_{h+1}^{k}\left(P_{s_h^j,a_h^j,b_h^j,h}-\mathbf{1}_{s_{h+1}^{j}}\right)\left(\overline{V}_{h+1}^{j}-\underline{V}_{h+1}^{j}\right)$$

$$+\sum_{(k,h)}(\theta_{h+1}^{k}-\tilde{\theta}_{h+1}^{k})\left(P_{s_h^j,a_h^j,b_h^j,h}-\mathbf{1}_{s_{h+1}^{j}}\right)\left(\overline{V}_{h+1}^{j}-\underline{V}_{h+1}^{j}\right)$$

$$=\sum_{(k,h)}\tilde{\theta}_{h+1}^{k}\left(P_{s_h^j,a_h^j,b_h^j,h}-\mathbf{1}_{s_{h+1}^{j}}\right)\left(\overline{V}_{h+1}^{j}-\underline{V}_{h+1}^{j}\right)$$

$$+ \sum_{(k,h)\in\overline{\mathcal{K}}} (\theta_{h+1}^k - \tilde{\theta}_{h+1}^k)\left(P_{s_h^j,a_h^j,b_h^j,h} - \mathbf{1}_{s_{h+1}^j}\right)\left(\overline{V}_{h+1}^j - \underline{V}_{h+1}^j\right). \tag{59}$$

Since $\tilde{\theta}_{h+1}^k$ is independent of $s_{h+1}^k$, by Azuma's inequality, with probability at least $1-\delta$, it holds that

$$\sum_{(k,h)} \tilde{\theta}_{h+1}^k \left(P_{s_h^k,a_h^k,b_h^k,h} - \mathbf{1}_{s_{h+1}^k}\right)\left(\overline{V}_{h+1}^k - \underline{V}_{h+1}^k\right) \le 6\sqrt{TH^2\iota}. \tag{60}$$

Moreover, we have

$$\sum_{(k,h)\in\overline{\mathcal{K}}} (\theta_{h+1}^k - \tilde{\theta}_{h+1}^k)\left(P_{s_h^k,a_h^k,b_h^k,h} - \mathbf{1}_{s_{h+1}^k}\right)\left(\overline{V}_{h+1}^k - \underline{V}_{h+1}^k\right)$$

$$= \sum_{s,a,b,h} \sum_{(k,h)\in\overline{\mathcal{K}}} \mathbf{1}\{(s_h^k,a_h^k,b_h^k) = (s,a,b)\}(\theta_{h+1}^k - \tilde{\theta}_{h+1}^k)\left(P_{s_h^k,a_h^k,b_h^k,h} - \mathbf{1}_{s_{h+1}^k}\right)\left(\overline{V}_{h+1}^k - \underline{V}_{h+1}^k\right)$$

$$= \sum_{s,a,b,h} (\theta_{h+1}(s,a,b) - \tilde{\theta}_{h+1}(s,a)) \sum_{(k,h)\in\mathcal{K}_h^\perp(s,a)} (\theta_{h+1}^k - \tilde{\theta}_{h+1}^k)\left(P_{s_h^k,a_h^k,b_h^k,h} - \mathbf{1}_{s_{h+1}^k}\right)\left(\overline{V}_{h+1}^k - \underline{V}_{h+1}^k\right)$$

$$\le \sum_{s,a,b,h} \mathcal{O}(H)\sqrt{|\mathcal{K}_h^\perp(s,a,b)|\iota} \tag{61}$$

$$\le \sum_{s,a,b,h} \mathcal{O}(H)\sqrt{\check{N}_h^{K+1}(s,a,b)\iota} \tag{62}$$

$$\le \mathcal{O}(H)\sqrt{SABH\iota \sum_{s,a,b,h} \check{N}_h^{K+1}(s,a,b)} \tag{62}$$

$$\le \mathcal{O}(H)\sqrt{SABH\iota(T/H)}, \tag{63}$$

where (61) holds with probability at least $1 - T\delta$ by Azuma's inequality and a union bound over all steps in $\overline{\mathcal{K}}$, (62) follows from Cauchy-Schwartz inequality, and (63) follows from the fact that the length of the last two stages for each $(s,a,b,h)$ tuple is only $O(1/H)$ fraction of the total number of visits.

Combining (59), (60) and (63), we obtain that with probability at least $1 - (T+1)\delta$, it holds that

$$\sum_{h=1}^{H}\sum_{k=1}^{K}(1 + \frac{1}{H})^{h-1}\xi_{h+1}^k \le \mathcal{O}(\sqrt{H^2SABT\iota}). \tag{64}$$

### H.3 Term $T_3$

Note that

$$\sum_{h=1}^{H}\sum_{k=1}^{K}(1 + \frac{1}{H})^{h-1}\overline{\beta}_h^k$$

$$\le 3\sum_{h=1}^{H}\sum_{k=1}^{K}\left(c_1\sqrt{\frac{\overline{\nu}_h^{\mathrm{ref},k}}{n_h^k}\iota} + c_2\sqrt{\frac{\check{\nu}_h^k}{\check{n}_h^k}\iota} + c_3\left(\frac{H\iota}{n_h^k} + \frac{H\iota}{\check{n}_h^k} + \frac{H\iota^{\frac{3}{4}}}{(n_h^k)^{\frac{3}{4}}} + \frac{H\iota^{\frac{3}{4}}}{(\check{n}_h^k)^{\frac{3}{4}}}\right)\right)$$

$$\le O\left(\sum_{h=1}^{H}\sum_{k=1}^{K}\left(\sqrt{\frac{\overline{\nu}_h^{\mathrm{ref},k}}{n_h^k}\iota} + \sqrt{\frac{\check{\nu}_h^k}{\check{n}_h^k}\iota}\right)\right) + O(SABH^3\iota\log T + (SAB\iota)^{\frac{3}{4}}H^{\frac{5}{2}}T^{\frac{1}{4}}), \tag{65}$$

where (65) follows from Lemma J.3 with $\alpha = \frac{3}{4}$ and $\alpha = 1$.

**Step i:** We bound $\sum_{h=1}^{H}\sum_{k=1}^{K}\sqrt{\frac{\overline{\nu}_h^{\mathrm{ref},k}}{n_h^k}\iota}$. We begin with the following technical lemmas.

**Lemma H.1.** *With probability at least $1 - 2T\delta$, it holds that for all $s, a, b, h, k$,*

$$\overline{Q}_h^k(s, a, b) \leq Q_h^{\pi^k}(s, a, b) + (H - h)\left(\beta + \frac{HSN_0}{\check{n}_h^k}\right),$$

$$\underline{Q}_h^k(s, a, b) \geq Q_h^{\pi^k}(s, a, b) - (H - h)\left(\beta + \frac{HSN_0}{\check{n}_h^k}\right),$$

$$\overline{V}_h^k(s) \leq V_h^{\pi^k}(s) + (H - h)\left(\beta + \frac{HSN_0}{\check{n}_h^k}\right),$$

$$\underline{V}_h^k(s) \geq V_h^{\pi^k}(s) - (H - h)\left(\beta + \frac{HSN_0}{\check{n}_h^k}\right).$$

The proof is provided in Appendix H.3.1.

**Lemma H.2.** *Conditioned on the successful event of Lemma H.1, with probability at least $1 - 4\delta$, it holds that*

$$\overline{\nu}_h^{\mathrm{ref},k} - \mathbb{V}(P_{s_h^k,a_h^k,b_h^k,h}, V_{h+1}^{\pi_k}) \leq 4H\beta + \frac{12H^2\beta + 18H^3SN_0}{n_h^k} + 20H^2\sqrt{\frac{\iota}{n_h^k}}.$$

The proof is provided in Appendix H.3.2.

**Lemma H.3** (Lemma C.5 in [17]). *With probability at least $1 - \delta$, it holds that*

$$\mathbb{V}(P_{s_h^k,a_h^k,b_h^k,h}, V_{h+1}^{\pi^k}) \leq \mathcal{O}(HT + H^3\iota).$$

Combining Lemma H.2, Lemma H.3 and Lemma J.3 (see Appendix J), we have

$$\sum_{h=1}^H \sum_{k=1}^K \sqrt{\frac{\overline{\nu}_h^{\mathrm{ref},k}}{n_h^k}\iota}$$

$$\leq \sum_{h=1}^H \sum_{k=1}^K \sqrt{\frac{\mathbb{V}(P_{s_h^k,a_h^k,b_h^k,h}, V_{h+1}^{\pi^k})}{n_h^k}\iota}$$

$$+ \sum_{h=1}^H \sum_{k=1}^K \sqrt{\left(\frac{4H\beta}{n_h^k} + \frac{12H^2\beta + 18H^3SN_0}{(n_h^k)^2} + 20H^2\frac{\iota^{\frac{1}{2}}}{(n_h^k)^{\frac{3}{2}}}\right)\iota}$$

$$\leq O\left(\sum_{s,a,b,h} \sqrt{N_h^{K+1}(s,a,b)\mathbb{V}(P_{s,a,b,h}, V_{h+1}^{\pi^k})\iota}\right)$$

$$+ O\left(\sum_{s,a,b,h} \sqrt{N_h^{K+1}(s,a,b)H\beta\iota} + (S^{\frac{3}{2}}ABH^{\frac{5}{2}}N_0^{\frac{1}{2}} + SABH^2\beta^{\frac{1}{2}})\iota^{\frac{1}{2}}\log T + (SAB\iota)^{\frac{3}{4}}H^{\frac{7}{4}}T^{\frac{1}{4}}\right)$$

$$\leq O\left(\sqrt{SABH^2T\iota} + \sqrt{SABH^2\beta T\iota} + (S^{\frac{3}{2}}ABH^{\frac{5}{2}}N_0^{\frac{1}{2}} + SABH^2\beta^{\frac{1}{2}})\iota^{\frac{1}{2}}\log T + (SAB\iota)^{\frac{3}{4}}H^{\frac{7}{4}}T^{\frac{1}{4}}\right).$$
(66)

**Step ii:** We bound $\sum_{h=1}^H \sum_{k=1}^K \sqrt{\frac{\check{\nu}_h^k}{\check{n}_h^k}\iota}$.

By Lemma E.1, Lemma E.2 and Corollary E.3, we have

$$\check{\nu}_h^k \leq \frac{1}{\check{n}_h^k} \sum_{i=1}^{\check{n}_h^k} \left(\overline{V}_{h+1}^{\check{\ell}_i} - \overline{V}_{h+1}^{\mathrm{ref},\check{\ell}_i}\right)^2 (s_{h+1}^{\check{\ell}_i})$$

$$\leq \frac{1}{\check{n}_h^k} \sum_{i=1}^{\check{n}_h^k} \left(\overline{V}_{h+1}^{\check{\ell}_i} - \underline{V}_{h+1}^{\check{\ell}_i}\right)^2 (s_{h+1}^{\check{\ell}_i}) + \frac{1}{\check{n}_h^k} \sum_{i=1}^{\check{n}_h^k} \left(\overline{V}_{h+1}^{\mathrm{ref},\check{\ell}_i} - \underline{V}_{h+1}^{\mathrm{ref},\check{\ell}_i}\right)^2 (s_{h+1}^{\check{\ell}_i})$$

$$\leq \frac{2}{\check{n}_h^k} H^2 S N_0 + 2\beta^2.$$

Combining the above inequality with Lemma J.3, we obtain

$$\sum_{h=1}^{H} \sum_{k=1}^{K} \sqrt{\frac{\check{\nu}_h^k \iota}{\check{n}_h^k}} \leq \mathcal{O}\left(\sqrt{SABH^3\beta^2 T\iota} + SABH^3\sqrt{SN_0\iota}\log T\right). \tag{67}$$

Combining (65), (66) and (67), we obtain that with probability at least $1 - O(T)\delta$, it holds that

$$\sum_{h=1}^{H} \sum_{k=1}^{K} (1 + \frac{1}{H})^{h-1} \overline{\beta}_h^k \leq O\big(\sqrt{SABH^2T\iota} + \sqrt{SABH^2\beta T\iota} + \sqrt{SABH^3\beta^2 T\iota}$$

$$+ S^{\frac{3}{2}} ABH^3 N_0^{\frac{1}{2}} \iota \log T + SABH^2 \beta^{\frac{1}{2}} \iota^{\frac{1}{2}} \log T + (SAB\iota)^{\frac{3}{4}} H^{\frac{5}{2}} T^{\frac{1}{4}}\big). \tag{68}$$

### H.3.1 PROOF OF LEMMA H.1

Fix an episode $k$. The proof is based on induction over $h = H, H-1, \ldots, 1$. Note first that the claim clearly holds for $h = H$. Assume the inequalities hold at step $h+1$.

By the update rule of the action-value function, we have

$$\overline{Q}_h^k(s,a,b) \leq r_h(s,a,b) + \frac{1}{\check{n}} \sum_{i=1}^{\check{n}} \overline{V}_{h+1}^{\check{\ell}_i}(s_{h+1}^{\check{\ell}_i}) + \gamma$$

$$= r_h(s,a,b) + \frac{1}{\check{n}} \sum_{i=1}^{\check{n}} \overline{V}_{h+1}^k(s_{h+1}^{\check{\ell}_i}) + \gamma + \frac{1}{\check{n}} \sum_{i=1}^{\check{n}} \left(\overline{V}_{h+1}^{\check{\ell}_i}(s_{h+1}^{\check{\ell}_i}) - \overline{V}_{h+1}^k(s_{h+1}^{\check{\ell}_i})\right)$$

$$\leq r_h(s,a,b) + P_{s,a,b,h}\overline{V}_{h+1}^k + \frac{1}{\check{n}} \sum_{i=1}^{\check{n}} \left(\overline{V}_{h+1}^{\check{\ell}_i}(s_{h+1}^{\check{\ell}_i}) - \overline{V}_{h+1}^k(s_{h+1}^{\check{\ell}_i})\right) \tag{69}$$

$$\leq r_h(s,a,b) + P_{s,a,b,h}V_{h+1}^{\pi^k} + (H-h+1)\left(\beta + \frac{HSN_0}{\check{n}}\right)$$

$$+ \frac{1}{\check{n}} \sum_{i=1}^{\check{n}} \left(\overline{V}_{h+1}^{\check{\ell}_i}(s_{h+1}^{\check{\ell}_i}) - \overline{V}_{h+1}^k(s_{h+1}^{\check{\ell}_i})\right) \tag{70}$$

$$\leq Q_h^{\pi^k} + (H-h+1)\left(\beta + \frac{HSN_0}{\check{n}}\right) + \frac{1}{\check{n}} \sum_{i=1}^{\check{n}} \left(\overline{V}_{h+1}^{\check{\ell}_i}(s_{h+1}^{\check{\ell}_i}) - \underline{V}_{h+1}^{\check{\ell}_i}(s_{h+1}^{\check{\ell}_i})\right) \tag{71}$$

$$\leq Q_h^{\pi^k}(s,a,b) + (H-h+1)\left(\beta + \frac{HSN_0}{\check{n}}\right) + \frac{1}{\check{n}} \sum_{i=1}^{\check{n}} (H\lambda_{h+1}^{\check{\ell}_i} + \beta)$$

$$\leq Q_h^{\pi^k}(s,a,b) + (H-h)\left(\beta + \frac{HSN_0}{\check{n}}\right), \tag{72}$$

where (69) holds with probability at least $1-\delta$ by Azuma's inequality, (70) follows from the induction hypothesis, and (71) follows from Lemma E.1.

Moreover, by the update rule of the value function, we have

$$\overline{V}_h^k(s) = \mathbb{E}_{(a,b)\sim\pi_k} \overline{Q}_h^k(s,a,b)$$

$$\leq \mathbb{E}_{(a,b)\sim\pi_k} Q_h^{\pi^k}(s,a,b) + (H-h)\left(\beta + \frac{HSN_0}{\check{n}}\right)$$

$$\leq V_h^{\pi^k}(s) + (H-h)\left(\beta + \frac{HSN_0}{\check{n}}\right).$$

The other direction for the pessimistic (action-)value function can be proved similarly. Finally, taking the union bound over all steps gives the desired result.

### H.3.2 PROOF OF LEMMA H.2

We first provide bound on $\overline{\nu}_h^{\text{ref},k} - \mathbb{V}(P_{s_h^k,a_h^k,b_h^k,h}, \overline{V}_{h+1}^{\text{ref},\ell_i})$. Recall (25) that

$$\overline{\nu}^{\text{ref}} - \frac{1}{n_h^k} \sum_{i=1}^{n_h^k} \mathbb{V}(P_{s_h^k,a_h^k,b_h^k,h}, \overline{V}_{h+1}^{\text{ref},\ell_i}) = -\frac{1}{n_h^k}(\chi_6 + \chi_7 + \chi_8),$$

where

$$\chi_6 = \sum_{i=1}^{n_h^k} \left( (P_{s_h^k,a_h^k,b_h^k,h}(\overline{V}_{h+1}^{\text{ref},\ell_i})^2 - (\overline{V}_{h+1}^{\text{ref},\ell_i}(s_{h+1}^{\ell_i}))^2 \right),$$

$$\chi_7 = \frac{1}{n_h^k} \left( \sum_{i=1}^{n_h^k} \overline{V}_{h+1}^{\text{ref},\ell_i}(s_{h+1}^{\ell_i}) \right)^2 - \frac{1}{n_h^k} \left( \sum_{i=1}^{n_h^k} P_{s_h^k,a_h^k,b_h^k,h} \overline{V}_{h+1}^{\text{ref},\ell_i} \right)^2,$$

$$\chi_8 = \frac{1}{n_h^k} \left( \sum_{i=1}^{n_h^k} P_{s_h^k,a_h^k,b_h^k,h} \overline{V}_{h+1}^{\text{ref},\ell_i} \right)^2 - \sum_{i=1}^{n_h^k} (P_{s_h^k,a_h^k,b_h^k,h} \overline{V}_{h+1}^{\text{ref},\ell_i})^2.$$

By Azuma's inequality, with probability at least $1 - 2\delta$, it holds that

$$|\chi_6| \le H^2\sqrt{2n_h^k\iota}, \quad |\chi_7| \le 2H^2\sqrt{2n_h^k\iota}.$$

Moreover, we have

$$-\chi_8 = \sum_{i=1}^{n_h^k} \left( P_{s_h^k,a_h^k,b_h^k,h} \overline{V}_{h+1}^{\text{ref},\ell_i} \right)^2 - \frac{1}{n_h^k} \left( \sum_{i=1}^{n_h^k} P_{s_h^k,a_h^k,b_h^k,h} \overline{V}_{h+1}^{\text{ref},\ell_i} \right)^2$$

$$\le \sum_{i=1}^{n_h^k} \left( P_{s_h^k,a_h^k,b_h^k,h} \overline{V}_{h+1}^{\text{ref},\ell_i} \right)^2 - \frac{1}{n_h^k} \left( \sum_{i=1}^{n_h^k} P_{s_h^k,a_h^k,b_h^k,h} \overline{V}_{h+1}^{\text{REF}} \right)^2 \qquad (73)$$

$$= \sum_{i=1}^{n_h^k} \left( \left( P_{s_h^k,a_h^k,b_h^k,h} \overline{V}_{h+1}^{\text{ref},\ell_i} \right)^2 - \left( P_{s_h^k,a_h^k,b_h^k,h} \overline{V}_{h+1}^{\text{REF}} \right)^2 \right)$$

$$\le 2H^2 \sum_{i=1}^{n_h^k} P_{s_h^k,a_h^k,b_h^k,h} \lambda_{h+1}^{\ell_i}$$

$$= 2H^2 \left( \sum_{i=1}^{n_h^k} \lambda_{h+1}^{\ell_i}(s_{h+1}^{\ell_i}) + \sum_{i=1}^{n_h^k} (P_{s_h^k,a_h^k,b_h^k,h} - \mathbf{1}_{s_{h+1}^{\ell_i}})\lambda_{h+1}^{\ell_i} \right)$$

$$\le 2H^2 S N_0 + 2H^2\sqrt{2n_h^k\iota}, \qquad (74)$$

where (73) follows from the fact that $\overline{V}_{h+1}^{\text{ref},k} \ge \overline{V}_{h+1}^{\text{REF}}$ for any $k, h$, and (74) holds with probability at least $1 - \delta$ by Azuma's inequality.

We have

$$\overline{\nu}_h^{\text{ref},k} - \frac{1}{n_h^k} \sum_{i=1}^{n_h^k} \mathbb{V}(P_{s_h^k,a_h^k,b_h^k,h}, \overline{V}_{h+1}^{\text{ref},\ell_i}) \le \frac{2H^2 S N_0}{n_h^k} + 8H^2\sqrt{\frac{\iota}{n_h^k}}. \qquad (75)$$

Therefore,

$$\overline{\nu}_h^{\text{ref},k} - \mathbb{V}(P_{s_h^k,a_h^k,b_h^k,h}, V_{h+1}^{\pi^k})$$

$$= \frac{1}{n_h^k} \sum_{i=1}^{n_h^k} \left( \mathbb{V}(P_{s_h^k,a_h^k,b_h^k,h}, \overline{V}_{h+1}^{\text{ref},\ell_i}) - \mathbb{V}(P_{s_h^k,a_h^k,b_h^k,h}, V_{h+1}^{\pi^k}) \right) + \left( \overline{\nu}_h^{\text{ref},k} - \frac{1}{n_h^k} \sum_{i=1}^{n_h^k} \mathbb{V}(P_{s_h^k,a_h^k,b_h^k,h}, \overline{V}_{h+1}^{\text{ref},\ell_i}) \right)$$

$$\leq \frac{1}{n_h^k} \sum_{i=1}^{n_h^k} \left( \mathbb{V}(P_{s_h^k,a_h^k,b_h^k,h}, \overline{V}_{h+1}^{\mathrm{ref},\ell_i}) - \mathbb{V}(P_{s_h^k,a_h^k,b_h^k,h}, V_{h+1}^{\pi^k}) \right) + \frac{2H^2 SN_0}{n_h^k} + 8H^2 \sqrt{\frac{\iota}{n_h^k}} \tag{76}$$

$$\leq \frac{4H}{n_h^k} \sum_{i=1}^{n_h^k} \left| P_{s_h^k,a_h^k,b_h^k,h}(\overline{V}_{h+1}^{\mathrm{ref},\ell_i} - V_{h+1}^{\pi^k}) \right| + \frac{2H^2 SN_0}{n_h^k} + 8H^2 \sqrt{\frac{\iota}{n_h^k}}$$

$$= \frac{4H}{n_h^k} \sum_{i=1}^{n_h^k} \left| P_{s_h^k,a_h^k,b_h^k,h}(\overline{V}_{h+1}^{\mathrm{ref},\ell_i} - V_{h+1}^{\pi^k} + V_{h+1}^* - V_{h+1}^*) - H\left(\beta + \frac{HSN_0}{\check{n}_h^k}\right) + H\left(\beta + \frac{HSN_0}{\check{n}_h^k}\right) \right|$$
$$+ \frac{2H^2 SN_0}{n_h^k} + 8H^2 \sqrt{\frac{\iota}{n_h^k}}$$

$$\leq \frac{4H}{n_h^k} \sum_{i=1}^{n_h^k} P_{s_h^k,a_h^k,b_h^k,h}(\overline{V}_{h+1}^{\mathrm{ref},\ell_i} - V_{h+1}^*) + \frac{4H}{n_h^k} \sum_{i=1}^{n_h^k} P_{s_h^k,a_h^k,b_h^k,h}\left(V_{h+1}^{\pi^k} - V_{h+1}^* + H\left(\beta + \frac{HSN_0}{\check{n}_h^k}\right)\right)$$
$$+ \frac{4H^2}{n_h^k}\left(\beta + \frac{HSN_0}{\check{n}_h^k}\right) + \frac{2H^2 SN_0}{n_h^k} + 8H^2 \sqrt{\frac{\iota}{n_h^k}} \tag{77}$$

$$\leq \frac{4H}{n_h^k} \sum_{i=1}^{n_h^k} (\overline{V}_{h+1}^{\mathrm{ref},\ell_i} - V_{h+1}^*)(s_{h+1}^{\ell_i}) + \frac{4H}{n_h^k} \sum_{i=1}^{n_h^k} \left( (V_{h+1}^{\pi^k} - V_{h+1}^*)(s_{h+1}^{\ell_i}) + H\left(\beta + \frac{HSN_0}{\check{n}_h^k}\right) \right)$$
$$+ \frac{4H^2}{n_h^k}\left(\beta + \frac{HSN_0}{\check{n}_h^k}\right) + \frac{2H^2 SN_0}{n_h^k} + 20H^2 \sqrt{\frac{\iota}{n_h^k}} \tag{78}$$

$$\leq \left(4H\beta + \frac{4H^2 SN_0}{n_h^k}\right) + \frac{8H^2}{n_h^k}\left(\beta + \frac{HSN_0}{\check{n}_h^k}\right)$$
$$+ \frac{4H^2}{n_h^k}\left(\beta + \frac{HSN_0}{\check{n}_h^k}\right) + \frac{2H^2 SN_0}{n_h^k} + 20H^2 \sqrt{\frac{\iota}{n_h^k}} \tag{79}$$

$$= 4H\beta + \frac{12H^2 \beta}{n_h^k} + 20H^2 \sqrt{\frac{\iota}{n_h^k}} + \frac{6H^2 SN_0}{n_h^k} + \frac{12H^3 SN_0}{n_h^k \check{n}_h^k}$$

$$\leq 4H\beta + \frac{12H^2 \beta + 18H^3 SN_0}{n_h^k} + 20H^2 \sqrt{\frac{\iota}{n_h^k}},$$

where (76) follows from (75), (77) follows from Lemma E.1 and Lemma H.1, (78) holds with probability at least $1 - 2\delta$ by Azuma's inequality, and (79) follows from Lemma E.1 and Lemma H.1.

## H.4 Term $T_4$

The proof is similar to that for the term $\sum_{h=1}^{H} \sum_{k=1}^{K} (1 + \frac{1}{H})^{h-1} \overline{\beta}_h^k$. In the following, we will present the key steps, and provide the proof whenever necessary.

By Lemma J.3, we have

$$\sum_{h=1}^{H} \sum_{k=1}^{K} (1 + \frac{1}{H})^{h-1} \overline{\beta}_h^k$$
$$\leq 3 \sum_{h=1}^{H} \sum_{k=1}^{K} \left( c_1 \sqrt{\frac{\overline{\nu}_h^{\mathrm{ref},k}}{n_h^k}} \iota + c_2 \sqrt{\frac{\check{\nu}_h^k}{\check{n}_h^k}} \iota + c_3 \left( \frac{H\iota}{n_h^k} + \frac{H\iota}{\check{n}_h^k} + \frac{H\iota^{\frac{3}{4}}}{(n_h^k)^{\frac{3}{4}}} + \frac{H\iota^{\frac{3}{4}}}{(\check{n}_h^k)^{\frac{3}{4}}} \right) \right)$$
$$\leq O\left( \sum_{h=1}^{H} \sum_{k=1}^{K} \left( \sqrt{\frac{\overline{\nu}_h^{\mathrm{ref},k}}{n_h^k}} \iota + \sqrt{\frac{\check{\nu}_h^k}{\check{n}_h^k}} \iota \right) \right) + O(SABH^3 \iota \log T + (SAB\iota)^{\frac{3}{4}} H^{\frac{5}{2}} T^{\frac{1}{4}}). \tag{80}$$

**Step i:** Bound term $\sum_{h=1}^{H} \sum_{k=1}^{K} \sqrt{\frac{\nu_h^{\mathrm{ref},k}}{n_h^k}} \iota$.

**Lemma H.4.** *Conditioned on the successful event of Lemma H.1, with probability at least $1 - 4\delta$, it holds that*

$$\nu_h^{\mathrm{ref},k} - \mathbb{V}(P_{s_h^k,a_h^k,b_h^k,h}, V_{h+1}^{\pi_k}) \leq 4H\beta + \frac{12H^2\beta + 18H^3SN_0}{n_h^k} + 20H^2\sqrt{\frac{\iota}{n_h^k}}.$$

The proof is provided in Appendix H.4.1.

Combining Lemma H.3, Lemma H.4 and Lemma J.3, we have

$$\sum_{h=1}^{H}\sum_{k=1}^{K}\sqrt{\frac{\nu_h^{\mathrm{ref},k}}{n_h^k}\iota}$$
$$\leq O\left(\sqrt{SABH^2T\iota} + \sqrt{SABH^2\beta T\iota} + (S^{\frac{3}{2}}ABH^{\frac{5}{2}}N_0^{\frac{1}{2}} + SABH^2\beta^{\frac{1}{2}})\iota^{\frac{1}{2}}\log T + (SAB\iota)^{\frac{3}{4}}H^{\frac{7}{4}}T^{\frac{1}{4}}\right).$$

(81)

**Step ii:** Bound $\sum_{h=1}^{H}\sum_{k=1}^{K}\sqrt{\frac{\check{\nu}_h^k}{\check{n}_h^k}\iota}$. By Lemma E.1, Lemma E.2 and Corollary E.3, we have

$$\check{\nu}_h^k \leq \frac{1}{\check{n}_h^k}\sum_{i=1}^{\check{n}_h^k}\left(\underline{V}_{h+1}^{\check{\ell}_i} - \underline{V}_{h+1}^{\mathrm{ref},\check{\ell}_i}\right)^2(s_{h+1}^{\check{\ell}_i})$$
$$\leq \frac{1}{\check{n}_h^k}\sum_{i=1}^{\check{n}_h^k}\left(\overline{V}_{h+1}^{\check{\ell}_i} - \underline{V}_{h+1}^{\check{\ell}_i}\right)^2(s_{h+1}^{\check{\ell}_i}) + \frac{1}{\check{n}_h^k}\sum_{i=1}^{\check{n}_h^k}\left(\overline{V}_{h+1}^{\mathrm{ref},\check{\ell}_i} - \underline{V}_{h+1}^{\mathrm{ref},\check{\ell}_i}\right)^2(s_{h+1}^{\check{\ell}_i})$$
$$\leq \frac{2}{\check{n}_h^k}H^2SN_0 + 2\beta^2.$$

Combining the above inequality with Lemma J.3, we obtain

$$\sum_{h=1}^{H}\sum_{k=1}^{K}\sqrt{\frac{\check{\nu}_h^k\iota}{\check{n}_h^k}} \leq \mathcal{O}\left(\sqrt{SABH^3\beta^2T\iota} + SABH^3\sqrt{SN_0\iota}\log T\right).$$

(82)

Therefore, combining (80), (81) and (82) gives that with probability at least $1 - O(T)\delta$, it holds that

$$\sum_{h=1}^{H}\sum_{k=1}^{K}(1 + \frac{1}{H})^{h-1}\overline{\beta}_h^k \leq O\big(\sqrt{SABH^2T\iota} + \sqrt{SABH^2\beta T\iota} + \sqrt{SABH^3\beta^2T\iota}$$
$$+ S^{\frac{3}{2}}ABH^3N_0^{\frac{1}{2}}\iota\log T + SABH^2\beta^{\frac{1}{2}}\iota^{\frac{1}{2}}\log T + (SAB\iota)^{\frac{3}{4}}H^{\frac{5}{2}}T^{\frac{1}{4}}\big).$$

(83)

### H.4.1 PROOF OF LEMMA H.4

Recall (39) that

$$\nu^{\mathrm{ref}} - \frac{1}{n_h^k}\sum_{i=1}^{n_h^k}\mathbb{V}(P_{s_h^k,a_h^k,b_h^k,h}, \underline{V}_{h+1}^{\mathrm{ref},\ell_i}) = -\frac{1}{n_h^k}(\underline{\chi}_6 + \underline{\chi}_7 + \underline{\chi}_8),$$

where

$$\underline{\chi}_6 = \sum_{i=1}^{n_h^k}\left((P_{s_h^k,a_h^k,b_h^k,h}(\underline{V}_{h+1}^{\mathrm{ref},\ell_i})^2 - (\underline{V}_{h+1}^{\mathrm{ref},\ell_i}(s_{h+1}^{\ell_i}))^2\right),$$

$$\underline{\chi}_7 = \frac{1}{n_h^k}\left(\sum_{i=1}^{n_h^k}\underline{V}_{h+1}^{\mathrm{ref},\ell_i}(s_{h+1}^{\ell_i})\right)^2 - \frac{1}{n_h^k}\left(\sum_{i=1}^{n_h^k}P_{s_h^k,a_h^k,b_h^k,h}\underline{V}_{h+1}^{\mathrm{ref},\ell_i}\right)^2,$$

$$\underline{\chi}_8 = \frac{1}{n_h^k}\left(\sum_{i=1}^{n_h^k}P_{s_h^k,a_h^k,b_h^k,h}\underline{V}_{h+1}^{\mathrm{ref},\ell_i}\right)^2 - \sum_{i=1}^{n_h^k}(P_{s_h^k,a_h^k,b_h^k,h}\underline{V}_{h+1}^{\mathrm{ref},\ell_i})^2.$$

By Azuma's inequality, with probability at least $1 - 2\delta$, it holds that

$$|\underline{\chi}_6| \leq H^2\sqrt{2n_h^k\iota}, \quad |\underline{\chi}_7| \leq 2H^2\sqrt{2n_h^k\iota}.$$

The term $\underline{\chi}_8$ is bounded slightly differently from $\chi_8$ as follows:

$$
-\underline{\chi}_8 = \sum_{i=1}^{n_h^k}\left(P_{s_h^k,a_h^k,b_h^k,h}\underline{V}_{h+1}^{\mathrm{ref},\ell_i}\right)^2 - \frac{1}{n_h^k}\left(\sum_{i=1}^{n_h^k}P_{s_h^k,a_h^k,b_h^k,h}\underline{V}_{h+1}^{\mathrm{ref},\ell_i}\right)^2
$$

$$
\leq \sum_{i=1}^{n_h^k}\left(P_{s_h^k,a_h^k,b_h^k,h}\underline{V}_{h+1}^{\mathrm{REF}}\right)^2 - \frac{1}{n_h^k}\left(\sum_{i=1}^{n_h^k}P_{s_h^k,a_h^k,b_h^k,h}\underline{V}_{h+1}^{\mathrm{ref},\ell_i}\right)^2 \tag{84}
$$

$$
= \frac{1}{n_h^k}\left(\sum_{i=1}^{n_h^k}P_{s_h^k,a_h^k,b_h^k,h}\underline{V}_{h+1}^{\mathrm{REF}}\right)^2 - \frac{1}{n_h^k}\left(\sum_{i=1}^{n_h^k}P_{s_h^k,a_h^k,b_h^k,h}\underline{V}_{h+1}^{\mathrm{ref},\ell_i}\right)^2
$$

$$
\leq 2H^2\sum_{i=1}^{n_h^k}P_{s_h^k,a_h^k,b_h^k,h}\lambda_{h+1}^{\ell_i}
$$

$$
= 2H^2\left(\sum_{i=1}^{n_h^k}\lambda_{h+1}^{\ell_i}(s_{h+1}^{\ell_i}) + \sum_{i=1}^{n_h^k}(P_{s_h^k,a_h^k,b_h^k,h} - \mathbf{1}_{s_{h+1}^{\ell_i}})\lambda_{h+1}^{\ell_i}\right)
$$

$$
\leq 2H^2SN_0 + 2H^2\sqrt{2n_h^k\iota}, \tag{85}
$$

where (84) follows from the fact that $\underline{V}_{h+1}^{\mathrm{ref},k} \leq \underline{V}_{h+1}^{\mathrm{REF}}$ for any $k,h$, and (85) holds with probability at least $1 - \delta$ due to Azuma's inequality. Therefore,

$$
\underline{\nu}_h^{\mathrm{ref},k} - \frac{1}{n_h^k}\sum_{i=1}^{n_h^k}\mathbb{V}(P_{s_h^k,a_h^k,b_h^k,h},\underline{V}_{h+1}^{\mathrm{ref},\ell_i}) \leq \frac{2H^2SN_0}{n_h^k} + 8H^2\sqrt{\frac{\iota}{n_h^k}}. \tag{86}
$$

By a similar argument as in Appendix H.3.2, we can obtain the desired result

$$
\underline{\nu}_h^{\mathrm{ref},k} - \mathbb{V}(P_{s_h^k,a_h^k,b_h^k,h},V_{h+1}^{\pi^k})
$$

$$
= \frac{1}{n_h^k}\sum_{i=1}^{n_h^k}\left(\mathbb{V}(P_{s_h^k,a_h^k,b_h^k,h},\underline{V}_{h+1}^{\mathrm{ref},\ell_i}) - \mathbb{V}(P_{s_h^k,a_h^k,b_h^k,h},V_{h+1}^{\pi^k})\right) + \left(\underline{\nu}_h^{\mathrm{ref},k} - \frac{1}{n_h^k}\sum_{i=1}^{n_h^k}\mathbb{V}(P_{s_h^k,a_h^k,b_h^k,h},\underline{V}_{h+1}^{\mathrm{ref},\ell_i})\right)
$$

$$
\leq \frac{1}{n_h^k}\sum_{i=1}^{n_h^k}\left(\mathbb{V}(P_{s_h^k,a_h^k,b_h^k,h},\underline{V}_{h+1}^{\mathrm{ref},\ell_i}) - \mathbb{V}(P_{s_h^k,a_h^k,b_h^k,h},V_{h+1}^{\pi^k})\right) + \frac{2H^2SN_0}{n_h^k} + 8H^2\sqrt{\frac{\iota}{n_h^k}} \tag{87}
$$

$$
\leq \frac{4H}{n_h^k}\sum_{i=1}^{n_h^k}\left|P_{s_h^k,a_h^k,b_h^k,h}(\underline{V}_{h+1}^{\mathrm{ref},\ell_i} - V_{h+1}^{\pi^k})\right| + \frac{2H^2SN_0}{n_h^k} + 8H^2\sqrt{\frac{\iota}{n_h^k}}
$$

$$
= \frac{4H}{n_h^k}\sum_{i=1}^{n_h^k}\left|P_{s_h^k,a_h^k,b_h^k,h}(\underline{V}_{h+1}^{\mathrm{ref},\ell_i} - V_{h+1}^{\pi^k} + V_{h+1}^* - V_{h+1}^*) - H\left(\beta + \frac{HSN_0}{\check{n}_h^k}\right) + H\left(\beta + \frac{HSN_0}{\check{n}_h^k}\right)\right|
$$

$$
\qquad + \frac{2H^2SN_0}{n_h^k} + 8H^2\sqrt{\frac{\iota}{n_h^k}}
$$

$$
\leq \frac{4H}{n_h^k}\sum_{i=1}^{n_h^k}P_{s_h^k,a_h^k,b_h^k,h}(V_{h+1}^* - \underline{V}_{h+1}^{\mathrm{ref},\ell_i}) + \frac{4H}{n_h^k}\sum_{i=1}^{n_h^k}P_{s_h^k,a_h^k,b_h^k,h}\left(V_{h+1}^* - V_{h+1}^{\pi^k} + H\left(\beta + \frac{HSN_0}{\check{n}_h^k}\right)\right)
$$

$$
\qquad + \frac{4H^2}{n_h^k}\left(\beta + \frac{HSN_0}{\check{n}_h^k}\right) + \frac{2H^2SN_0}{n_h^k} + 8H^2\sqrt{\frac{\iota}{n_h^k}} \tag{88}
$$

$$\leq \frac{4H}{n_h^k}\sum_{i=1}^{n_h^k}(V_{h+1}^* - \underline{V}_{h+1}^{\mathrm{ref},\ell_i})(s_{h+1}^{\ell_i}) + \frac{4H}{n_h^k}\sum_{i=1}^{n_h^k}\left((V_{h+1}^* - V_{h+1}^{\pi^k})(s_{h+1}^{\ell_i}) + H\left(\beta + \frac{HSN_0}{\check{n}_h^k}\right)\right)$$

$$+ \frac{4H^2}{n_h^k}\left(\beta + \frac{HSN_0}{\check{n}_h^k}\right) + \frac{2H^2 SN_0}{n_h^k} + 20H^2\sqrt{\frac{\iota}{n_h^k}} \qquad (89)$$

$$\leq \left(4H\beta + \frac{4H^2 SN_0}{n_h^k}\right) + \frac{8H^2}{n_h^k}\left(\beta + \frac{HSN_0}{\check{n}_h^k}\right)$$

$$+ \frac{4H^2}{n_h^k}\left(\beta + \frac{HSN_0}{\check{n}_h^k}\right) + \frac{2H^2 SN_0}{n_h^k} + 20H^2\sqrt{\frac{\iota}{n_h^k}} \qquad (90)$$

$$= 4H\beta + \frac{12H^2\beta}{n_h^k} + 20H^2\sqrt{\frac{\iota}{n_h^k}} + \frac{6H^2 SN_0}{n_h^k} + \frac{12H^3 SN_0}{n_h^k \check{n}_h^k}$$

$$\leq 4H\beta + \frac{12H^2\beta + 18H^3 SN_0}{n_h^k} + 20H^2\sqrt{\frac{\iota}{n_h^k}},$$

where (87) follows from (86), (88) follows from Lemma E.1 and Lemma H.1, (89) holds with probability at least $1 - 2\delta$ by Azuma's inequality, and (90) follows from Lemma E.1 and Lemma H.1.

## H.5 Summarizing Terms $T_1$-$T_4$ Together

Recall that $\beta = \frac{1}{\sqrt{H}}$, and $N_0 = \frac{c_4 SABH^5\iota}{\beta^2} = O(SABH^6\iota)$. By combining (54), (58), (64), (68) and (83), we conclude that with probability at least $1 - O(H^2 T^4)\delta$, the following bound holds:

$$\sum_{h=1}^{H}\sum_{k=1}^{K}(1 + \frac{1}{H})^{h-1}\Lambda_{h+1}^k$$

$$\leq O(\log T)\cdot(H^2 SN_0 + H\sqrt{T\iota}) + O(H\sqrt{SABT\iota})$$

$$+ O\big(\sqrt{SABH^2 T\iota} + \sqrt{SABH^2\beta T\iota} + \sqrt{SABH^3\beta^2 T\iota}$$

$$+ S^{\frac{3}{2}}ABH^3 N_0^{\frac{1}{2}}\iota\log T + SABH^2\beta^{\frac{1}{2}}\iota^{\frac{1}{2}}\log T + (SAB\iota)^{\frac{3}{4}}H^{\frac{5}{2}}T^{\frac{1}{4}}\big)$$

$$= O\left(\sqrt{SABH^2 T\iota} + H\sqrt{T\iota}\log T + (SAB\iota)^{\frac{3}{4}}H^{\frac{5}{2}}T^{\frac{1}{4}}\right)$$

$$+ O\left(\sqrt{SABH^2\beta T\iota} + \sqrt{SABH^3\beta^2 T\iota} + SABH^2\beta^{\frac{1}{2}}\iota^{\frac{1}{2}}\log T\right)$$

$$+ O\left((H^2 SN_0 + S^{\frac{3}{2}}ABH^3 N_0^{\frac{1}{2}}\iota)\log T\right)$$

$$= O\left(\sqrt{SABH^2 T\iota} + H\sqrt{T\iota}\log T + S^2(AB)^{\frac{3}{2}}H^8\iota^{\frac{3}{2}}T^{\frac{1}{4}}\right). \qquad (91)$$

## I Proof of Lemma E.6 (Final Step)

Our construction of the correlated policy is inspired by the "certified policies" in [5].

Based on the trajectory of the distributions $\{\pi_h^k\}_{h\in[H],k\in[K]}$ specified by Algorithm 3, we construct a correlated policy $\widehat{\pi}_h^k = \widehat{\mu}_h^k \times \widehat{\nu}_h^k$ for each $(h,k) \in [H]\times[K]$. The max-player's policies $\widehat{\mu}_h^k$ and $\widehat{\mu}_{h+1}^k[s,a,b]$ are defined in Algorithm 4, and the min-player's policies can be defined similarly. Further, we define the final output policy $\pi^{\mathrm{out}}$ in Algorithm 2, which first uniformly samples an index $k$ from $[K]$, and then proceeds with $\widehat{\pi}_1^k$. We remark that based on Algorithm 4 and Algorithm 5, the policies $\widehat{\mu}_h^k, \widehat{\nu}_h^k, \widehat{\mu}_{h+1}^k[s,a,b], \widehat{\nu}_{h+1}^k[s,a,b]$ do not depend on the history before step $h$. Therefore, the action-value functions are well-defined for the corresponding steps.

In order to show Lemma E.6, it suffices to show the following inequalities

$$\overline{Q}_h^k(s,a,b) \geq Q_h^{\dagger,\widehat{\nu}_{h+1}^k[s,a,b]}(s,a,b), \quad \overline{V}_h^k(s) \geq V_h^{\dagger,\widehat{\nu}_h^k}(s),$$

$$\underline{Q}_h^k(s,a,b) \geq Q_h^{\widehat{\mu}_{h+1}^k[s,a,b],\dagger}(s,a,b), \quad \underline{V}_h^k(s) \geq V_h^{\widehat{\mu}_h^k,\dagger}(s).$$

---

**Algorithm 4** Certified policy $\widehat{\mu}_h^k$ (max-player version)

---

1: Initialize $k' \leftarrow k$.
2: **for** step $h' \leftarrow h, h+1, \ldots, H$ **do**
3:      Receive $s_{h'}$, and take action $a_{h'} \sim \mu_h^{k'}(\cdot|s_{h'})$.
4:      Observe $b_{h'}$, and sample $j \leftarrow \mathrm{Unif}([N_{h'}^{k'}(s_{h'}, a_{h'}, b_{h'})])$
5:      Set $k' \leftarrow \check{\ell}_{h',j}^{k'}$.

---

**Algorithm 5** Policy $\widehat{\mu}_{h+1}^k[s, a, b]$ (max-player version)

---

1: Sample $j \leftarrow \mathrm{Unif}([N_h^k(s, a, b)])$
2: $k' \leftarrow \check{\ell}_{h,j}^k$.
3: **for** step $h' \leftarrow h+1, \ldots, H$ **do**
4:      Receive $s_{h'}$, and take action $a_{h'} \sim \mu_h^{k'}(\cdot|s_{h'})$.
5:      Observe $b_{h'}$, and sample $j \leftarrow \mathrm{Unif}([N_{h'}^{k'}(s_{h'}, a_{h'}, b_{h'})])$
6:      Set $k' \leftarrow \check{\ell}_{h',j}^{k'}$.

---

due to the definition of output policy in Algorithm 2.

Consider a fixed tuple $(s, a, b, h, k)$. Note that the result clearly holds for any $s, a, b$ that is in its first stage, due to our initialization of $\overline{Q}_h^k(s, a, b), \underline{Q}_h^k(s, a, b)$ and $\overline{V}_h^k(s), \underline{V}_h^k(s)$. In the following, we focus on the case where those values have been updated at least once before the $k$-th episode.

Our proof is based on induction on $k$. Note first that the claim clearly holds for $k = 1$. For $k \geq 2$, assume the claim holds for all $u \in [1 : k-1]$. If those values are not updated in the $k$-th episode, then the claim clearly holds.In the following, we consider the case where those values has just been updated.

(I) We show $\overline{Q}_h^k(s, a, b) \geq Q_h^{\dagger, \widehat{\nu}_{h+1}^k[s,a,b]}(s, a, b)$.

Recall the update rule of the optimistic action-value function

$$\overline{Q}_h(s, a, b) \leftarrow \min\left\{ r_h(s, a, b) + \frac{\check{\overline{v}}}{\check{n}} + \gamma, r_h(s, a, b) + \frac{\overline{\mu}^{\mathrm{ref}}}{n} + \frac{\check{\overline{\mu}}}{\check{n}} + \overline{\beta}, \overline{Q}_h^k(s, a, b) \right\}.$$

Besides the last term, there are two non-trivial cases and we will show both of the first two terms are lower-bounded by $Q_h^{\dagger, \widehat{\nu}_{h+1}^k[s,a,b]}(s, a, b)$.

For the first case, we have

$$\overline{Q}_h^k(s, a, b) = r_h(s, a, b) + \frac{1}{\check{n}_h^k} \sum_{i=1}^{\check{n}_h^k} \overline{V}_{h+1}^{\check{\ell}_i}(s_{h+1}^{\check{\ell}_i}) + \gamma_h^k$$

$$\geq r_h(s, a, b) + \frac{1}{\check{n}_h^k} \sum_{i=1}^{\check{n}_h^k} \overline{V}_{h+1}^{\dagger, \widehat{\nu}_{h+1}^{\check{\ell}_i}}(s_{h+1}^{\check{\ell}_i}) + \gamma_h^k \tag{92}$$

$$\geq \frac{1}{\check{n}_h^k} \sum_{i=1}^{\check{n}_h^k} \overline{Q}_h^{\dagger, \widehat{\nu}_{h+1}^{\check{\ell}_i}}(s, a, b) \tag{93}$$

$$\geq \sup_\mu \frac{1}{\check{n}_h^k} \sum_{i=1}^{\check{n}_h^k} \overline{Q}_h^{\mu, \widehat{\nu}_{h+1}^{\check{\ell}_i}}(s, a, b) \tag{94}$$

$$\geq \overline{Q}_h^{\dagger, \widehat{\nu}_{h+1}^k[s,a,b]}(s, a, b), \tag{95}$$

where (92) follows from the induction hypothesis, (93) follows from the Azuma's inequality, (94) follows from the fact that taking the maximum out of the summation does not increase the sum, and

(95) follows from the construction of policy $\widehat{\nu}_{h+1}^k[s, a, b]$ (obtained via the min-player's counterpart of Algorithm 5).

For the second case,

$$\overline{Q}_h^k(s, a, b) = r_h(s, a, b) + \frac{1}{n_h^k} \sum_{i=1}^{n_h^k} \overline{V}_{h+1}^{\text{ref}, \ell_i}(s_{h+1}^{\ell_i}) + \frac{1}{\check{n}_h^k} \sum_{i=1}^{\check{n}_h^k} \left( \overline{V}_{h+1}^{\check{\ell}_i} - \overline{V}_{h+1}^{\text{ref}, \check{\ell}_i} \right)(s_{h+1}^{\check{\ell}_i}) + \overline{\beta}_h^k$$

$$\geq r_h(s, a, b) + P_h \left( \frac{1}{\check{n}_h^k} \sum_{i=1}^{\check{n}_h^k} \overline{V}_{h+1}^{\check{\ell}_i} \right)(s, a, b) + \chi_1 + \chi_2 + \overline{\beta}_h^k$$

$$\geq r_h(s, a, b) + P_h \left( \frac{1}{\check{n}_h^k} \sum_{i=1}^{\check{n}_h^k} \overline{V}_{h+1}^{\dagger, \widehat{\nu}_{h+1}^{\check{\ell}_i}} \right)(s, a, b) \tag{96}$$

$$= \frac{1}{\check{n}_h^k} \sum_{i=1}^{\check{n}_h^k} \overline{Q}_h^{\dagger, \widehat{\nu}_{h+1}^{\check{\ell}_i}}(s, a, b)$$

$$\geq \sup_{\mu} \frac{1}{\check{n}_h^k} \sum_{i=1}^{\check{n}_h^k} \overline{Q}_h^{\mu, \widehat{\nu}_{h+1}^{\check{\ell}_i}}(s, a, b) \tag{97}$$

$$\geq \overline{Q}_h^{\dagger, \widehat{\nu}_{h+1}^k[s, a, b]}(s, a, b), \tag{98}$$

where

$$\chi_1(k, h) = \frac{1}{n} \sum_{i=1}^n \left( \overline{V}_h^{\text{ref}, \ell_i}(s_{h+1}^{\ell_i}) - \left( P_h \overline{V}_{h+1}^{\text{ref}, \ell_i} \right)(s, a, b) \right),$$

$$\overline{W}_{h+1}^\ell = \overline{V}_{h+1}^\ell - \overline{V}_{h+1}^{\text{ref}, \ell}$$

$$\chi_2(k, h) = \frac{1}{\check{n}} \sum_{i=1}^{\check{n}} \left( \overline{W}_{h+1}^{\check{\ell}_i}(s_{h+1}^{\check{\ell}_i}) - \left( P_h \overline{W}_{h+1}^{\check{\ell}_i} \right)(s, a, b) \right).$$

Here, (96) follows from the concentration result $\overline{\beta} \geq \chi_1 + \chi_2$ (see (29)), (97) follows from the fact that taking the maximum out of summation does not increase the sum, and (98) follows from the construction of policy $\widehat{\nu}_{h+1}^k[s, a, b]$ (obtained via the min-player's counterpart of Algorithm 5).

(II) We show $\overline{V}_h^{k+1}(s) \geq V_h^{\dagger, \check{\nu}_h^k}(s)$.

Note that

$$\overline{V}_h^k(s) = (\mathbb{D}_{\pi_h^k} \overline{Q}_h^k)(s) \geq \sup_{\mu} (\mathbb{D}_{\mu \times \nu_h^k} \overline{Q}_h^k)(s)$$

$$\geq \sup_{\mu} \mathbb{E}_{a \sim \mu, b \sim \nu_h^k} Q_h^{\dagger, \widehat{\nu}_{h+1}^k[s, a, b]}(s, a, b) = V_h^{\dagger, \check{\nu}_h^k}(s),$$

where the first inequality follows from the property of the CCE oracle and the second inequality follows from the induction hypothesis.

The other side of bounds can be proved similarly for $\underline{Q}_h^k(s, a, b)$, $Q_h^{\check{\mu}_{h+1}^k[s, a, b], \dagger}(s, a, b)$, $\underline{V}_h^k(s)$, and $Q_h^{\check{\mu}_{h+1}^k, \dagger}(s)$.

## J    SUPPORTING LEMMAS

**Lemma J.1** (Azuma-Hoefdding's inequality). *Suppose $\{X_k\}_{k \geq 0}$ is a martingale and $|X_k - X_{k-1}| \leq c_k$ almost surely. Then, for all positive integers $N$ and all positive $\epsilon$, it holds that*

$$\mathbb{P}[|X_N - X_0| \geq \epsilon] \leq 2 \exp \left( -\frac{\epsilon^2}{2 \sum_{k=1}^N c_k^2} \right).$$

**Lemma J.2** (Lemma 10 in [40]). *Let $\{M_n\}_{n \geq 0}$ be martingale such that $M_0 = 0$ and $|M_n - M_{n-1}| \leq c$ for some $c > 0$ and any $n \geq 1$. Let $\text{Var}_n = \sum_{k=1}^n \mathbb{E}[(M_k - M_{k-1})^2 | \mathcal{F}_{k-1}]$ for $n \geq 0$, where $\mathcal{F}_k = \sigma(M_1, M_2, \ldots, M_k)$. Then for any positive integer $n$, and any $\epsilon, p > 0$, we have*

$$\mathbb{P}\left[|M_n| \geq 2\sqrt{\text{Var}_n \log \frac{1}{p}} + 2\sqrt{\epsilon \log \frac{1}{p}} + 2c \log \frac{1}{p}\right] \leq \left(\frac{2nc^2}{\epsilon} + 2\right) p.$$

**Lemma J.3** (Variant of Lemma 11 in [40]). *For any $\alpha \in (0, 1)$ and non-negative weights $\{w_h(s, a)\}_{s \in \mathcal{S}, a \in \mathcal{A}, b \in \mathcal{B}, h \in [H]}$, it holds that*

$$\sum_{k=1}^K \sum_{h=1}^H \frac{w_h(s_h^k, a_h^k, b_h^k)}{(n_h^k)^\alpha} \leq \frac{2^\alpha}{1 - \alpha} \sum_{s,a,b,h} w_h(s, a, b)(N_h^{K+1}(s, a, b))^{1-\alpha},$$

$$\sum_{k=1}^K \sum_{h=1}^H \frac{w_h(s_h^k, a_h^k, b_h^k)}{(\check{n}_h^k)^\alpha} \leq \frac{2^{2\alpha} H^\alpha}{1 - \alpha} \sum_{s,a,b,h} w_h(s, a, b)(N_h^{K+1}(s, a, b))^{1-\alpha}.$$

*In the case $\alpha = 1$, it holds that*

$$\sum_{k=1}^K \sum_{h=1}^H \frac{w_h(s_h^k, a_h^k, b_h^k)}{n_h^k} \leq 2 \sum_{s,a,b,h} w_h(s, a, b) \log(N_h^{K+1}(s, a, b)),$$

$$\sum_{k=1}^K \sum_{h=1}^H \frac{w_h(s_h^k, a_h^k, b_h^k)}{\check{n}_h^k} \leq 4H \sum_{s,a,b,h} w_h(s, a, b) \log(N_h^{K+1}(s, a, b)).$$