# OpenReview forum: "Improving Sample Efficiency of Model-Free Algorithms for Zero-Sum Markov Games"
_ICLR.cc/2024/Conference — Submitted to ICLR 2024_

### Official Review · Reviewer_WHVB · 2023-10-29

**Soundness:** 3 good
**Presentation:** 2 fair
**Contribution:** 3 good
**Rating:** 5
**Confidence:** 3

**Summary:**

This paper considers MARL in 2p0s Markov games and proposes the first model-free algorithm w/ optimal dependency on H, which is only achieved by model-based approaches before.

**Strengths:**

1. The contribution is interesting for the MARL literature as model-free algorithms consumes much smaller memory than model-based ones.

2. The min-gap policy looks like an intersting innovation.

**Weaknesses:**

1. The description of technical innovations is not so clear. In the end of page 5, I cannot understand how V^ref helps. The authors shall elaborate why this overcomes the 1/H factor. (BTW, is the V bar in the last line V bar_h instead?)

2. The result still suffers from the curse of multiagents (ie AB instead of A+B). The authors discussed it in the end of Sec 4.1. While understanding it's technically hard, it is still a pity.

Minor: The paper is using (1) for both equations and citations. It's sometimes confusing.

**Questions:**

1. Is the min-gap policy completely novel? Given the maximizing and minimizing agent in 2p0s games, it sounds straightforward. It also appears in the gap-dependent RL literature (eg Xu et al COLT'21). If possible, it would be nice to discuss more related works with a (seemingly) similar technique and to distinguish your version to theirs.

2. see weakness 1.

I'm giving a 5 because of unabling to understand the main technical contribution. I'd be happy to reevaluate after the authors' feedback.

---

> ### Author Response · Authors · 2023-11-20
> **Reply to reviewer WHVB: part I**
>
> We appreciate the effort and time of the reviewer. The main concern is that the main technical contribution is unclear to the reviewer. In our general response, we recap our novelty in algorithm design and theoretical analysis, and we have revised our paper to emphasize the technical contribution. We sincerely request the reviewer to re-evaluate our work based on the revision, and we are open to further discussions.
>
> Q: The description of technical innovations is not so clear.
>
> A: Thanks for the question. We elaborate the novelty in the algorithm design and theoretical analysis in the general response, and we also incorporate it in our revised version.
>
> Q: In the end of page 5, I cannot understand how $V^{ref}$ helps. The authors shall elaborate why this overcomes the $1/H$ factor.
>
> A: Generally speaking, an accurate estimate $V^{ref}$ provides a better estimate of $V-V^{ref}$ in comparison to directly estimating $V$. First of all, we show that learning $\beta$-optimal ($\beta=O(1/H)$) reference value functions $V^{ref}$ requires $O(SABH^5/\beta^2)$ samples, which is independent of the number $K$ of episodes (Corollary 4.5). Then, with an access to such a $\beta$-optimal reference value functions $V^{ref}$, applying concentration inequality to the term $V-V^{ref}$ results in a better sample complexity as their difference is upper bounded by $\beta$, compared with applying the concentration directly to $V$ whose value is upper bounded by $H$. The above two facts altogether offset the weakness of using only $O(H)$ fraction of data.
>
> Q: BTW, is the $\bar{V}$ in the last line $\bar{V}_h$ instead?
>
> A: We thank the reviewer for the careful review. The $\bar{V}$ in the last line is $\bar{V}_h$. We fixed the notation in our revision.
>
> Q: The result still suffers from the curse of multiagents (i.e. $AB$ instead of $A+B$). The authors discussed it in the end of Sec 4.1. While understanding it's technically hard, it is still a pity.
>
> A: Thanks for the comment. We agree that it is very important to investigate the "curse of multi-agency", for example, in our context, to understand whether the reference-advantage decomposition technique can be used to further improve the sample complexity by designing a V-learning type algorithm (which has been used in the literature to deal with multi-agency). Below, we discuss our initial thoughts and potential technical challenges we expect along this direction. A main step is to first incorporate reference-advantage decomposition into the update of optimistic value functions, i.e., we replace the standard update rule (for both optimistic and pessimistic value functions) $V_h(s)=r(s)+P_h V_{h+1}(s)+bonus$ by the following reference advantage-based update rule $V_h(s)=r_h(s)+P_hV_{h+1}^{ref}+P_h(V_{h+1}-V_{h+1}^{ref})+bonus$. Moreover, reference value functions will be updated ($\beta$-approximate $V^{ref}$ are learned) once sufficient samples are collected (similar to lines 14-15 in our algorithm). Further, the success of V-learning relies on the adversarial bandit subroutine where the loss function is $l=(\beta+1-r-V+V^{ref})/(\beta+1)$. Main challenge here is to ensure that the term handled by such adversarial subroutine benefit from the reference-advantage decomposition. Another potential challenge is to incorporate variance reduction technique into adversarial bandit subroutine. We leave this interesting problem as our future work.
>
> Q: Minor: The paper is using (1) for both equations and citations. It's sometimes confusing.
>
> A: Thanks for the comment. We have fixed it in the revised version.

---

> ### Author Response · Authors · 2023-11-20
> **Reply to reviewer WHVB: part II**
>
> Q: Is the min-gap policy completely novel? Given the maximizing and minimizing agent in 2p0s games, it sounds straightforward. It also appears in the gap-dependent RL literature (eg Xu et al COLT'21 [1]). If possible, it would be nice to discuss more related works with a (seemingly) similar technique and to distinguish your version to theirs.
>
> A: Thanks for the question. To our best knowledge, our work is the first to propose the {\bf min-gap} based reference-advantage decomposition.
>
> We would like to clarify the difference between the **min-gap** in our context and the {\bf gap} in gap-dependent RL. In our context, the {\bf min-gap} refers to the smallest gap between the pair of optimistic and pessimistic V functions in the history. We use min-gap based approach to facilitate the reference-advantage decomposition, a variance reduction technique. Recently, gap-dependent RL has become increasingly popular, where the **gap** of a state-action $(s,a)$ refers to the suboptimality gap of action $a$ at the state $s$ (i.e., the advantage function at $(s,a)$). A relevant notion called smallest gap in gap-dependent RL is simply the smallest gap among all state-action pairs, which often appears in the final regret result in gap-dependent RL. To conclude, the **min-gap** in our context and the {\bf gap} in gap-dependent RL are two different concepts and the techniques are incomparable. In our revised version, we make a footnote in page 5 to clarify this point.
>
> [1] Haike Xu, Tengyu Ma, and Simon Shaolei Du. Fine-Grained Gap-Dependent Bounds for Tabular MDPs via Adaptive Multi-Step Bootstrap. In proceedings of the 34th Annual Conference on Learning Theory, 2021.

---

> ### Author Response · Authors · 2023-11-22
>
> Dear Reviewer,
>
> As the author-reviewer discussion period will end soon, we will appreciate it if you could check our response to your review comments. This way, if you have further questions and comments, we can still reply before the author-reviewer discussion period ends. If our response resolves your concerns, we kindly ask you to consider raising the rating of our work. Thank you very much for your time and efforts!

---

### Official Review · Reviewer_GLai · 2023-10-29

**Soundness:** 3 good
**Presentation:** 3 good
**Contribution:** 3 good
**Rating:** 6
**Confidence:** 3

**Summary:**

For the first time, the paper proves a model-free algorithm for two-player zero-sum Markove games can achieve the same sample complexity as the best model-based algorithm. To achieve this, the paper proposes a method that features reference-advantage decomposition and a novel design of updating the reference value functions as the pair of optimistic and pessimistic value functions.

**Strengths:**

1. The contribution of the paper is unique and may have a broad impact. Although the work is based on recent work in single-agent RL, this paper solves critical technical difficulties in applying similar analysis to multi-player Markov games.
2. The proposed method and the theoretical analysis seem sound to me.
3. The paper is well written, although it is not easy to follow due to dense notation.

**Weaknesses:**

1. Since the single-agent work [1] is very related, it would be better if a short introduction about it is given.
2. Some intuitive explanations may be needed for readers to catch the general idea before section 3, which discusses the details of the algorithms directly. It would be even better if there were examples.

[1] Zhang, Zihan, Yuan Zhou, and Xiangyang Ji. "Almost optimal model-free reinforcement learningvia reference-advantage decomposition." Advances in Neural Information Processing Systems 33 (2020): 15198-15207.

**Questions:**

None

---

> ### Author Response · Authors · 2023-11-20
> **Reply to reviewer GLai**
>
> We thank the reviewer for the constructive suggestion. We have posted the revised version to improve the clarity of the paper. We will be happy to answer any follow-up questions.
>
> Q: Since the single-agent work [1] is very related, it would be better if a short introduction about it is given.
>
> A: Thanks for the suggestion! In our revision, we briefly introduce reference-advantage decomposition in single-agent RL [1] to help the reader digest the technical contributions of our work.
>
> [1] Zihan Zhang, Yuan Zhou, and Xiangyang Ji. Almost optimal model-free reinforcement learning via reference-advantage decomposition. In Proceedings of the 34th International Conference on Neural Information Processing Systems, 2020.
>
> Q: Some intuitive explanations may be needed for readers to catch the general idea before section 3, which discusses the details of the algorithms directly. It would be even better if there were examples.
>
> A: Thanks for the comment. We are aware that the complicated algorithm (Algorithm 1) may be hard to follow and may not be the efficient way to convey the general idea. In our revision, We presented a schematic algorithm and defer the details of the algorithms to the appendix. Besides, in the revision, we give a short introduction to the reference-advantage decomposition in single-agent RL [1], elaborate the difficulty and our contribution of extending such a technique to zero-sum Markov game before introducing our main algorithm.
>
> [1] Zihan Zhang, Yuan Zhou, and Xiangyang Ji. Almost optimal model-free reinforcement learning via reference-advantage decomposition. In Proceedings of the 34th International Conference on Neural Information Processing Systems, 2020.

---

### Official Review · Reviewer_iTn4 · 2023-11-01

**Soundness:** 3 good
**Presentation:** 4 excellent
**Contribution:** 3 good
**Rating:** 6
**Confidence:** 4

**Summary:**

The purpose of the paper is to provide a model-free algorithm for two-player zero-sum Markov games that exhibits best case dependence on H, which is the horizon as well as S which is the size of the state space. The paper notes that it is the first work to achieve this type of optimality. The paper is based on Q-learning, where its main novelty is in introducing a variance reduction technique for single-player reinforcement learning. However, because the games does not exhibit monotonicity the way single-player systems do, the authors introduce the use of two estimates of value functions: optimistic value functions and pessimistic value functions. The general framework they use comes from the Nash Q-learning framework, but that framework does not provide optimality bounds with respect to the horizon which is the subject of the current work.

**Strengths:**

The authors provide an algorithm for two-player zero-sum games with self-play that improves upon best known results. The lower bound for two-player zero-sum Markov games is Omega(H^3S(A+B)/eps^2) . The work of "Near-Optimal Reinforcement Learning with Self-Play" gives bounds that are O(H^5SABi/eps^2). The present work gives a bound O(H^3SAB/eps^2), which is a large improvement when the horizon is very large. Another strength is the novelty of the proof technique. The proof technique draws upon a variance reduction technique in single player settings. In order to adequately incorporate the variance reduction technique, the authors store optimistic and pessimistic estimates of the optimal value function, rather than only optimistic which is used in the single player setting. Doing this allows the authors to obtain convergence guarantees.

**Weaknesses:**

I think that it would be helpful to provide a comparison to the proof techniques used in "Near-Optimal Reinforcement Learning with Self-Play". This is because the algorithm is largely inspired by their algorithm. Additionally, both algorithms use upper and lower bounds for the value function (I'm not sure if I'm using the term here in the right way), which is one of the crucial pieces that allows the algorithm of the current work to get the optimality in H bound. In this way, the key novelty would be clearer.
I also think that because the algorithm builds upon another algorithm and modifies the bound to make the dependence on H stronger by a factor of H^2, it may be useful to incorporate more than one theorem to show extensions such as incorporating the use of function approximation etc. Or perhaps the authors could look at an extension where the agents play against each other.

**Questions:**

I really like your results. However, I have a question regarding your proof technique. I understand that there is a monotonicity assumption that is required to be overcome in order for single player reinforcement learning settings to be applied to multi-agent reinforcement learning. Prior works have used various properties to overcome this monotonicity assumption (see the work of https://arxiv.org/abs/2303.09716 for example). However, to me it is not immediately clear how simply storing reference values of optimistic and pessimistic value functions overcomes the monotonicity assumption. This is because when there is no noise, i.e., perfect estimates of the value functions, then that would mean that there would be no need for an optimistic or pessimistic bound and only one bound would be needed. But if that is the case, when the monotonicity issue still would not go away, since the monotonicity issue inevitably arises in games versus single player settings. So then I'm confused what is the exact role of the optimistic and pessimistic estimates of the optimal value function.
Another question I have is what is the difference between your proof techniques and those in "Near-Optimal Reinforcement Learning with Self-Play". This is because they also use optimistic and pessimistic estimates of the optimal value function. However, their bounds are clearly less tight than yours since they are of order H^3. Could you please provide a comparison to that work or point me to where you discuss it in the paper?

---

> ### Author Response · Authors · 2023-11-20
> **Reply to reviewer iTn4: part I**
>
> We thank the reviewer for the helpful comments. It seems the concern lies in the proof techniques and the misunderstanding of the monotonicity in this context. We have posted a revised version to further clarify the proof techniques as well as min-gap based approach (regarding monotonicity). We notice that the reviewer iTn4 gives decent scores to our paper (soundness 3/presentation 4/contribution 3), but the final rating is below the threshold. We kindly ask the reviewer to consider raising the score of your evaluation if our response and revised version resolves your concerns to a satisfactory level. We would be glad to answer any follow-up questions.
>
>
> Q: I think that it would be helpful to provide a comparison to the proof techniques used in "Near-Optimal Reinforcement Learning with Self-Play" [1]. This is because the algorithm is largely inspired by their algorithm. Additionally, both algorithms use upper and lower bounds for the value function (I'm not sure if I'm using the term here in the right way), which is one of the crucial pieces that allows the algorithm of the current work to get the optimality in $H$ bound. In this way, the key novelty would be clearer.
>
> A: Thanks for the great suggestion! Precisely speaking, our algorithm is inspired by both [1] and [2]. We propose a novel min-gap based reference-advantage decomposition technique for zero-sum Markov game, which is essential for improving the sample complexity in [1]. In our revision, we provide a comparison to the proof techniques used in [1] and [2] in Appendix B.
>
> [1] Yu Bai, Chi Jin, and Tiancheng Yu. Near-optimal reinforcement learning with self-play. In Advances in Neural Information Processing Systems, 2020.
>
> [2] Zihan Zhang, Yuan Zhou, and Xiangyang Ji. Almost optimal model-free reinforcement learning via reference-advantage decomposition. In Proceedings of the 34th International Conference on Neural Information Processing Systems, 2020.
>
>
> Q: I also think that because the algorithm builds upon another algorithm and modifies the bound to make the dependence on $H$ stronger by a factor of $H^2$, it may be useful to incorporate more than one theorem to show extensions such as incorporating the use of function approximation etc. Or perhaps the authors could look at an extension where the agents play against each other.
>
> A: We thank the reviewer for pointing out the potential extension of our work! Our work focuses on the tabular case and Q-learning algorithms. We expect variance reduction techniques for function approximation in single-agent (e.g. [2]) could be used in Markov games, but they are in sharp contrast to the reference-advantage decomposition for tabular case. We will leave this as our future work.
>
>
> [2] Yi Fei, Tianhao Wang, Dongruo Zhou, and Quanquan Gu. Variance-Aware Off-Policy Evaluation with Linear Function Approximation,  In Advances in Neural Information Processing Systems, 2021.

---

> ### Author Response · Authors · 2023-11-20
> **Reply to reviewer iTn4: part II**
>
> Q: To me it is not immediately clear how simply storing reference values of optimistic and pessimistic value functions overcomes the monotonicity assumption. This is because when there is no noise, i.e., perfect estimates of the value functions, then that would mean that there would be no need for an optimistic or pessimistic bound and only one bound would be needed. But if that is the case, when the monotonicity issue still would not go away, since the monotonicity issue inevitably arises in games versus single player settings. So then I'm confused what is the exact role of the optimistic and pessimistic estimates of the optimal value function.
>
> A: We would like to clarify monotonicity in the context of this work. In this work, we keep track of the value difference between the optimistic and pessimistic value functions in each iteration, and only the pair of optimistic and pessimistic value functions with the smallest gap are recorded (Algorithm 3 line 17-20). The monotonicity in this paper refers to the non-increasing gap of the recorded pair of optimistic and pessimistic value functions during the iteration. Thanks to the novel min-gap based reference-advantage decomposition, we ensure the monotonicity of the the pair of recorded optimistic and pessimistic value functions, which is the key to obtain an accurate reference value and improve the horizon dependence.
>
> Going to the example of perfect estimate of the value functions, in this case, only one bound is needed and the gap between the optimistic and pessimistic value functions remains 0 during the whole learning process, which is also non-increasing.
>
> Q: Another question I have is what is the difference between your proof techniques and those in "Near-Optimal Reinforcement Learning with Self-Play" [1]. This is because they also use optimistic and pessimistic estimates of the optimal value function. However, their bounds are clearly less tight than yours since they are of order $H^3$. Could you please provide a comparison to that work or point me to where you discuss it in the paper?
>
> A: We highlight the main differences of our algorithm from Nash Q-learning [1] in the following (and also in Appendix B in our revised version). (i) The main difference is that we incorporate **variance reduction** technique -- reference-advantage decomposition, into the update of the value functions, which results in an improved horizon dependence. Note that our algorithm does not take the original algorithm in [3] directly, but with a novel design of the min-gap based scheme. Specifically, unlike the single-agent scenario [3], the optimistic (or pessimistic) value function in Markov games does not necessarily preserve the monotone property due to the nature of the CCE oracle. In order to obtain the ``best" optimistic and pessimistic value function pair, we proposed to update the reference value functions as the pair of optimistic and pessimistic value functions whose value difference is the smallest (i.e., with the minimal gap) in the history. It turns out that such a design is critical to guarantee the provable sample efficiency. (ii) Nash Q-learning updates the value function with a learning rate, while our algorithm adopts **greedy update**. (iii) We adopt the **stage-based update framework** with the value functions updated at the end of each stage, which induces a lower switching cost. But Nash Q-learning updates value functions at each episode.
>
> We also want to reiterate that due to the incorporation of the new min-gap based technique, our analysis involves several novel developments. Please see our general response above (and Appendix B in our revised version).
>
> [1] Yu Bai, Chi Jin, and Tiancheng Yu. Near-optimal reinforcement learning with self-play. In Advances in Neural Information Processing Systems, 2020.

---

> ### Author Response · Authors · 2023-11-22
>
> Dear Reviewer,
>
> As the author-reviewer discussion period will end soon, we will appreciate it if you could check our response to your review comments. This way, if you have further questions and comments, we can still reply before the author-reviewer discussion period ends. If our response resolves your concerns, we kindly ask you to consider raising the rating of our work. Thank you very much for your time and efforts!

---

### Official Review · Reviewer_iS4i · 2023-11-01

**Soundness:** 2 fair
**Presentation:** 1 poor
**Contribution:** 2 fair
**Rating:** 5
**Confidence:** 2

**Summary:**

The paper studies a model free algorithm to compute equilibria in two players zero-sum finite horizon Markov games. Technically the authors obtain an optimal dependency on the horizon $H$, by combining the reference-advantage decomposition and the CCE collapse technique into a novel Q-learning like algorithm.

**Strengths:**

The work present a model-free algorithm for equilibrium computation in two players zero-sum finite horizon Markov games that matches the best known model-based algorithm for the same setting, and it is suboptimal only in the cardinality of the action space ($O(A^2)$ instead of $O(A)$). Given the recent attention of multi-agent RL this is an important achievement and in line with the interests of the ICLR community.

**Weaknesses:**

The main weakness of the paper is clarity. In my opinion, an accurate revision is essential to significantly enhance the quality of the paper, and high level ideas could be communicated more efficiently. In particular the paper assumes that the reader is familiar with [40] and [36], and without a deep understanding of these works the reader is confronted with high level ideas that are hard to grasp. This leaves the reader unsure on the technical contributions of the paper.

**Questions:**

1) What does "min-gap" exactly refers to? It is never defined formally. Does it refers to $\Delta(s,h)$ which is the minimum gap over all episodes and this ensures monotonicity?
2) You say that using the techniques of [40] for single agent RL is challenging due to the fact the upper bounds are no longer monotonic. It is unclear way this is needed. More precisely, the last two paragraphs on page 4 are really hard to read and do not convey clearly neither the challenges nor the contributions.
3) Algorithm 1 gives too many unnecessary details, and simplifying it would help the reader understand better its workings. In particular, I think lines 9 to 13 are never commented and could be written with $O$ notation or something alike.
4) In algorithm 1 the actions played come from a correlated strategy (line 6)? Can the learning be decoupled by marginalising $\pi$ or the marginalisation has to happen only at the end of the learning procedure?
5) On page 5 the authors refer to a "standard update". What is this standard update and what is "bonus"?
6) It is never explained what CCE has to do with the problem of computing Nash equilibrium. After some tome I arrived at the conclusion that it is the "collapse" idea first introduced in "Cai, Yang, et al. "Zero-sum polymatrix games: A generalization of minmax." Mathematics of Operations Research (2016)" in which the CCE of a $\epsilon$-perturbed zero sum game is an $\epsilon$-CCE of the zero sum games which marginalisation is in turn a $O(\epsilon)$ of the Nash of the zero sum game. I get that this idea was already presented in [36] but the authors should at least presented at a high level this idea.
7) I think you are missing to discuss the decentralised model-free algorithm of "On improving model-free algorithms for decentralised multi-agent reinforcement learning", but there's no issue there as it obtains a $O(H^5)$ sample complexity, although the dependency on $A$ is optimal

---

> ### Author Response · Authors · 2023-11-20
> **Reply to reviewer iS4i: part I**
>
> We appreciate the effort and time of the reviewer! The major concern that affected the reviewer's evaluation is the clarity issue. We posted the revised paper to improve the clarity of the paper and sincerely hope that the reviewer can re-evaluate our work based on the lastest version. We are open to any more discussions.
>
> Q: The main weakness of the paper is clarity. In my opinion, an accurate revision is essential to significantly enhance the quality of the paper, and high level ideas could be communicated more efficiently. In particular the paper assumes that the reader is familiar with [1] and [2], and without a deep understanding of these works the reader is confronted with high level ideas that are hard to grasp. This leaves the reader unsure on the technical contributions of the paper.
>
> A: We thank the reviewer for bringing the clarity issue up! In our general response, we elaborate the technical novelties in algorithm design and analysis. We acknowledge that in the current version, the readers may have difficulty understanding the high level ideas and technical contributions of our work without knowing relevant works [1] and [3]. In our revision, we introduce in greater detail the reference-advantage decomposition in single-agent RL [1], and the difficulties when extending it to two-player zero-sum Markov games.
>
> [1] Zihan Zhang, Yuan Zhou, and Xiangyang Ji. Almost optimal model-free reinforcement learning via reference-advantage decomposition. In Proceedings of the 34th International Conference on Neural Information Processing Systems, 2020.
>
> [2] Qiaomin Xie, Yudong Chen, Zhaoran Wang, and Zhuoran Yang. Learning zero-sum simultaneous-move markov games using function approximation and correlated equilibrium. In Proceedings of Thirty Third Conference on Learning Theory, 2020.
>
> [3] Yu Bai, Chi Jin, and Tiancheng Yu. Near-optimal reinforcement learning with self-play. In Advances in Neural Information Processing Systems, 2020.
>
> Q: What does "min-gap" exactly refers to? It is never defined formally. Does it refers to $\Delta(s,h)$ which is the minimum gap over all episodes and this ensures monotonicity?
>
> A: Your understanding is correct. In Section 3, we presented the min-gap based reference-advantage decomposition in detail and we elaborated the "min-gap" at the end of page 4. We further clarify this point in the revision.
>
> Q: You say that using the techniques of [40] for single agent RL is challenging due to the fact the upper bounds are no longer monotonic. It is unclear why this is needed. More precisely, the last two paragraphs on page 4 are really hard to read and do not convey clearly neither the challenges nor the contributions.
>
> A: Thanks for the helpful comment! We would like to give a detailed explanation in our revision, and we briefly discuss it below.
>
> In single agent RL, the V function preserves the monotonic structure if the corresponding Q function is non-increasing, which is crucial for the reference-advantage decomposition. In two-player zero-sum game, one naive thought is to obtain the non-increasing optimistic V function and non-decreasing pessimistic V function. However, this is no longer possible due to the policy updating by the CCE oracle. Compared to single agent RL where the policy is updated by selecting an greedy action, the policy updated by the CCE oracle does not preserve such a property. In other words, the optimistic and pessimistic V functions are not monotonic even if the corresponding optimistic and pessimistic Q functions are monotonic. To resolve the issue, we observe that if the gap between the pair of optimistic and pessimistic V value functions are monotonic, the reference-advantage decomposition still works. To this end, we propose min-gap based approach to keep track of the smallest gap between the pair of optimistic and pessimistic V functions in the history and record the corresponding pair of V functions. Upon receiving enough samples (Algorithm 1 line 22), the pair of reference V function is then updated as the recorded pair of V functions.

---

> ### Author Response · Authors · 2023-11-20
> **Reply to reviewer iS4i: part II**
>
> Q: Algorithm 1 gives too many unnecessary details, and simplifying it would help the reader understand better its workings. In particular, I think lines 9 to 13 are never commented and could be written with $O$ notation or something alike.
>
> A: Thanks for the great suggestion! We agree that a schematic algorithm would be easier to follow. In the revision, we present the following simplified algorithm in the main paper and defer the detailed version to the appendix. In the simplified algorithm, we keep the stage-based update backbone, and for each stage, we only outline the main steps including sample collection and accumulator update, optimistic and pessimistic value function update and policy update, tracking the pair of min-gap value functions, and reference value function selection.
>
> Q: In algorithm 1 the actions played come from a correlated strategy (line 6)? Can the learning be decoupled by marginalising $\pi$ or the marginalisation has to happen only at the end of the learning procedure?
>
> A: In Algorithm 1 line 6, the actions played come from a correlated strategy. The learning cannot be decoupled by marginalising $\pi$ (as elaborated in the following), and the marginalisation happens at the end of the learning procedure (Algorithm 2).
>
> Our algorithm falls into the popular centralized Q-learning algorithms for Markov game [3]. In Q-learning algorithm, we keep track of the Q functions (of state and joint actions), and the exploration policy $\pi$ computed by the Q functions (by CCE oracle) in each iteration is correlated in nature. However, given the exploration policies, we are able to extract the marginalized policy at the end of the learning process (certified policy in page 6).
>
> [3] Yu Bai, Chi Jin, and Tiancheng Yu. Near-optimal reinforcement learning with self-play. In Advances in Neural Information Processing Systems, 2020.
>
> Q: On page 5 the authors refer to a "standard update". What is this standard update and what is "bonus"?
>
> A: In our original version, we explicitly give the standard update rule (informal here) $Q_h(s,a,b)=r_h(s,a,b)+(P_hV_{h+1})(s,a,b)+bonus$, and the bonus terms in both the standard update and the reference-advantage decomposition are specified in Algorithm line 9-11. In the revision, we clarify the standard update rule in (1) and (2), and the bonus terms are specified in Algorithm 3 line 9-11.
>
> Q: It is never explained what CCE has to do with the problem of computing Nash equilibrium. After some time I arrived at the conclusion that it is the "collapse" idea first introduced in "Cai, Yang, et al. "Zero-sum polymatrix games: A generalization of minmax." Mathematics of Operations Research (2016)" in which the CCE of a $\epsilon$-perturbed zero sum game is an $\epsilon$-CCE of the zero sum games which marginalisation is in turn a $O(\epsilon)$ of the Nash of the zero sum game. I get that this idea was already presented in [2] but the authors should at least presented at a high level this idea.
>
> A: Thanks for pointing out the clarity issue! The equivalence of CCE and NE in two-player zero-sum Markov games is also elaborated in [2] and [4]. We include the relationship between CCE and Nash equilibrium in zero-sum Markov games in our revision.
>
> [2] Qiaomin Xie, Yudong Chen, Zhaoran Wang, and Zhuoran Yang. Learning zero-sum simultaneous-move markov games using function approximation and correlated equilibrium. In Proceedings of Thirty Third Conference on Learning Theory, 2020.
>
> [4] Qinghua Liu, Tiancheng Yu, Yu Bai, and Chi Jin. A sharp analysis of model-based reinforcement learning with self-play. In Proceedings of the 38th International Conference on Machine Learning, 2021.
>
> Q: I think you are missing to discuss the decentralised model-free algorithm of "On improving model-free algorithms for decentralised multi-agent reinforcement learning" [5], but there's no issue there as it obtains a $O(H^5)$ sample complexity, although the dependency on $A$ is optimal.
>
> A: We would like to bring the reviewer's attention that we have made a comparison between our result and [2] in our main result (last paragraph in Section 4.1).
>
> [5] Weichao Mao, Lin Yang, Kaiqing Zhang, and Tamer Basar. On improving model-free algorithms for decentralized multi-agent reinforcement learning. In Proceedings of the 39th International Conference on Machine Learning, 2022.

---

> ### Author Response · Authors · 2023-11-22
>
> Dear Reviewer,
>
> As the author-reviewer discussion period will end soon, we will appreciate it if you could check our response to your review comments. This way, if you have further questions and comments, we can still reply before the author-reviewer discussion period ends. If our response resolves your concerns, we kindly ask you to consider raising the rating of our work. Thank you very much for your time and efforts!

---

### Author Response · Authors · 2023-11-20
**General response**

We thank all reviewers for their effort and time to evaluate our work. We post our revised version and incorporate the reviewers' comments.

Based on reviewers' feedback, we elaborate the novelty in the algorithm design and theoretical analysis in the following (which is incorporated in the revised version).

**Novelty in algorithm design:** Our main novel design idea lies in the **min-gap** based advantage reference value decomposition. Unlike the single-agent scenario [Zhang et al. 2020], the optimistic (or pessimistic) value function in Markov games does not necessarily preserve the monotone property due to the nature of the CCE oracle. In order to obtain the ``best" optimistic and pessimistic value function pair, we propose to update the reference value functions as the pair of optimistic and pessimistic value functions whose value difference is the smallest (i.e., with the minimal gap) in the history. It turns out that such a design is critical to guarantee the provable sample efficiency.

**Novelty in analysis:**  Due to the adoption of the new min-gap based reference-advantage decomposition technique, several new error terms arise in our analysis. Our main development lies in establishing a few new properties on the cumulative occurrence of the large V-gap and the cumulative bonus term, which enable the upper-bounding of those new error terms.
More specifically, as we explain in our proof outline in Section 4.2, our analysis includes the following novel developments. (i) Step I shows that the Nash equilibrium (action-)value functions are always bounded between the optimistic and pessimistic (action-)value functions (see Lemma 4.3). Our new technical development here lies in proving the inequality with respect to the action-value function, whose update rule features the min-gap reference-advantage decomposition. (ii) Step II shows that the reference value can be learned with bounded sample complexity (see Lemma 4.4).
Our new development here lies in handling an additional martingale difference arising due to the CCE oracle. (iii) In step IV, in order to bound the term in line 327, there are a few new developments. First, we need to bound both the optimistic and pessimistic accumulative bonus terms, and the analysis is more refined compared to that for single-agent RL. Second, the analysis of the optimistic accumulative bonus term needs to handle the CCE oracle together with the new min-gap base reference-advantage decomposition for two-player zero-sum Markov game.
We will elaborate these contributions in greater detail to better highlight our new developments.

---

### Meta-Review · Area_Chair_JDG8 · 2023-12-06

**Metareview:**

This paper was rather borderline. Overall, while the reaction of the reviewers has been lukewarm, the idea of the paper of combining Nash-Q learning with equilibrium collapse seems new and worth exploring further. In particular, I think that the framework of analysis that the authors introduce (based on the min-gap) is interesting, and that the combination of techniques they propose to control the error terms in the analysis might become a useful reference for follow-up papers. Initially, the reviewers seemed to have reservations about the novelty of the paper and whether the techniques are too incremental to push for acceptance. The response of the authors did a good job at highlighting the approach. However, only one reviewer was able to verify the new claims of the authors. Since I liked the paper, but this point of incrementally and better positioning within existing analysis technique is so central to this theoretical paper, and only one expert reviewer was able to take a look at the new claims, this paper posed a challenging choice which was brought up to the SAC. After discussion, the SAC and I agreed that the best course of action would be not rush this towards acceptance, to make sure more experts can verify the new claims.

**Justification For Why Not Higher Score:**

While I think some of the ideas in the paper fit well into the growing literature on the topic, and the paper would find an audience of interested people as a poster, it was decided that the paper should go through another round of expert review.

**Justification For Why Not Lower Score:**

N/A

---

### Decision · Program_Chairs · 2024-01-16

Reject